# Adgrg6/Gpr126 is required for compact wall integrity and establishing trabecular identity during cardiac trabeculation

Swati Srivastava [1,8], Felix Gunawan [2,9,10], Silvia Vergarajauregui [1,3], Alessandra Gentile [2,11,12], Miriam Angeloni[4], Sarah C. Petersen [5,13], Stefan Günther [6], Fulvia Ferrazzi [4,7], Didier Y. R. Stainier [2] & Felix B. Engel [1,3] ✉

How adhesion G protein-coupled receptors (aGPCRs) control development remains unclear. aGPCR Adgrg6/Gpr126 has been associated with heart trabeculation. Defects in this process cause cardiomyopathies and cardiac dysfunction. How cardiomyocytes attain trabecular identity is poorly understood. Here, we show that different domains of Gpr126 distinctly regulate compact wall integrity and trabecular identity. Maternal zygotic (MZ) *gpr126*[stl47] early truncation mutants exhibit hypotrabeculation, whereby N-cadherin distributes randomly along apical/basal/lateral membranes of compact layer cardiomyocytes. In contrast, zygotic and MZ *gpr126*[st49] mutants, expressing a N-terminal fragment lacking the GPS motif (NTF[ΔGPS]), exhibit a multilayered ventricular wall containing polarized cardiomyocytes with normal N-cadherin localization and increased Notch activity. Notably, endocardially expressed gpr126 C-terminal fragment (CTF) reinstates trabeculation in *gpr126*[st49] mutants. Collectively, our data reveal domain-specific roles of Gpr126 during trabeculation, whereby the NTF is required for maintaining cell-cell adhesion and compact wall integrity, whereas the CTF is essential to provide trabecular identity.

Adhesion G protein-coupled receptors (aGPCRs) are required for the regulation of various biological and developmental processes[1]. Accordingly, mutations in aGPCRs have been associated with a wide variety of human diseases[1–3]. Moreover, ~34% of all FDA-approved drugs target 108 members of the GPCR superfamily, highlighting the translational accessibility of GPCRs as therapeutic targets[4]. Thus, exploring the roles of understudied GPCRs offers significant unexplored therapeutic opportunities.

aGPCRs are characterized by a seven-transmembrane (7TM) domain and a long extracellular domain (ECD) consisting of various

[1]Department of Nephropathology, Institute of Pathology, Experimental Renal and Cardiovascular Research, Friedrich-Alexander-Universität Erlangen-Nürnberg (FAU), Erlangen, Germany. [2]Developmental Genetics, Max-Planck-Institute for Heart and Lung Research, Bad Nauheim, Germany. [3]Department of Cardiology, Friedrich-Alexander-Universität Erlangen-Nürnberg (FAU), Erlangen, Germany. [4]Institute of Pathology, Friedrich-Alexander-Universität Erlangen-Nürnberg (FAU), Erlangen, Germany. [5]Department of Developmental Biology, Washington University in St. Louis, St. Louis, MO, USA. [6]ECCPS Bioinformatics and Deep Sequencing Platform, Max Planck Institute for Heart and Lung Research, Bad Nauheim, Germany. [7]Department of Nephropathology, Institute of Pathology, Friedrich-Alexander-Universität Erlangen-Nürnberg (FAU), Erlangen, Germany. [8]Present address: Buchmann Institute for Molecular Life Sciences (BMLS), Institute of Cell Biology and Neuroscience, Goethe University Frankfurt, Frankfurt am Main, Germany. [9]Present address: Institute of Cell Biology, ZMBE, Faculty of Medicine, University of Münster, Münster, Germany. [10]Present address: Department of Cell and Systems Biology, Faculty of Arts and Science, University of Toronto, Toronto, ON, Canada. [11]Present address: Centre for Developmental Neurobiology, Institute of Psychiatry, Psychology and Neuroscience, King's College London, London, UK. [12]Present address: MRC Centre for Neurodevelopmental Disorders, King's College London, London, UK. [13]Present address: Department of Neuroscience, Kenyon College, Gambier, OH, USA. ✉e-mail: felix.engel@uk-erlangen.de

adhesion domains. They are autoproteolytically cleaved in the endoplasmic reticulum within a conserved GPCR Autoproteolysis Inducing (GAIN) domain into a C-terminal fragment (CTF) containing the 7TM domain and an N-terminal fragment (NTF) containing the ECD[5,6]. Subsequently, NTF and CTF are transported to the plasma membrane, where they can reconstitute to form a receptor[7,8]. The signaling mechanisms of aGPCRs and their domain-specific functions remain poorly understood. Various models of receptor activation have been proposed[9–11] while there are also reports that aGPCRs use their ECDs to mediate functions in a 7TM-independent manner[12].

Adgrg6/Gpr126 has been reported to be required for the development of several tissues/organs, including the peripheral nervous system, intervertebral disc, ear, and placenta[13–17]. Several genome-wide Association and RNA-sequencing (RNA-Seq) studies have associated variants in the human *GPR126* locus with diseases like adolescent idiopathic scoliosis, intellectual disability, severe arthrogryposis multiplex congenita, periodontitis, chronic obstructive pulmonary disease (COPD), and various human cancers[18–21]. Notably, the cellular basis for these Gpr126-linked syndromes is not well understood.

In the heart, *Gpr126* is known to be transiently expressed in the endocardium in embryonic mice (E9.5 to E13.5)[22,23]. *gpr126* is also expressed in the heart of zebrafish embryos[13,22], but the cellular origin of expression was not determined. Global depletion of *Gpr126* affects cardiac trabeculation in mice[17,22,24] and zebrafish[22]. Waller-Evans et al. described the first *Gpr126* knockout (KO) mouse line, which was characterized by mid-gestation lethality exhibiting circulatory failure like congestion, edema, and internal hemorrhage with secondary defects in trabeculation and a normally developed placenta[24]. Subsequently, two different *Gpr126* KO mouse models were reported exhibiting embryonic lethality at E11.5 associated with hypotrabeculation, myocardial wall thinning, arrhythmia, bradycardia, and mitochondrial defects[22,25]. Morpholino-mediated *gpr126* knockdown in zebrafish also resulted in hypotrabeculation, and rescue experiments using Gpr126-NTF$^{\Delta GPS}$ (NTF$^{\Delta GPS}$) suggested that NTF$^{\Delta GPS}$ is a stable, secreted protein sufficient for heart development[22]. It was further reported that miR-27a/b is a post-transcriptional regulator of Gpr126 and modulation of *gpr126* expression in zebrafish by inducing or inhibiting miR-27b resulted in defective trabeculation[26]. However, in a recent study utilizing a new *Gpr126* KO mouse line, it has been proposed that placental defects are the primary cause of heart abnormalities[17]. The authors also did not observe any trabeculation defects in newly created *gpr126* zebrafish mutants, *gpr126^{bns341}* (predicted early truncation) *and gpr126^{bns342}* (promoter-less allele)[17].

Cardiac trabeculation is a crucial process during ventricular chamber development where the cardiomyocytes in the heart wall organize themselves to form two distinct layers, an outer compact wall and an inner trabecular layer. This process depends on communications between the epicardium, myocardium, and endocardium[27,28]. There are several signaling pathways/molecular regulators important for cardiac trabeculation. First, endocardial Notch signaling, a critical early regulator of trabeculation[29–31], which in zebrafish is activated in the endocardium at 24 hpf and becomes restricted to atrioventricular valves and the outflow tract by 72 hpf[32], regulates Neuregulin (Nrg1 in mice /nrg2a in zebrafish) in the endocardium which mediates ErbB2 signaling in the myocardium[33–38]. Second, cardiomyocyte crowding triggers their delamination during which adherens junctions between cardiomyocytes are remodeled[39]. N-cadherin, the major component of cardiac adherens junctions, re-localizes from the lateral to the basal domain in compact wall cardiomyocytes. In contrast, trabecular cardiomyocytes exhibit a punctate N-cadherin distribution on their entire surface[39]. The first trabeculae appear in the outer curvature of the ventricle at 60 hpf. Third, these delamination events in trabecular cardiomyocytes activate Notch signaling in their neighboring compact wall cardiomyocytes[40–42], which subsequently inhibit their

delamination[42] by suppressing the actomyosin machinery and preserving myocardial wall architecture[42]. The first appearance of Notch activity in ventricular cardiomyocytes overlaps with the onset of trabeculation (60 hpf). Notch-negative cardiomyocytes depolarize and delaminate, adopting a trabecular identity. In contrast to endocardial Notch signaling, the genetic network underlying myocardial Notch-mediated fate specification remains poorly understood.

In this study, we utilized zebrafish *gpr126* genetic models with domain-specific loss and gain of function approaches as well as rescue experiments, to assess the role of Gpr126 in zebrafish heart development. Our data suggest that both the NTF and CTF of Gpr126 are required for trabeculation. The NTF contributes to maintaining compact wall integrity at the onset of trabeculation by maintaining cell-cell junctions. The CTF subsequently confers trabecular identity to cardiomyocytes through modulation of myocardial Notch activity. Altogether, our results provide new insights into the domain-specific functions of Gpr126 and shed light on the regulation of the mechanisms underlying Notch-mediated fate specification during trabeculation.

## Results

### The *stl47* and *st49* mutant alleles produce truncated versions of Gpr126

In order to understand the role of Gpr126 as well as its NTF and CTF, the zebrafish *gpr126* mutant lines *gpr126^{stl47}*[43] and *gpr126^{st49}*[15], which have been previously characterized for peripheral nervous system defects and semicircular canal morphogenesis defects, were analyzed. The *stl47* allele carries a 5 + 3 insertion deletion (indel) at Asparagine 68, which produces a frameshift-induced stop codon in the CUB domain, predicting a truncated Gpr126 protein of 96 amino acids (Fig. S1a, b)[43]. The *st49* allele carries a point mutation that converts a tyrosine-encoding codon into a stop codon near the GPS motif; as such, *gpr126^{st49}* mutant alleles could encode a truncated Gpr126 protein of 782 amino acids corresponding to the NTF without the GPS motif (NTF$^{\Delta GPS}$) (Fig. S1a, b)[15]. Both mutations result in a puffy ear phenotype caused by abnormal semicircular canal morphogenesis, confirming previously published data (Fig. S1c)[43]. qPCR analysis revealed no differences in *gpr126* mRNA levels in hearts obtained from 96 h post fertilization (hpf) wild type (WT), *gpr126^{stl47}*, and *gpr126^{st49}* embryos (Fig. S1d). These data suggest that these mutant *gpr126* mRNAs are not subjected to nonsense-mediated RNA decay, confirming previous data for *gpr126^{st49}*[15].

Thus far, it has not been tested whether the predicted truncated proteins are produced from the *gpr126^{stl47}* and *gpr126^{st49}* alleles. As no working anti-Gpr126 antibodies are currently available, we ectopically expressed zebrafish full-length *gpr126* cDNA as well as *gpr126* cDNA containing the *stl47* or *st49* mutation in the human retinal pigment epithelial cell line ARPE-19, which is routinely used for protein expression studies. The encoded proteins have an N-terminal HA-tag and a C-terminal GFP-tag (Fig. 1a). Immunofluorescence and Western blot (WB) analyses showed that WT and mutant forms of Gpr126 were expressed (Figs. 1 and S2). Notably, for the mutant forms, although the N-terminal HA signal was present, no C-terminal GFP signal was detected (Fig.1b, c), indicating that the mutations resulted in truncated forms of Gpr126. These data were supported by WB analyses. Anti-GFP antibodies detected a band of ~70 kDa in the WT-transfected cells, which corresponds to Gpr126-CTF-GFP (69 kDa) (Fig. 1d). No bands were detected in cells expressing the mutant forms. Anti-HA antibodies detected two bands of ~90 and ~100 kDa in the WT-transfected cells, which correspond to HA-NTF$^{\Delta GPS}$ (91 kDa) (Fig. 1d). Similarly, a band of ~90 kDa was detected in cells expressing the *st49*-mutant form, which is predicted to encode the NTF$^{\Delta GPS}$. In cells expressing the *stl47*-mutant form, a band of ~10 kDa was detected, the size of which corresponds to the predicted truncated Gpr126 protein containing the signal peptide and the partial CUB domain (11 kDa).

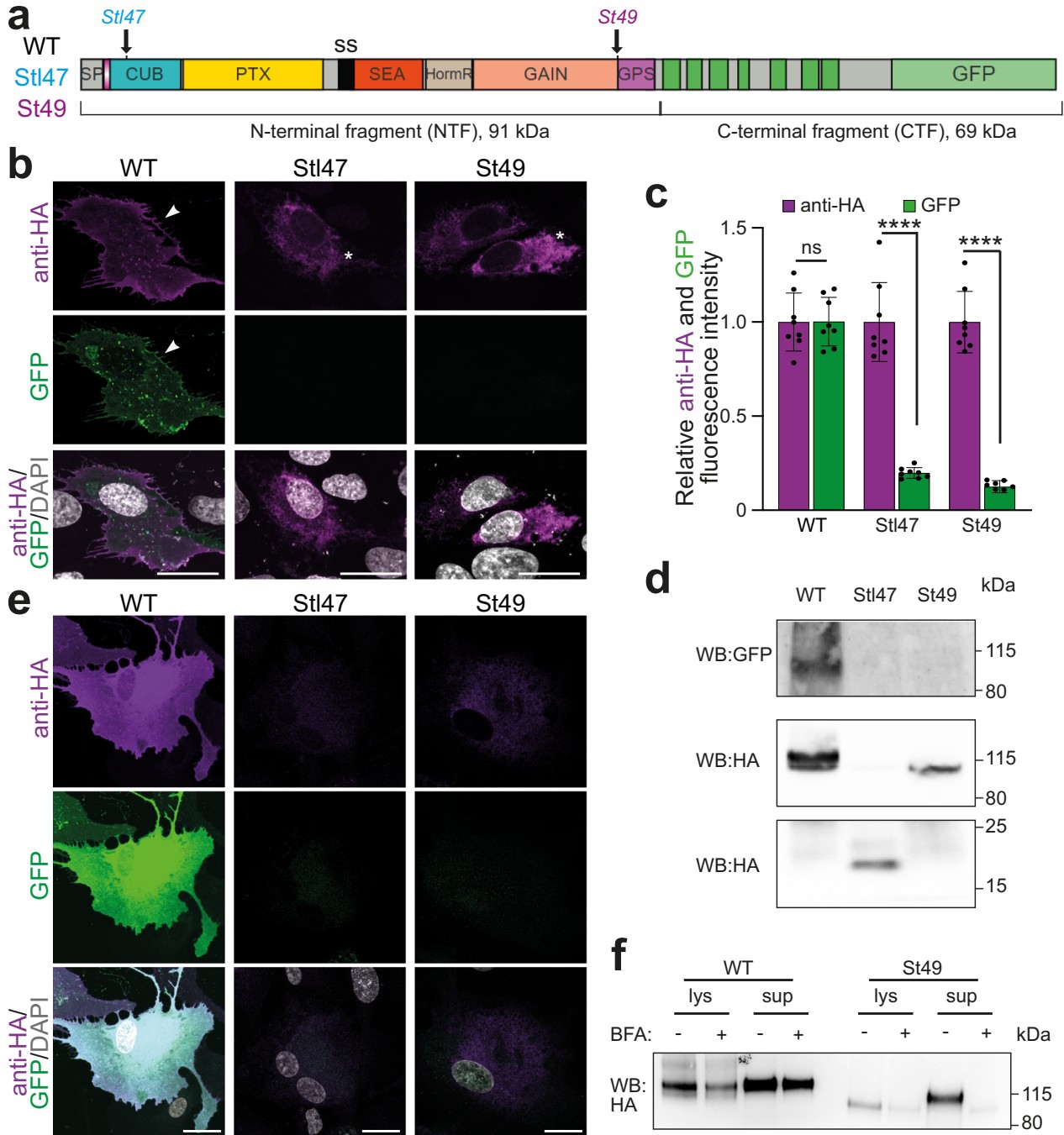

**Fig. 1 | *gpr126* mutants can produce truncated Gpr126 proteins. a** Schematic representation of Gpr126 protein tagged with HA at its N-terminus and GFP at its C-terminus indicating *stl47* and *st49* mutation sites and known splice sites (S). The illustration was generated with Illustrator for Biological Sequences 2.0. **b** Representative images of ARPE-19 cells transfected with HA-gpr126-GFP (WT), HA-gpr126^stl47^-GFP (Stl47), and HA-gpr126^st49^-GFP (St49) stained for HA (purple). Nuclei were visualized with DAPI (gray). Arrowheads: membrane localization of HA and GFP. Asterisks: intracellular localization of HA. Scale bars: 20 μm. **c** Quantification of (**b**). Fluorescence intensity was normalized to intensities of HA-gpr126-GFP (WT)-transfected cells. $n = 8$ cells/ each construct. Each point represents a cell. One way ANOVA revealed that the WT was significantly different from the mutants ($p < 0.0001$). The *p*-values for individual groups were calculated using

two-tailed unpaired t-test, $p = 0.9735$ (WT), $p < 0.0001$ (Stl47); $p < 0.0001$ (St49). Data are shown as mean ± SD. ****: $p < 0.0001$; ns not statistically significant. **d** Representative western blot (WB) analysis of lysates from ARPE-19 cells transfected with HA-gpr126-GFP (WT), HA-gpr126^stl47^-GFP (Stl47), and HA-gpr126^st49^-GFP (St49) using antibodies against GFP and HA; as indicated. $n = 3$ independent experiments. **e** Representative images of ARPE-19 cells stably expressing HA-gpr126-GFP (WT), HA-gpr126^stl47^-GFP (Stl47), and HA-gpr126^st49^-GFP (St49) stained for HA (purple). Nuclei were visualized with DAPI (gray). Scale bars: 20 μm. **f** Representative WB analysis of lysates (lys) and supernatant (sup) from control and brefeldin A (BFA)-treated ARPE-19 cells stably transfected with HA-gpr126-GFP (WT), HA-gpr126^stl47^-GFP (Stl47), and HA-gpr126^st49^-GFP (St49) using antibodies against HA; as indicated. $n = 3$ independent experiments.

WT Gpr126 localized to the plasma membrane 24 h post-transfection (Fig. 1b). In contrast, the mutant Gpr126 forms failed to localize to the membrane (Fig. 1b), and in a minority of transfected cells, cytoplasmic expression was detected (Fig. S2a, b). Notably,

previously we have shown that ectopic expression of NTF^ΔGPS^ in HeLa as well as H9c2 cells results in a stable, secreted protein[22]. Thus, the failure to localize to the membrane and the reduced number of HA-positive cells (Fig. S2a, b) could be explained by the loss of the 7TM

domain in both truncated forms. To confirm that a stable, secreted NTF$^{\Delta GPS}$ was produced by the *gpr126$^{st49}$* cDNA, the lysates and culture medium from ARPE-19 cells transfected with WT or St49 constructs were analyzed by WB (Fig. S2c). These data revealed that the culture medium contained NTF/NTF$^{\Delta GPS}$. Moreover, treating the cells with brefeldin A, which blocks the trans-Golgi secretory pathway[44,45], significantly decreased the amount of NTF/NTF$^{\Delta GPS}$ in the medium. Given that transient overexpression of proteins can lead to protein artifacts, we generated stable ARPE-19 lines expressing WT as well as St49 and Stl47 mutants of Gpr126 for a more reliable analysis (Fig. 1e). We observed membrane localization of WT Gpr126, while the mutant forms failed to localize to the membrane. In addition, there was no detectable cytoplasmic accumulation of the mutant forms, indicating that the truncated forms are not retained in the cytoplasm. WB analyses of the lysates and culture medium from the stable lines, with and without treatment with brefeldin A, confirmed that NTF$^{\Delta GPS}$ is secreted in the St49 overexpressing cells (Fig. 1f). These data suggest that *gpr126$^{st49}$* mutants express and secrete NTF$^{\Delta GPS}$, which previously has been reported to be sufficient for heart development[22].

Altogether, these data demonstrate that expressing zebrafish *gpr126* cDNA containing the *stl47* or *st49* mutation in ARPE-19 cells results in the production of the predicted truncated forms of Gpr126: the zebrafish mutant *gpr126$^{stl47}$* expresses the first 96 amino acids of Gpr126 containing a part of the CUB domain, whereas *gpr126$^{st49}$* expresses secreted NTF$^{\Delta GPS}$.

## Maternal zygotic *gpr126$^{stl47}$* mutants exhibit hypotrabeculation

Previous reports aiming to define the role of Gpr126 during heart development are conflicting[17,22,24]. To determine whether Gpr126 is required for heart development, the *gpr126$^{stl47}$* line was crossed with the reporter line *Tg(myl7:EGFP-hsa.HRAS)* in which GFP is localized at the cardiomyocyte plasma membrane. In vivo confocal imaging of the hearts of WT siblings showed clear luminal projections into the ventricle, forming a complex network of trabeculae at 96 and 120 hpf (Fig. 2a, c). The analysis of zygotic (Z) *gpr126$^{stl47}$* mutant hearts revealed, at 120 hpf, no obvious gross morphological defects or trabeculation defects as compared to WT siblings (Fig. 2a, b). These data suggested initially that Gpr126 is not required for heart development in zebrafish. However, the lack of a phenotype in Z *gpr126$^{stl47}$* mutant hearts might be explained by compensation through maternal deposition of WT mRNA or protein into the egg.

Notably, it has previously been reported that *gpr126$^{stl47}$* maternal zygotic (MZ) mutants exhibit a more severe phenotype during Schwann cell development[43]. It has been described that 96% of aGPCR have at least some levels of maternal expression[46]. The qPCR data for Gpr126 show detectable expression at 3 hpf, suggesting the presence of maternal mRNA[46]. To confirm the maternal deposition of *gpr126* RNA, we performed in situ hybridization in embryos prior to zygotic genome activation to visualize the maternal transcript and observed that WT as well as the MZ *gpr126$^{stl47}$* and MZ *gpr126$^{st49}$* mutants display *gpr126* expression at 2 hpf (Fig. S3a).

Unlike Z mutants, MZ mutants have no maternal deposition of WT mRNA or protein into the egg and exhibited more pronounced defects in the expression of myelin basic protein than Z mutants[43]. We observed trabeculation defects in MZ *gpr126$^{stl47}$* mutants at 96 and 120 hpf (Fig. 2c): at 120 hpf, we observed in 16.7% (5 out of 30) of mutants that trabeculation was absent and the ventricular wall was single layered; in 26.7% (8 out of 30) of mutants, hypotrabeculation was detected; and in 20% (6 out of 30) of mutants abnormal multi-layered ventricular walls were observed (Fig. 2d). Thus, in total 63.4% (19 out of 30) of the MZ *gpr126$^{stl47}$* exhibited trabeculation defects, whereas 36.6% exhibited normal trabeculation. Note, a pericardial oedema was sporadically observed in MZ *gpr126$^{stl47}$* mutants, as previously reported[43]. Embryos with strong pericardial oedema were excluded from this study. We did not

observe any other gross morphological defects in the MZ *gpr126$^{st47}$* mutant larvae or developmental delays compared with WT larvae (Fig. S3b). For further characterization, we analyzed MZ animals and their WT control by IMARIS-based 3D surface reconstruction of the ventricular chambers at 120 hpf (Fig. 2e). In WT larvae, distinct muscular ridges in the ventricular lumen could be visualized. In contrast, MZ *gpr126$^{st47}$* mutants exhibited a smooth ventricular wall with no trabecular projections.

These data indicate that maternal contribution of Gpr126 rescues the zygotic loss in around two-thirds of *gpr126$^{st47}$* mutants, indicating incomplete penetrance, and that Gpr126 is required for cardiac chamber development in zebrafish.

## Zygotic and maternal zygotic *gpr126$^{st49}$* mutants exhibit inner cardiac wall multilayering

Morpholino-mediated knockdown experiments, including rescue experiments, previously suggested that the NTF$^{\Delta GPS}$ is sufficient for trabeculation and heart development[22]. However, in vivo confocal imaging of zygotic *gpr126$^{st49}$* hearts revealed that a majority of mutant hearts exhibited a multilayered ventricular wall (60%, Fig. 2a, b). The analysis of MZ *gpr126$^{st49}$* mutants recapitulated the multilayered phenotype (66.67%, 20 out of 30) observed in the Z *gpr126$^{st49}$* mutants (Fig. 2c, d), with a similar phenotypic penetrance and expressivity. In addition, 6.67% (2 out of 30) of the mutants exhibited hypotrabeculation. IMARIS-based 3D surface reconstruction revealed a thick compact wall with few or no trabeculae in MZ *gpr126$^{st49}$* mutant hearts (Fig. 2e). In contrast, no obvious defect in the endocardium or the cardiac jelly between the endocardium and the myocardium was detected at 72 and 96 hpf, when analyzing MZ *gpr126$^{st47}$* and *gpr126$^{st49}$* mutants in the background of *Tg(myl7:mKATE-CAAX); Tg(krt4:GFP)*[47], which label the myocardium and the endocardium, respectively (Fig. S4). The only apparent difference is that the endocardium is not folded as normal, which appears to be due to the lack or a reduced number of trabeculae. These data suggested that either NTF$^{\Delta GPS}$ exhibits a dominant effect or that the CTF is required in conjunction with the NTF for proper heart development. To exclude the possibility that the multilayering phenotype of *gpr126$^{st49}$* hearts is due to a dominant effect of NTF$^{\Delta GPS}$, we analyzed the Z and MZ heterozygotes of the mutants and observed that the heterozygotes of both mutants do not exhibit any obvious trabeculation defects (Fig. S5), indicating that NTF$^{\Delta GPS}$ exhibits no dominant effect.

As our new data appeared to be in contrast to our previous findings, we revisited our previously published data regarding overexpressing NTF$^{\Delta GPS}$ in *gpr126* morphants[22]. While previously the focus was only on the "reappearance of trabeculae", a closer look revealed that the data could also be interpreted as hypertrabeculation or multilayering (note, the analysis was limited to 80 hpf due to the morpholino-based approach). Therefore, we analyzed myocardial-specific (*myl7*-driven) stable expression of NTF$^{\Delta GPS}$ in WT and MZ *gpr126$^{st47}$* mutants as well as mosaic expression in WT larvae, as our previous data indicate that NTF acts as a secreted molecule on cardiomyocytes[22]. Notably, overexpression of NTF$^{\Delta GPS}$ in MZ *gpr126$^{st47}$* mutants led to a multilayering phenotype at 120 hpf, similar to MZ *gpr126$^{st49}$* mutants, while WT larvae overexpressing NTF$^{\Delta GPS}$ exhibited normal trabeculation (Fig. S6a, b). Mosaic overexpression of NTF$^{\Delta GPS}$ by injecting single-cell-stage *Tg(myl7:EGFP-hsa.HRAS)* embryos with a *myl7:gpr126NTF-p2a-tdTomato* plasmid revealed that mosaic NTF$^{\Delta GPS}$ overexpression in cardiomyocytes leads to local multilayering at 96 hpf in 62% of the injected embryos (5 out of 8) (Fig. S6c). The tdTomato-overexpressing cardiomyocytes adhere together along with tdTomato-negative cardiomyocytes, while the rest of the ventricular wall is monolayered.

These data suggest that *gpr126$^{st49}$* mutants express a functional NTF$^{\Delta GPS}$ and that the NTF is not sufficient for proper heart development, revising our previous conclusion[22].

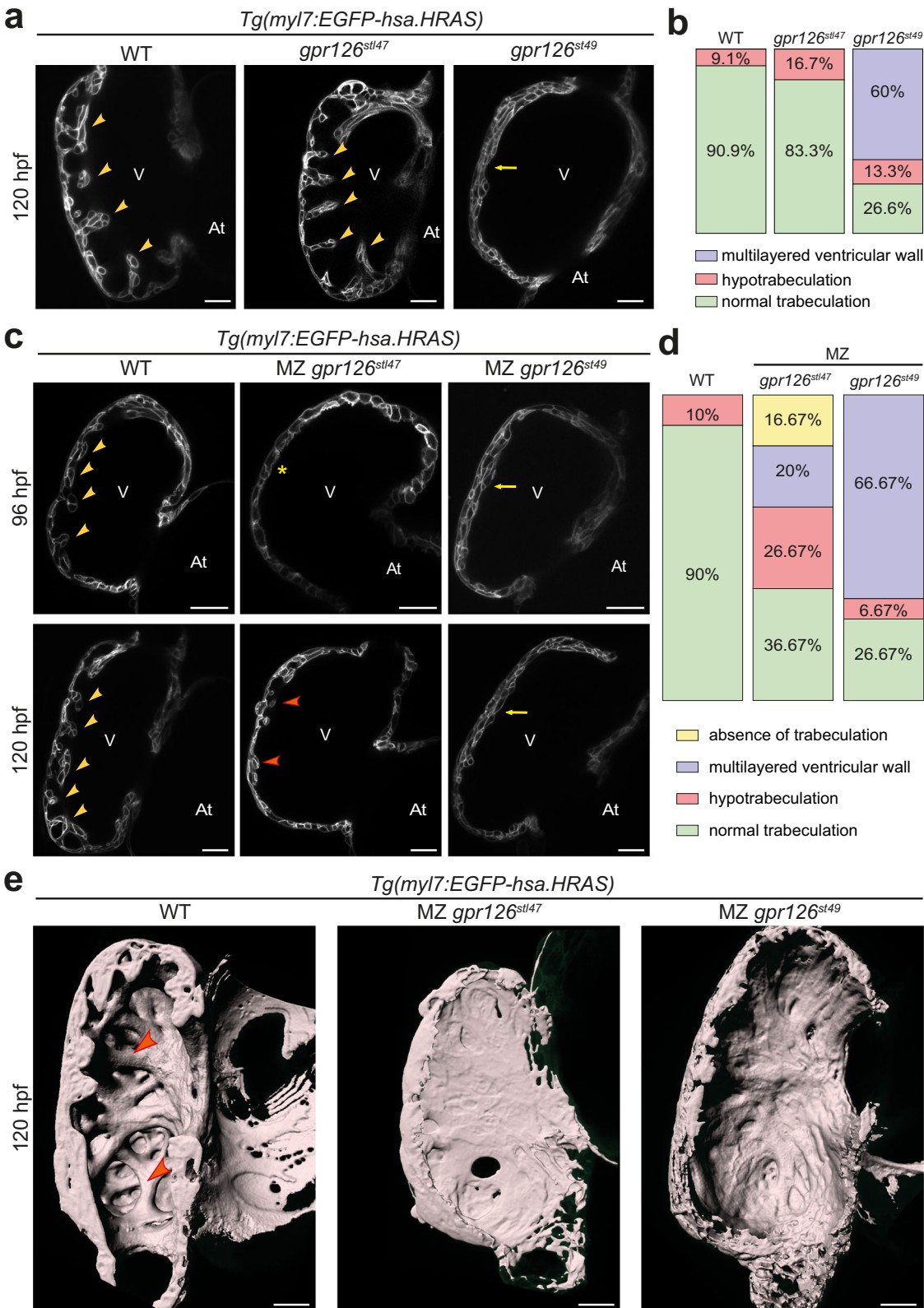

**Fig. 2 | Gpr126-NTF and -CTF are required for trabeculation. a** Representative confocal images (mid-sagittal sections) of hearts of WT, zygotic *gpr126^stl47^*, and zygotic *gpr126^st49^* larvae crossed into *Tg(myl7:EGFP-hsa.HRAS)* background at 120 h post fertilization (hpf). Arrowheads: trabeculae, Arrows: multilayered ventricular wall. V: ventricle. At: atrium. Scale bars: 20 μm. **b** Quantification of trabeculation phenotypes in 120 hpf WT (*n* = 11), zygotic *gpr126^stl47^* (*n* = 12), and zygotic *gpr126^st49^* (*n* = 15) larvae. **c** Representative confocal images (mid-sagittal sections) of hearts of WT, maternal zygotic (MZ) *gpr126^stl47^*, and MZ *gpr126^st49^* larvae crossed into

*Tg(myl7:EGFP-hsa.HRAS)* background at 96 and 120 hpf. Yellow arrowheads: trabeculae, orange arrowheads: shorter trabeculae Asterisks: absence of trabeculae. Arrows: multilayered ventricular wall. V: ventricle. At: atrium. Scale bars: 20 μm. **d** Quantification of trabeculation phenotypes in 120 hpf WT (*n* = 20), MZ *gpr126^stl47^* (*n* = 30), and MZ *gpr126^st49^* (*n* = 30) larvae. **e** 3D surface rendering of the myocardium of 120 hpf WT and mutants as indicated. Arrowheads: trabecular projections. Scale bars: 20 μm.

## Gpr126 mutants do not exhibit modified exon usage

Alternative splicing is a common feature of adhesion GPCRs[48] and the variable penetrance of trabeculation phenotypes in the *gpr126* mutants could be a result of compensatory mechanisms through splice isoforms. To exclude the possibility that alternatively spliced mRNAs result in a functional protein compensating for the truncated non-functional transcripts, we performed differential exon usage (DEU) analysis of 72 hpf whole larvae (hereafter referred to as "full body DEU analysis") as well as 96 hpf larval hearts (hereafter referred to as "heart-specific DEU analysis") for MZ *gpr126^st47^*, MZ *gpr126^st49^* and their WT littermates to determine the presence of alternatively spliced *gpr126* isoforms. The *gpr126^st47^* and *gpr126^st49^* mutants introduce a stop codon in Exon 3 and Exon 16, respectively (Fig. S7a)[15,43,49]. Full body DEU analysis between WT and mutants highlighted exon counting bin (B) 006 as differentially used (p-adjusted = 0.001; Fig. S7b). This exon counting bin corresponds to part of exon 4 of the canonical transcript isoform adgrg6-204 (Table S1). Notably, the exon usage plot suggested lower usage for MZ *gpr126^st49^*. However, RT-PCR analysis utilizing RNA isolated from 72 hpf WT and MZ *gpr126^st49^* larvae with primers spanning B004 to B007 did not confirm DEU analysis results, as no additional band (lacking B006) was detected (Fig. S7c). These RT-PCR data suggest there is no compensation through alternatively spliced isoforms in the MZ *gpr126^st49^* mutants.

Similarly, heart-specific DEU analysis did not show any significant difference in the use of exon counting bins between WT and mutants, further supporting that alternative splicing in the heart is not compensating for the predicted non-functional transcripts. However, different from what was observed in the full body DEU analysis, heart-specific DEU results showed lower use of B008, corresponding to exon 6 of the canonical transcript isoform, in all samples, i.e., both WT and mutants (Fig. S7d). RT-PCR using full-body and heart-specific RNA from WT samples confirmed DEU analysis results, showing that the predominant *gpr126* isoform in the heart lacks B008 (Fig. S7e), validating DEU analysis for alternative splicing assessment. These findings are consistent with the known canonical transcript structure of *gpr126*, as B002 (belonging to an alternative exon 1 in adgrg6-201, Table S1) could not be detected, and align with previously described splice isoforms that exclude Exon 24 of adgrg6-204, reflected in the lower usage of B028 (Table S1) in both full body and heart-specific samples. Notably, skipping of exon 6 results in deletion of 23 aa in zebrafish, affecting gpr126 protein conformation and signaling[50]. Consistently, the construct used in this study for rescuing NTF function lacks exon 6. Taken together, our results suggest that alternative splicing in the heart is not compensating for the predicted non-functional transcripts.

## Rescue experiments reveal that Gpr126-CTF is required for cardiac trabeculation

Our data indicate that the observed phenotype in *gpr126^st49^* mutants is due to the lack of the CTF and not due to a dominant effect of the expressed NTF^ΔGPS^. If this assumption is correct, then it should be possible to rescue the *gpr126^st49^* mutant phenotype by CTF overexpression. Therefore, we first assessed whether *gpr126* is, like in mice, expressed in endocardial cells of the zebrafish heart. We performed fluorescence-activated cell sorting (FACS) for endocardial cells and cardiomyocytes using isolated cells from extracted hearts of *Tg(myl7:EGFP-hsa.HRAS)*; *Tg(kdrl:nls-mCherry)* larvae at 48 and 96 hpf. Endocardial and myocardial enrichment was confirmed by the relative expression levels of *kdrl* and *myl7*, respectively. qPCR analysis showed that in zebrafish hearts, *gpr126* is enriched in endocardial cells compared to cardiomyocytes (Fig. 3a). This result was confirmed by RNAscope analysis (Fig. S8). Therefore, we generated a stable endothelial-specific transgenic line, *Tg(fli1a:gpr126CTF-p2a-tdTomato)* (Fig. 3b), that overexpresses a protein that is post-translationally cleaved into tdTomato and Gpr126-CTF, to perform rescue experiments in *gpr126^st49^* mutants. The CTF domain is tagged with HA at its N-terminus

after the signal peptide and FLAG-tagged at the C-terminus. In addition, a P2Y$_{12}$ N-terminus peptide was fused to the Gpr126 CTF to facilitate plasma membrane localization of the receptor (Fig. 3b)[9]. Anti-FLAG whole mount immunostaining of 72 hpf *Tg(fli1a:gpr126CTF-p2a-tdTomato)* embryos showed that the CTF is localized to the cell membrane in endocardial as well as vascular endothelial cells (Fig. 3c). *Tg(fli1a:gpr126CTF-p2a-tdTomato)* larvae did not show any gross morphological defects (Fig. S9a). The analysis of 96 hpf *Tg(fli1a:gpr126CTF-p2a-tdTomato)* larval hearts showed normal trabeculation and exhibited no obvious defects (Fig. S9b).

To determine whether CTF overexpression in the endocardium rescues the multilayered ventricular wall in MZ *gpr126^st49^* mutants, *Tg(fli1a:gpr126CTF-p2a-tdTomato)* were crossed with *gpr126^st49^* mutants in the background of *Tg(myl7:EGFP-hsa.HRAS)*. The analysis of 120 hpf MZ *gpr126^st49^*; *Tg(fli1a:gpr126CTF-p2a-tdTomato)*; *Tg(myl7:EGFP-hsa.HRAS)* hearts revealed that ectopic CTF expression, identified by tdTomato-positive endocardium, reversed the multilayering phenotype in MZ *gpr126^st49^* mutant hearts, resulting in the formation of trabecular projections into the ventricular lumen (Fig. 3d, e). MZ *gpr126^st49^* mutants overexpressing CTF exhibited clear muscular ridges in the ventricular chamber, while MZ *gpr126^st49^* mutants exhibited a thick, compact wall in the absence of mature trabecular ridges (Fig. 3f), as described above (Fig. 2e). These data show that endocardial expression of Gpr126-CTF rescues the multilayering phenotype observed in *gpr126^st49^* mutants, indicating that NTF^ΔGPS^ in *gpr126^st49^* mutants represents a fully functional NTF.

G proteins can signal through cAMP accumulation[25,51] and it has been shown that treatment with the adenylyl cyclase activator forskolin (25 μM) can rescue the myelination defects and puffy ear phenotype of *gpr126^st49^* mutants but not the radial sorting defects of *gpr126^st47^* mutants[43]. A phosphodiesterase inhibitor, 3-isobutyl-1-methyl-xanthine (IBMX, 100 μM), which raises cAMP levels independently of adenylyl cyclase activity, was used to rescue ear defects in other mutant *gpr126* alleles[13]. Therefore, we tested whether forskolin or IBMX treatment could also rescue the heart phenotype in *gpr126^st49^* mutants. We confirmed that these treatments rescued the puffy ear phenotype (Fig. S10a). Analysis of the hearts revealed that both treatments could partially rescue the multilayering phenotype in MZ *gpr126^st49^* mutants (Fig. S10b, upper panels). However, both treatments also blocked trabeculation, both in MZ *gpr126^st49^* mutant hearts and in WT hearts (Fig. S10b, lower panels). Thus, it remains unclear whether Gpr126 acts through cAMP activation to promote trabeculation.

Collectively, our data suggest that endocardial-specific Gpr126-CTF signaling is required for cardiac trabeculation in zebrafish.

## Gpr126-NTF is sufficient to maintain cell-cell adhesion and compact wall integrity

Compact wall integrity is required for proper initiation of trabeculation[52]. As N-cadherin is the major component of cardiac adherens junctions[39], we determined whether N-cadherin localization in compact wall cardiomyocytes prior to trabeculation is affected in the two mutants. *gpr126^st47^* and *gpr126^st49^* mutants were crossed in the background of *TgBAC(cdh2:cdh2-EGFP)* and analyzed at 72 hpf. Single-plane confocal images of WT hearts showed that N-cadherin specifically localizes laterally in the membrane of compact layer cardiomyocytes with some expansion to the basal side (Y-shape pattern) (Figs. 4, and S11a) as previously published[39]. Maximum intensity projection images confirmed the lateral localization of N-cadherin in WT hearts (Figs. 4, and S11a). In contrast, in MZ *gpr126^st47^* mutant hearts, N-cadherin was distributed randomly, still laterally at the junctions but also at the apical and/or basal membrane of the compact layer cardiomyocytes, which was also obvious in maximum intensity projections (Fig. 4). The maximum intensity projection images exhibit an organized distribution of N-cadherin in WT hearts; however, in the MZ *gpr126^st47^* mutant hearts, the N-cadherin distribution is haphazard and

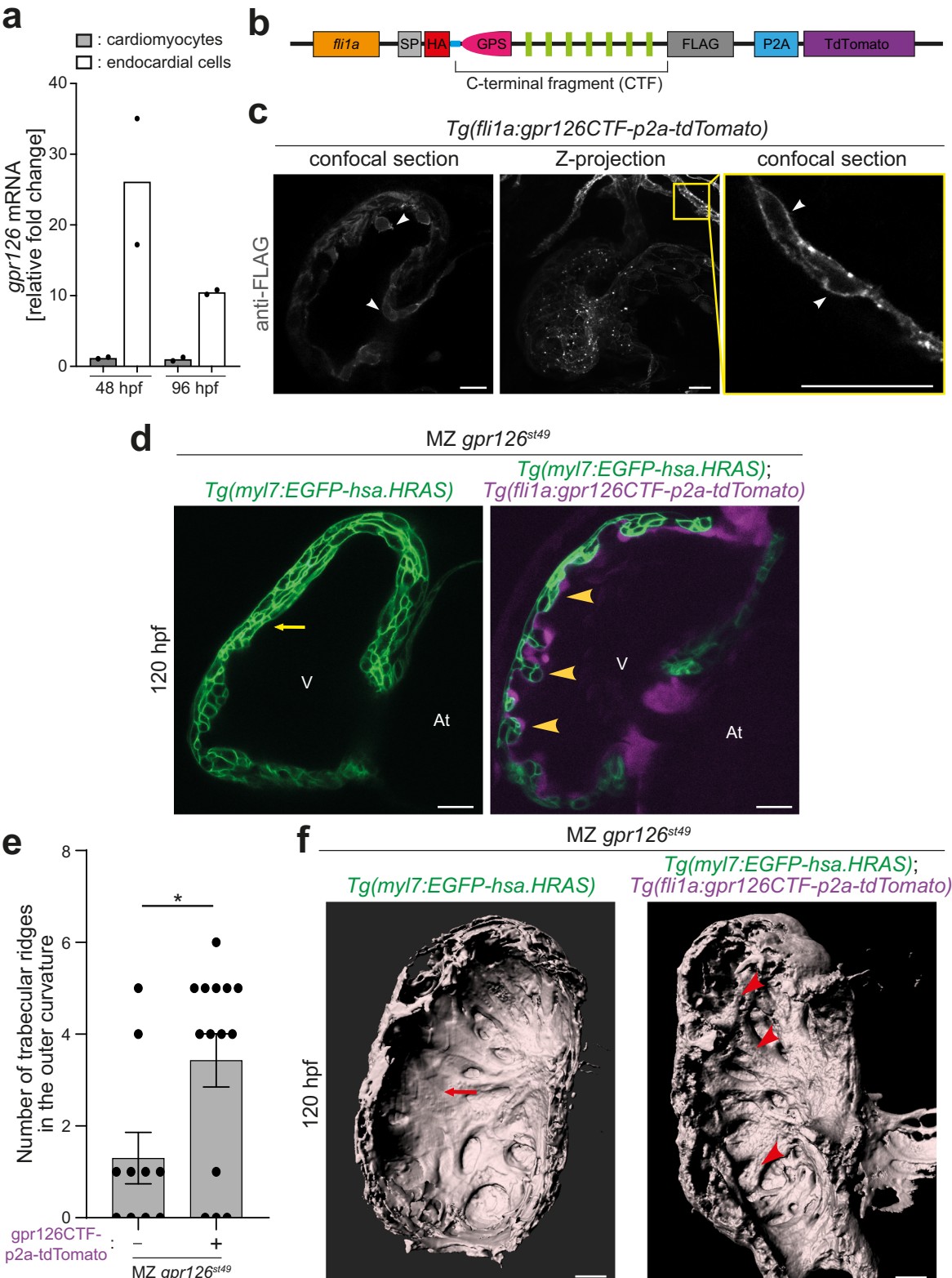

misorganized (Fig. S12). Notably, in the MZ *gpr126st49* mutants, which express NTF$^{\Delta GPS}$, N-cadherin was localized in a WT-like pattern at the junctions (laterally) of compact layer cardiomyocytes (Figs.4 and S11a). Overexpression of CTF alone in MZ *gpr126stl47* mutants did not rescue the compact wall integrity defects (Fig. S11b), further indicating the requirements of NTF to mediate cardiomyocyte adhesion. These data indicate that the NTF$^{\Delta GPS}$ is sufficient for the proper localization of

adherens junctions in cardiomyocytes of the compact wall prior to trabeculation.

## Gpr126-CTF is required to establish trabecular identity of cardiomyocytes

The compact layer cardiomyocytes exhibit apicobasal polarity, while trabecular layer cardiomyocytes are depolarized[41]. The observed

**Fig. 3 | Endocardial CTF overexpression reinstates trabeculation in MZ *gpr126^st49* mutants.** **a** qPCR data from FACS-sorted cardiomyocytes and endocardial cells from hearts at 48 and 96 h post fertilization (hpf). Relative mRNA levels of *gpr126* are enhanced in endocardial cells compared with cardiomyocytes. *n* = 2 independent experiments. **b** Schematic of the construct used in *Tg(fli1a:gpr126CTF-p2a-tdTomato)*. The p2a peptide allows for cleavage right after protein translation, separating Gpr126-CTF and tdTomato fluorescent protein. The P2Y12 N-terminus peptide (blue line between HA tag and GPS) is fused to the HA tag to facilitate membrane localization of Gpr126-CTF. **c** Single plane confocal image and Z-projection of anti-FLAG immunostaining of 72 hpf *Tg(fli1a:gpr126CTF-p2a-tdTomato)* embryos. Arrows: membrane localization. *n* = 4 independent experiments. **d** Representative confocal images (mid-sagittal sections) of *Tg(my7:EGFP-has.H-RAS)*; maternal zygotic (MZ) *gpr126^st49* and *Tg(my7:EGFP-has.HRAS)*;

*Tg(fli1a:gpr126CTF-p2a-tdTomato)*; MZ *gpr126^st49* hearts at 120 hpf. Yellow arrows: multi-layered ventricular wall; yellow arrowheads: trabeculae; V: ventricle; At: atrium. Scale bars: 20 μm. *n* = 3 independent experiments. **e** Quantification of (d). Number of trabecular ridges in MZ *gpr126^st49* hearts in the absence (*n* = 10) or presence (*n* = 14) of *gpr126CTF-P2A-TdTomato*. *n* represents the number of hearts for which the trabecular ridges in the outer curvature were quantified. The dots represent the individual hearts. *p* = 0.0183 (two-tailed unpaired Student's *t* test). Data are mean ± S.E.M. *: *p* < 0.05. **f** 3D surface rendering of the myocardium of 120 hpf *Tg(my7:EGFP-hsa.HRAS)*; MZ *gpr126^st49* and *Tg(my7:EGFP-hsa.HRAS)*; *Tg(fli1a:gpr126CTF-p2a-tdTomato)*; MZ *gpr126^st49* hearts. Red arrowheads: trabecular projections; red arrows: thick compact wall with absence of mature trabecular ridges. Scale bars: 20 μm.

multilayering phenotype of MZ *gpr126^st49* mutants raised the question whether the inner layer of cardiomyocytes consists of depolarized cardiomyocytes, indicating hypertrabeculation, or whether they consist of polarized cardiomyocytes caused by abnormal multilayering of the compact ventricular wall. To determine the apicobasal polarity of cardiomyocytes, MZ *gpr126^st49* mutants were crossed into the background of *Tg(myl7:EGFP-Podxl)*; *Tg(myl7:mKATE-CAAX)*, in which the cardiomyocyte membrane is labeled with mKATE and the apical membrane of polarized cardiomyocytes is marked by the apical protein podocalyxin fused with EGFP. In 96 hpf WT siblings, EGFP-podocalyxin is localized at the apical/abluminal side of compact layer cardiomyocytes and around the entire membrane of delaminated, trabecular cardiomyocytes, indicating their polarized and depolarized states, respectively (Fig. 5a). In contrast, in multilayered hearts of MZ *gpr126^st49* animals, cardiomyocytes of both the outer layer as well as the inner layers were mainly polarized (Fig. 5a). Quantification revealed that the percentage of depolarized cardiomyocytes in the inner layers of the ventricles was significantly reduced from 94.7% ± 1.7% in WT to 16.6% ± 2.6% in MZ *gpr126^st49* mutants (*p* < 0.001; WT: *n* = 6 hearts; MZ *gpr126^st49*: *n* = 12 hearts; per heart 15-25 cardiomyocytes were counted) (Fig. 5b). These data indicate that the cardiomyocytes in multilayered hearts of MZ *gpr126^st49* mutants fail to depolarize and consequently cannot seed the trabecular layer.

During depolarization, compact layer cardiomyocytes remodel their adherens junctions, which involves the relocalization of N-cadherin from the lateral to the basal membrane. To test whether adherens junction remodeling is affected in MZ *gpr126^st49* mutant hearts, N-cadherin localization was analyzed at 96 hpf, when multilayering was observed. In WT hearts, the compact layer cardiomyocytes showed basolateral distribution of N-cadherin, and trabecular layer cardiomyocytes had a punctate distribution (Fig. 5c), as previously described[39]. In contrast, in the multilayered MZ *gpr126^st49* mutant hearts, N-cadherin was localized at the lateral side of the compact layer and inner-layered cardiomyocyte wall (Figs. 5d and S11c). These data suggest that cardiomyocytes in the multilayered wall of *gpr126^st49* mutants fail to attain trabecular identity due to defects in remodeling adherens junctions and cardiomyocyte polarity defects.

It has been demonstrated that Notch signaling is activated during trabeculation upon cardiomyocyte delamination in a subset of compact wall cardiomyocytes to inhibit their delamination[40–42]. As MZ *gpr126^st47* mutants are characterized by hypotrabeculation, no change in myocardial Notch activity is expected. In contrast, we hypothesized that failure of *gpr126^st49* mutant cardiomyocytes to attain trabecular identity might be due to enhanced myocardial Notch activity. To test this hypothesis, *gpr126^st47* and *gpr126^st49* mutants were crossed in the background of *Tg(TP1:Venus-PEST)*, in which the Notch-responsive TP1 promoter drives expression of Venus-PEST, a fluorescent protein with a short half-life (~2 h). During trabeculation in WT larvae, Notch signaling can be detected in the atrioventricular canal (AVC), the outflow tract and a few cardiomyocytes in the compact layer of the myocardium (Fig. 6a). Analysis of the mutant hearts at 96 and 120 hpf

indicated that Notch activity was not altered in MZ *gpr126^st47* mutants (Fig. 6a). In contrast, Notch activity in the heart was markedly increased in MZ *gpr126^st49* mutants. Quantitative analysis of the Venus fluorescence intensity normalized to the signal in AVC valves confirmed that myocardial Notch activity was significantly increased in MZ *gpr126^st49* mutants (Fig. 6b). Notably, NTF^ΔGPS overexpression in MZ *gpr126^st47* mutants, which led to myocardial multilayering mimicking MZ *gpr126^st49* (Fig. 4b,c), also resulted in a substantial increase of Notch activity at 120 hpf (Fig. 6c, d). On the other hand, overexpression of CTF in the MZ *gpr126^st49* mutants, which restored normal trabeculation, also normalized myocardial Notch activity (Fig. 6e, f). In contrast, overexpression of NTF^ΔGPS and CTF in the WT background did not lead to any substantial changes in myocardial Notch activity (Fig. S13a, b). Additionally, overexpression of CTF in MZ *gpr126^st47* mutants failed to alter Notch activity (Fig. S13c, d).

Collectively, these data suggest that CTF signaling is required to inhibit myocardial Notch activity to allow cardiomyocytes to attain trabecular identity.

## Discussion

We conclude that Gpr126 is required for heart development. First, MZ *gpr126^st47* mutants, which encode a severely truncated Gpr126 protein, exhibit hypotrabeculation. Second, *gpr126^st49* hearts, with a truncated CTF but an intact NTF^ΔGPS, are mainly multilayered, characterized by defects in adherens junction remodeling and lack of depolarization. Third, endocardial Gpr126-CTF expression reinstates trabeculation in *gpr126^st49* mutants. Fourth, cell-cell adhesion and compact wall integrity are maintained in MZ *gpr126^st49* mutant hearts. Finally, myocardial Notch signal was markedly increased in MZ *gpr126^st49* mutant hearts, which was restored by CTF overexpression. Thus, our data indicate that both NTF and CTF are involved in distinct cellular mechanisms of ventricular chamber development. The NTF is required to maintain cell-cell adhesion in the compact layer to maintain compact wall architecture prior to the onset of trabeculation. The CTF is required to remodel cardiomyocyte adhesion, whereas NTF and CTF together modulate myocardial Notch activity. Notably, our conclusion that NTF and CTF are required at different stages during development is in agreement with observations during myelination, where NTF is required for radial sorting, which is a prerequisite for axon wrapping, which is dependent on CTF signaling[43].

The analysis of the *gpr126* mutant zebrafish lines revealed that zygotic *gpr126^st47* mutants exhibit no trabeculation phenotype. This suggested that maternal *gpr126* mRNA can compensate for the loss of zygotic gpr126. Consequently, MZ *gpr126^st47* mutants exhibit defective trabeculation. However, this raised the question of how a maternal mRNA can compensate for such an extended time period for the lack of zygotic mRNA. Notably, it had already previously been shown that MZ *gpr126^st47* mutants exhibit a more prominent phenotype regarding Schwann cell development. Importantly, at 96 hpf 42% of the Z *gpr126^st47* mutants showed normal myelin basic protein expression, while none of MZ *gpr126^st47* mutants expressed myelin basic protein[43].

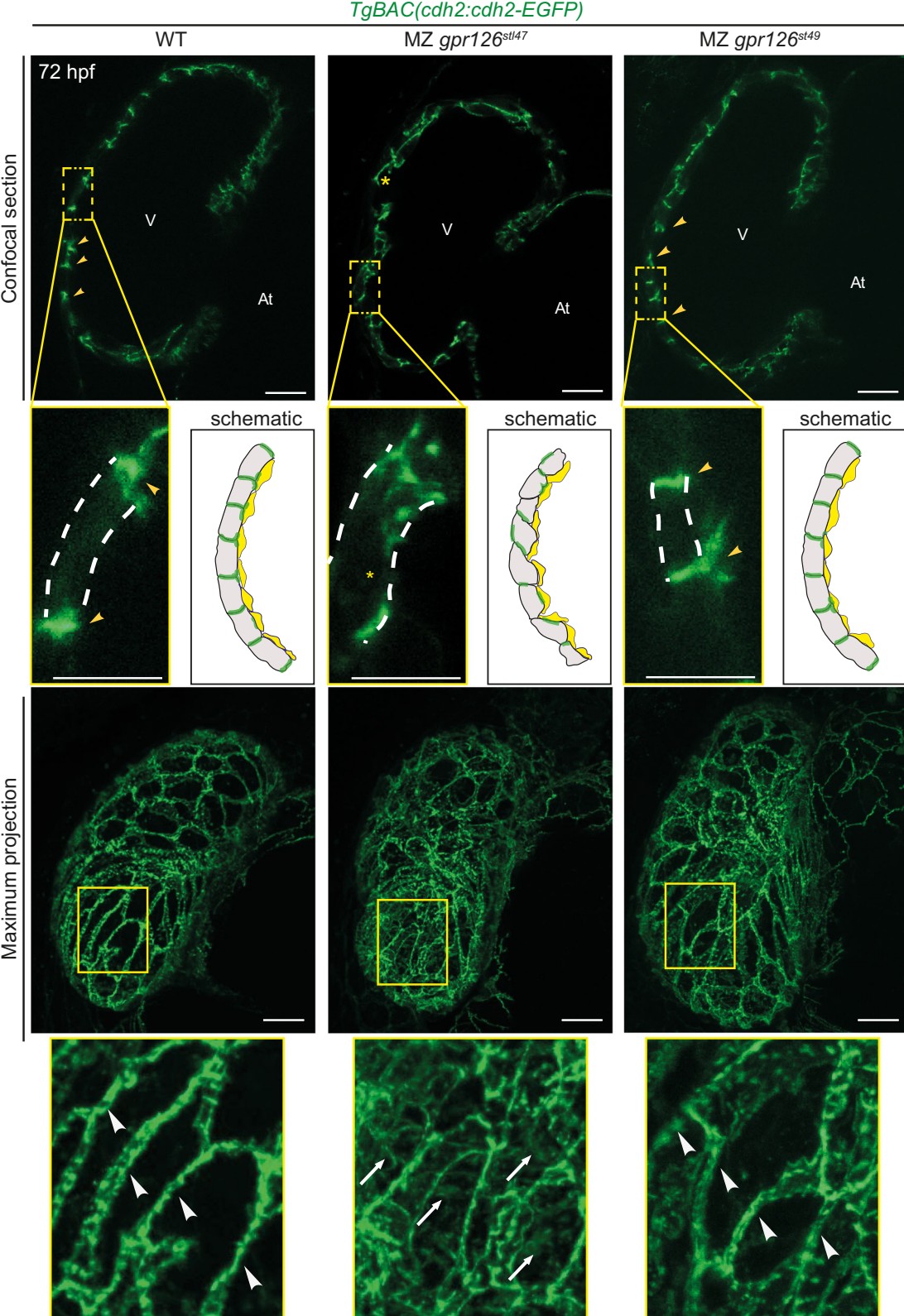

**Fig. 4 | Gpr126-NTF regulates compact wall integrity at the onset of trabeculation.** Representative confocal images (mid-sagittal sections and maximum intensity projections) of *TgBAC(cdh2:cdh2-GFP)*; WT (10/10) as well as maternal zygotic (MZ) *gpr126stl47* (10/12) and MZ *gpr126st49* (12/12) hearts at 72 h post fertilization (hpf). yellow arrowheads: lateral localization of N-cadherin; yellow asterisks: mislocalization of N-cadherin; white dotted line: outline of individual cardiomyocytes; white arrowheads: N-cadherin localization at the lateral junctions in the maximum projection; white arrows: mislocalized N-cadherin in the maximum projection; V: ventricle; At: atrium. Scale bars: 20 μm, 10 μm (magnified images). Schematic represents N-cadherin localization in the compact layer of WT, MZ *gpr126stl47* and MZ *gpr126st49* hearts at 72 hpf. Endocardium is presented in yellow. *n* = 4 independent experiments.

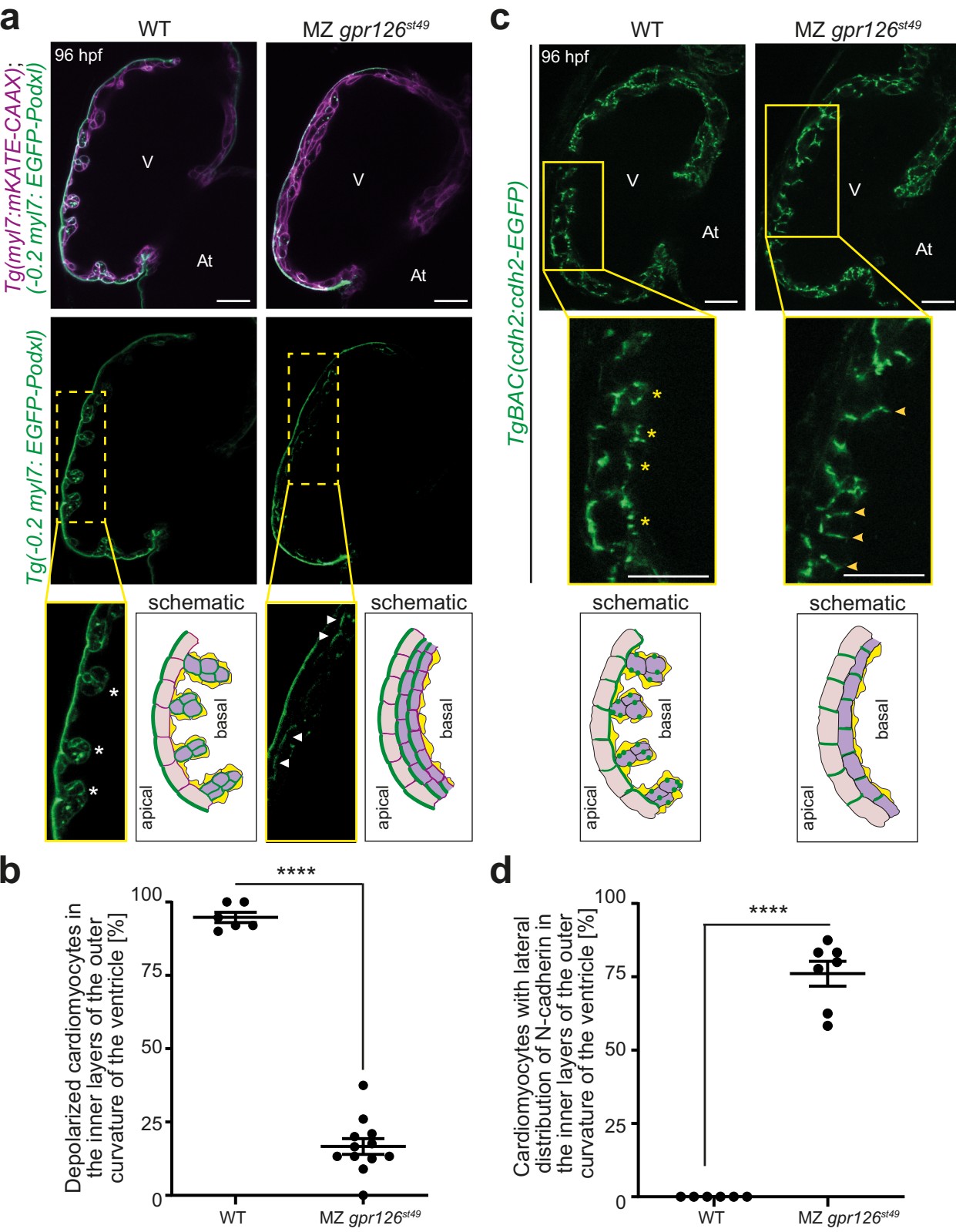

Collectively, our data indicate that the maternal *gpr126* mRNA and/or the encoded protein stability is high enough to allow normal trabeculation. However, this raises the question, why maternal *gpr126* mRNA/protein is not sufficient for normal trabeculation in *gpr126*[st49] mutants. The only difference is the presence of additional NTF[ΔGPS]. Thus, one possible explanation is that NTF[ΔGPS] acts in a dominant manner. However, our data argue against a dominant effect as for

example heterozygous *gpr126*[st49] mutants exhibit no phenotypes. Another possible explanation is that a balance of NTF- and CTF-mediated signaling is required for proper trabeculation. This hypothesis is supported by the observation of a substantial increase of Notch activity in *MZ gpr126*[st49] mutants, which could be normalized by the overexpression of CTF. Overall, all lines with an increased expression of NTF compared to CTF exhibit the multilayering phenotype, with the

**Fig. 5 | Gpr126-CTF aids in providing trabecular identity to cardiomyocytes.**
**a** Representative confocal images (mid-sagittal sections) of 96 h post fertilization (hpf) *Tg(myl7:EGFP-Podxl)*; *Tg(myl7:mKATE-CAAX)*; WT and maternal zygotic (MZ) *gpr126$^{st49}$* hearts. Asterisks: depolarized and delaminated cardiomyocytes; arrowheads: polarized cardiomyocytes; V: ventricle; At: atrium. Scale bars: 20 μm. Schematic represents the depolarized cardiomyocytes in the trabecular layer of WT hearts and polarized cardiomyocytes in the inner layers of *gpr126$^{st49}$* MZ hearts at 96 hpf. Endocardium is presented in yellow. **b** Percentage of depolarized cardiomyocytes in the inner layers of the outer curvature of the ventricles of WT (*n* = 6) and MZ *gpr126$^{st49}$* (*n* = 12) larvae. Each point represents a heart. *p* < 0.0001 (two-

tailed unpaired Student's *t* test). Data are mean ± S.E.M. ****: *p* < 0.0001.
**c** Representative confocal images (mid-sagittal sections) of 96 hpf *TgBAC(cdh2:cdh2-EGFP)*; WT (10/10) and MZ *gpr126$^{st49}$* (11/14) hearts. Asterisks: punctate distribution of N-cadherin around the cardiomyocyte; arrowheads: lateral distribution of N-cadherin; V: ventricle; At: atrium; Scale bars: 20 μm. Schematic represents N-cadherin localization in WT and MZ *gpr126$^{st49}$* hearts at 96 hpf. **d** Percentage of cardiomyocytes with lateral localization of N-cadherin in the inner layers of the outer curvature of the ventricles of WT (*n* = 6) and MZ *gpr126$^{st49}$* (*n* = 7) larvae. Each point represents a heart. *p* < 0.0001 (two-tailed unpaired Student's t-test). Data are mean ± S.E.M. ****: *p* < 0.0001.

exception of the NTF$^{\Delta GPS}$ overexpression line (Fig. S13a). Notably, in this line, cells express zygotic and maternal CTF and NTF and thus the ectopic expression of NTF might not be high enough to significantly impair the balance between NTF and CTF, or enough CTF is present to prevent increased Notch activity. In contrast, mosaic expression, usually mediating a markedly higher expression, was sufficient to induce local multilayering. In addition, the mosaic expression might result in adhesion heterogeneity between cardiomyocytes. Overall, our data suggest that the CTF is required to balance NTF-mediated signaling, preventing overactivation of myocardial Notch signaling.

Previously, several studies have reported that loss of Gpr126 results in hypotrabeculation[17,22,24]. However, no cellular or molecular mechanisms underlying this phenotype were identified. A recent study suggested that the observed heart defects in Gpr126 KO mice are secondary to placental defects[17]. Yet, in Gpr126 KO mouse models exhibiting heart defects[16,17,22,24], placenta defects were not always observed[16,22,24]. Furthermore, zebrafish do not have a placenta, which was provided as an explanation for the lack of trabeculation defects in *gpr126$^{bns341}$* (predicted early truncation) and *gpr126$^{bns342}$* (promoter-less allele) mutants[17]. The absence of a trabeculation phenotype in those mutants is in agreement with our data showing that zygotic *gpr126$^{stl47}$* mutant hearts exhibit no obvious trabeculation defect. Congruent with a more severe phenotype during Schwann cell development in MZ *gpr126$^{stl47}$* mutants[43], we observed hypotrabeculation phenotypes in MZ *gpr126$^{stl47}$* mutants. This suggests that the maternal deposition of mRNA may be sufficient to allow for normal heart development in zygotic mutants, including in *gpr126$^{bns341}$* and *gpr126$^{bns342}$* mutants. Furthermore, transcriptional adaptation has recently been described in zebrafish, where mutations that cause mutant mRNA degradation can trigger the upregulation of potential compensatory genes[53,54]. It will be interesting to test whether there is a heart phenotype in MZ *gpr126$^{bns341}$* and MZ *gpr126$^{bns342}$* mutants.

Previously reported rescue experiments in *gpr126* morphants suggested that NTF$^{\Delta GPS}$ is sufficient for heart development[22]. However, morpholinos exhibit short-lived effects and the latest analyzed time point was at 80 hpf. Here, we show that *gpr126$^{st49}$* mutants, which express NTF$^{\Delta GPS}$, exhibit a multilayered ventricular wall at 96 hpf. This suggests an important role of the CTF in trabeculation, which is further substantiated by endocardial-specific overexpression of CTF rescuing the multilayering phenotype in *gpr126$^{st49}$* mutant hearts. The data presented in this work correct our previous hypothesis that only NTF is sufficient for trabeculation.

The N-terminal regions of aGPCRs consist of various adhesion domains, which are capable of mediating cell-cell and cell-matrix interactions. While various studies have confirmed a role of aGPCRs in maintaining cell-cell adhesion in vitro[55,56], there is limited knowledge in vivo. Cardiomyocytes are linked to each other through intercalated discs (ICD), which allow the heart to function as a syncytium, with adherens junctions being a major ICD component[57]. The adherens junction defects observed in the cardiomyocytes of MZ *gpr126$^{stl47}$* mutants prior to the emergence of trabeculae reveal an early role of Gpr126 in maintaining cell-cell adhesion in the compact wall. As MZ *gpr126$^{st49}$* mutants do not exhibit these defects, the NTF appears to have an independent role in the proper localization of adherens

junctions in the compact layer, supporting our previous conclusion that NTF acts as a secreted ligand[22].

Further analysis of CTF-deleted MZ *gpr126$^{st49}$* mutants revealed the failure of cardiomyocytes to depolarize and modulate their adherens junctions. These results suggest that CTF-deleted cardiomyocytes fail to attain trabecular identity and cannot seed the trabecular layer, leading to the formation of a multilayered compact ventricular wall. The dual role of NTF and CTF in ventricular wall morphogenesis is similar to a previous study reporting a dual role for the hippo pathway effector Wwtr1 in maintaining compact wall architecture as well as modulating trabeculation[52].

It is likely that CTF signals in the endocardium through canonical G protein-coupled signaling, as observed in myelination during peripheral nervous system development[9,43]. However, as the treatment with cAMP-elevating drugs also affected trabeculation in WT hearts, it could not be determined whether CTF acts through canonical G-protein signaling. The effect on WT hearts is in agreement with recent reports that adrenergic signaling strengthens cardiomyocyte cohesion[58], altered cAMP levels are associated with impaired trabeculation[59], and that compartmentalized cAMP signaling is essential for proper heart development[60].

It has recently been shown that Gpr126 is a downstream target of endocardial Notch signaling and is a Notch effector throughout chamber development[31]. However, a role for *gpr126* regarding myocardial Notch is unknown. In recent years, myocardial Notch-mediated fate specification has emerged as a defining feature of trabecular morphogenesis[40,42] However, how myocardial Notch activity is regulated remains elusive. In zebrafish, the delaminating cardiomyocytes induce Notch activity in the adjoining compact wall cardiomyocytes which actively suppresses their delamination and prevents them from entering into the trabecular layer[42]. The enhanced myocardial Notch signaling in CTF-deleted MZ *gpr126$^{st49}$* mutants provides an explanation for why the CTF-deleted cardiomyocytes fail to attain trabecular identity and stay in the compact layer. The MZ *gpr126$^{stl47}$* mutants, however, do not form a proper compact wall due to the absence of NTF and hence do not reach the developmental stage where myocardial Notch activity could be enhanced. These results suggest a negative regulation of myocardial Notch activity by Gpr126 and link Gpr126 with Notch signaling-driven lateral inhibition during trabecular emergence. However, open questions regarding the interplay of Gpr126 and Notch receptors remain to be addressed. It has been shown recently that myocardial Notch signaling triggers a morphological transition in the myocardial surface area to facilitate ventricular development, through actinomyosin remodeling[61]. It would be interesting to explore in the future if the Gpr126-Notch axis has a role to play in this transition.

While our data demonstrate that endocardially expressed Gpr126 is required for regulating cardiomyocyte behavior, it remains unclear how. Endocardial-myocardial interactions are essential for ventricular chamber development, and several endocardial factors like Notch, Nrg2a, and Klf2 have previously been reported to affect myocardial patterning[31,36,37]. However, how endocardial factors affect cardiomyocyte behavior remains largely unknown and is currently under investigation. Recently, it has been suggested that endocardial protrusions regulate communication between endocardium and myocardium by

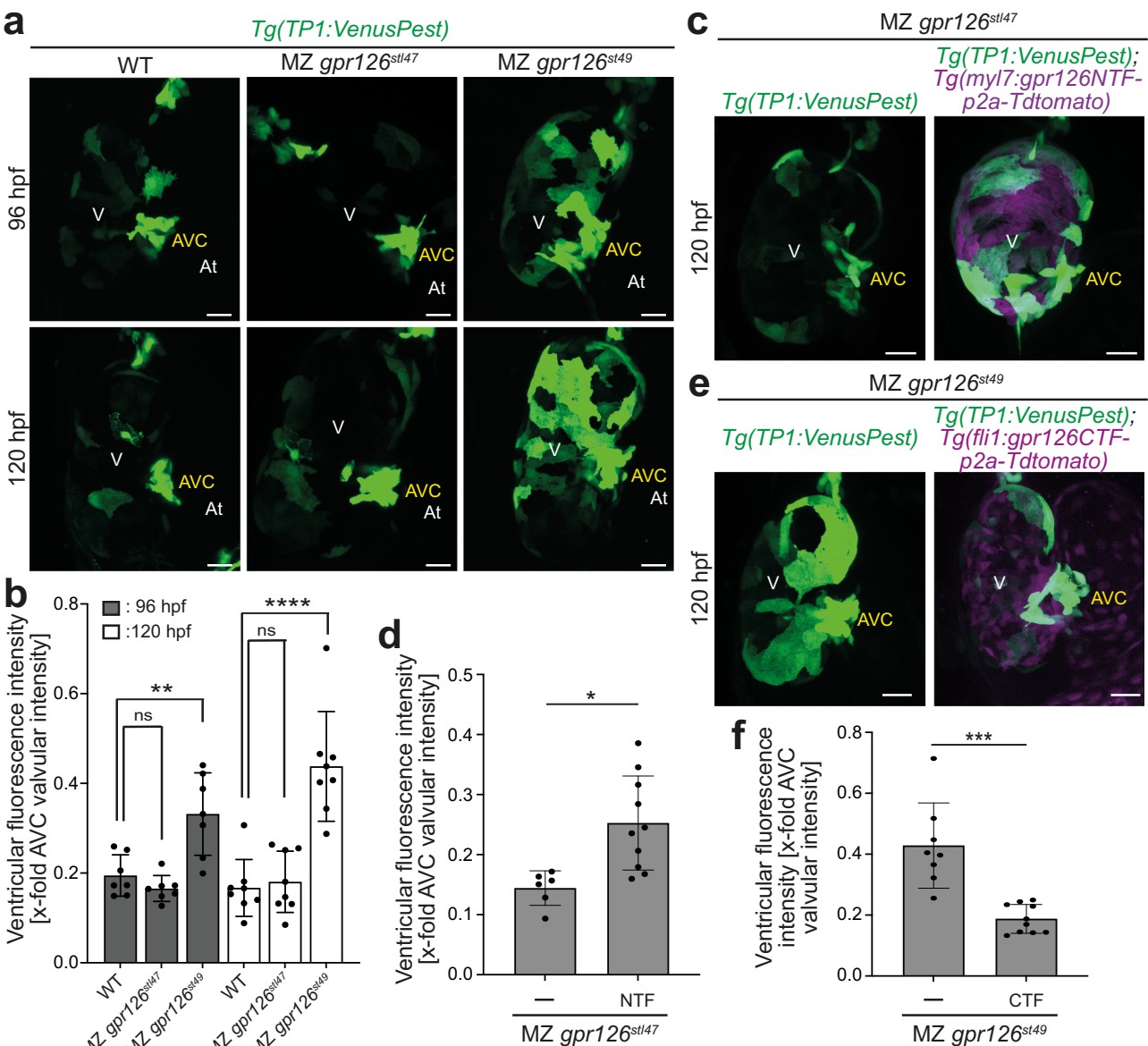

**Fig. 6 | *gpr126* modulates myocardial Notch activity. a** Maximum intensity projections of representative *Tg(TP1:Venus-PEST)*; WT, maternal zygotic (MZ) *gpr126^{stl47}*, and MZ *gpr126^{st49}* hearts showing active Notch signaling at 96 and 120 h post fertilization (hpf). MZ *gpr126^{st49}* hearts express significantly higher myocardial Notch activity as compared to WT and MZ *gpr126^{stl47}* hearts. V: ventricle. At: atrium. AVC: atrioventricular canal. Scale bars: 20 μm. **b** Quantification of Venus-Pest intensity with reference to valves for WT, MZ *gpr126^{stl47}*, and MZ *gpr126^{st49}* hearts at 96 hpf (*n* = 7) and 120 hpf (*n* = 8). Each point represents a heart. WT vs. MZ *gpr126^{stl47}*, 96 hpf: *p* = 0.5934. WT vs. MZ *gpr126^{st49}*, 96 hpf: *p* = 0.0011. WT vs. MZ *gpr126^{stl47}*, 120 hpf: *p* = 0.9323. WT vs. MZ *gpr126^{st49}*, 120 hpf: *p* < 0.0001 (one-way ANOVA with Dunnett's multiple comparisons test). Data are mean ± S.D. **: *p* < 0.01; ****: *p* < 0.0001, ns: not statistically significant. **c** Maximum intensity projections of representative *Tg(TP1:Venus-PEST)*; MZ *gpr126^{stl47}* and *Tg(TP1:Venus-PEST)*; *Tg(myl7:gpr126NTF-p2a-tdTomato)*; MZ *gpr126^{stl47}* hearts showing active Notch

signaling at 120 hpf. NTF overexpressing MZ *gpr126^{stl47}* hearts exhibit significantly increased myocardial Notch activity. **d** Quantification of Venus-Pest intensity with reference to valves at 120 hpf for *Tg(TP1:Venus-PEST)*; MZ *gpr126^{stl47}* (*n* = 6) and *Tg(TP1:Venus-PEST)*; *Tg(myl7:gpr126NTF-p2a-tdTomato)*; MZ *gpr126^{stl47}* (*n* = 10) hearts. Each point represents a heart. *p* = 0.0061 (two-tailed unpaired Student's t-test). Data are mean ± S.D. *: *p* < 0.05. **e** Maximum intensity projections of representative *Tg(TP1:Venus-PEST)*; MZ *gpr126^{st49}* and *Tg(TP1:Venus-PEST)*; *Tg(fli1:gpr126CTF-p2a-tdTomato)*; MZ *gpr126^{st49}* hearts showing active Notch signaling at 120 hpf. CTF overexpressing MZ *gpr126^{st49}* hearts exhibit reduced myocardial Notch activity as compared to MZ *gpr126^{st49}* hearts. **f** Quantification of Venus-Pest intensity with reference to valves at 120 hpf for *Tg(TP1:Venus-PEST)*; MZ *gpr126^{st49}* (*n* = 8) and *Tg(TP1:Venus-PEST)*; *Tg(fli1:gpr126CTF-p2a-tdTomato)*; MZ *gpr126^{st49}* (*n* = 10) hearts. Each point represents a heart. *p* = 0.0001 (two-tailed unpaired Student's *t* test). Data are mean ± S.D. ***: *p* < 0.001.

modulating Nrg/ErbB2 signaling[62,63], and heterogeneous expression of *erbb2* in the single-layer myocardium leads to tension heterogeneity initiating trabeculation, which triggers Notch activity in adjacent cardiomyocytes, suppressing erbb2, preventing their delamination[64]. However, various endocardial/endothelial angiocrine factors have also been shown to regulate ventricular compaction in humans[65]. These reports further corroborate that endocardial-myocardial crosstalk is an essential component of ventricular wall morphogenesis, which is

reinforced by Notch-mediated lateral inhibition and warrants further investigation into how endocardial factors like Gpr126 regulate trabeculation.

We propose a model where Gpr126-NTF is secreted from the endocardium and acts on the myocardium to maintain the cell-cell adhesion between compact wall cardiomyocytes. Once the compact wall integrity is achieved, trabeculation begins, and CTF signaling assists in modulating myocardial Notch activity to determine the fate

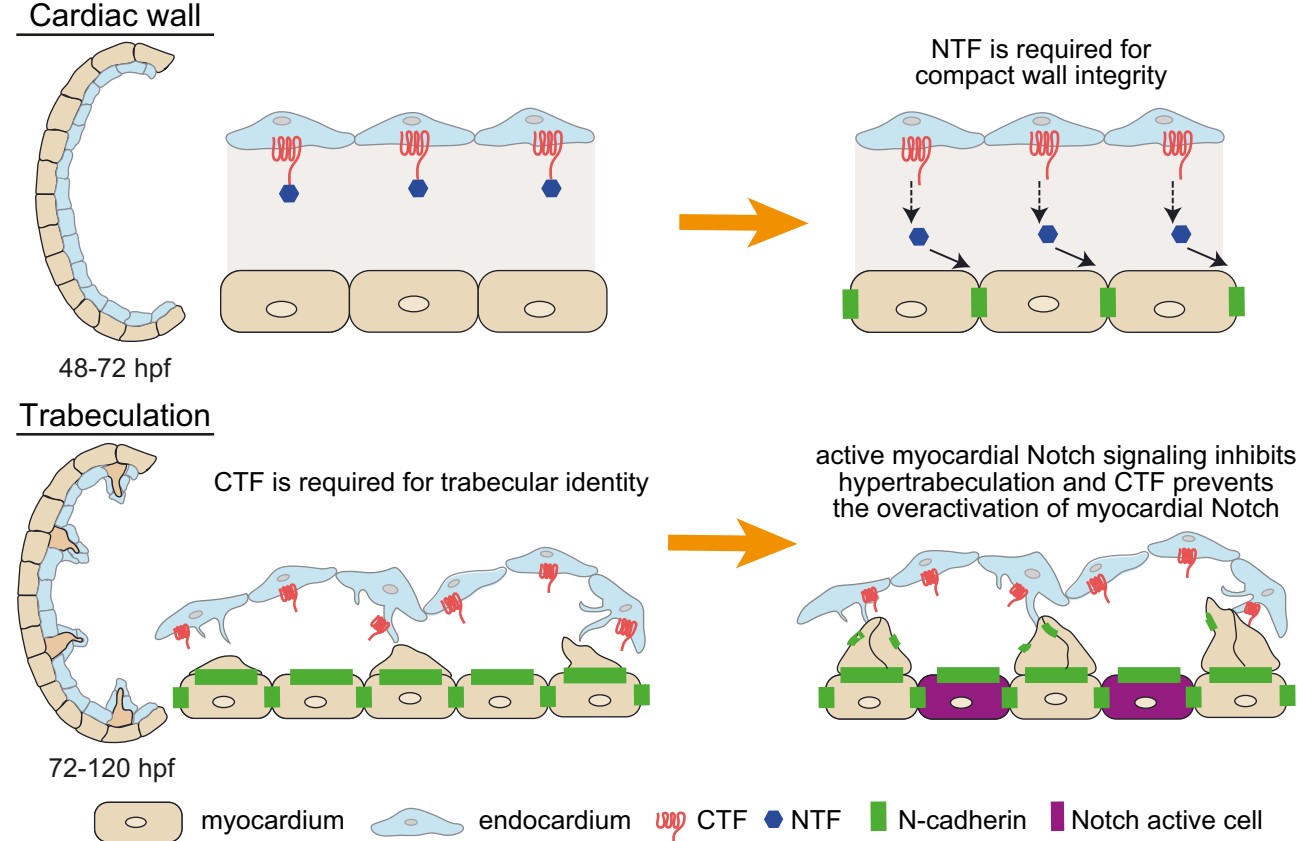

**Fig. 7 | Schematic model of the role of Gpr126-NTF and -CTF during ventricular wall development.** Our data support the following model for the role of Gpr126 during cardiac trabeculation: Gpr126 is expressed in the endocardium and cleaved into NTF and CTF. The NTF is secreted and acts on the myocardium to maintain the N-cadherin localization between the compact wall cardiomyocytes. Once the compact wall integrity is achieved, trabeculation begins, and CTF signaling assists in N-cadherin relocalization and restricting myocardial Notch activity to determine the fate of trabecular vs. non-trabecular cardiomyocytes. Our data suggest that the CTF is required to balance NTF-mediated signaling, preventing overactivation of myocardial Notch signaling. The endocardial cells form protrusions to interact directly with the cardiomyocytes and facilitate CTF signaling to act on the myocardium. This model has limitations as details regarding the polarized localization and how the NTF and CTF act on the myocardium are currently unknown. hpf: hours post fertilization.

of trabecular *vs.* non-trabecular cardiomyocytes. The NTF/CTF balance is essential for the Notch-mediated lateral inhibition, which is integral for the myocardial pattern formation (Fig. 7). Our model has its limitations, as various details regarding the polarized localization of Gpr126 and the actual mechanisms of action of NTF and CTF on the myocardium are currently unknown. Further investigation is needed to decipher the signaling mechanisms orchestrating the non-autonomous role of CTF on the myocardium. Thus, our study provides further evidence that aGPCRs play a role in cell adhesion in addition to cellular signal transduction[43,66]. Our results are of great importance considering that aGPCR mutations are associated with human diseases[67–69], congenital heart disease is the most common congenital disease affecting around 1% of live births[30,70,71], and the therapeutic potential of targeting aGPCRs[72].

## Methods
All research presented here complies with all relevant ethical regulations approved by the Tierschutzkommision (Animal Welfare Commission) of the REGIERUNG VON UNTERFRANKEN (local government, protocols: AZ.55.2.2-2532.2-1145-13 and AZ.I/39-1/FN003) and conforms to the guidelines from Directive 2010/63/EU of the European Parliament on the protection of animals used for scientific purposes.

### Zebrafish maintenance
Embryonic and adult zebrafish were raised and maintained in Tecniplast aquaculture main and stand-alone systems. The water temperature was maintained at 28.5 °C and the pH and conductivity were set at 7–7.5 and 450–500 μS. The light-dark cycle consisted of half an hour of gradual onset of light (8 am to 8:30 am) with 13.5 h of uninterrupted light and half an hour of gradual onset of darkness with 11.5 h of complete darkness. Zebrafish were fed three times a day, depending on age, with rotifers and granular food. Health monitoring was performed at least once per year.

### Zebrafish mutant and transgenic lines
In this study, the following transgenic and mutant lines were used: *Tg(myl7:EGFP-hsa.HRAS)s883*[73], *Tg(−0.2myl7:EGFP-podocalyxin)bns10*[41], *Tg(myl7:mKate-CAAX)sd11*, abbreviated as *myl7:mKate-CAAX*[74], *TgBAC(cdh2:cdh2-eGFP,crybb1:ECFP)zf517*, abbreviated as *TgBAC(cdh2:cdh2-GFP)*[75], *Tg(EPV.Tp1-Mmu.Hbb:Venus-Mmu.Odc1)s940*, abbreviated as *TP1:VenusPest*[76], *gpr126*[stl47 43], *gpr126*[st49 15], *Tg(krt4:GFP)*[sqet33 47], *Tg(fli1: gpr126-p2a-tdTomato)*, *Tg(myl7:gpr126NTF-p2a-tdTomato)*, and *Tg(fli1: tdTomato-p2a-gpr126CTF)*.

### Genotyping and matings of *gpr126* mutants
*stl47* were genotyped as previously described[43]. Briefly, the primers listed in Table S2 were used to amplify a 498 bp product from genomic DNA. Subsequently, the PCR products were digested with *Mfe1* restriction enzymes. As the *Mfe1* recognition site is disrupted in the mutant, only the WT PCR product is cleaved into 257 bp and 241 bp fragments. For *st49*, the primers listed in Table S2 were used to amplify a 604-base pair (bp) product from genomic DNA. The point mutation

in st49 introduces a *Mae*III recognition site, and the restriction digestion of the PCR product yields 437-bp and 167-bp fragments. The 604-bp product is retained in the digested WT DNA. For generating maternal zygotic (MZ) animals, homozygous *gpr126*[stl47] or *gpr126*[st49] mutant females were crossed with their heterozygous male counterparts. The WT siblings from heterozygous crosses were grown to adulthood, and their progeny were used as controls for MZ experiments and were referred to as WT cousins.

### RNA isolation from zebrafish larvae hearts and quantitative PCR

Larval zebrafish hearts were isolated from WT and mutant embryos in the background of *Tg(myl7:EGFP-hsa.HRAS)* at 120 hpf following the protocol used by Burns and MacRae[77]. Larvae (*n* = 200) were first anesthetized using 0.04% tricaine methanesulfonate (MS-222) and collected into 1.5 ml microcentrifuge tubes. The larvae were washed three times with ice-cold embryo disruption medium (EDM) and were resuspended in 1.25 ml EDM. Hearts were then fragmented from the larvae by mechanical disruption using a 5 ml syringe and a 1½ inch long 19-gauge needle. This was done by collecting the contents of the tube in the syringe through the needle and expelling it back into the microcentrifuge tube 30 times at the rate of 1 s per syringe motion. Fragmented larvae were filtered through a 105 μm nylon mesh, and the flow-through was collected in a 30 mm petri-dish. The flow-through was then applied to a 40 μm nylon mesh, which was then inverted, and the retained material was collected with EDM in a 30 mm petri dish. The beating hearts were identified with their EGFP expression under the Zeiss stereoscope and were collected into a 1.5 ml microcentrifuge tube containing RNAlater. All these steps were performed on ice. The hearts were then pelleted for 10 min at 10,000 x g. The supernatant was discarded, and TRIzol was added to the pellet, and RNA was isolated using RNeasy Micro Kit (Qiagen) according to manufacturer's instructions. For production of cDNA, 500 ng of RNA was reverse transcribed using Oligo (dT) 12-18mer primers and M-MLV Reverse Transcriptase (Thermo Fisher Scientific, Cat# 280250139) according to the manufacturer's instructions. qPCR assays were performed in experimental triplicates using SYBR Green (Bio-Rad) in a CFX Connect™ Real-Time PCR (Bio-Rad). *gpr126* expression values were normalized using the housekeeping gene *rpl13a* and fold changes were calculated using the $2^{-\Delta\Delta Ct}$ method. The primers for *gpr126* amplifies part of exon 3 and 4 and the sequences used for qPCR can be found in Table S2.

### RNAscope assay

Zebrafish embryos were fixed overnight at 48 and 96 hpf in 4% PFA in PBS at RT. A series of increasing methanol concentrations (25%, 50%, 75%, 2× 100%) in 0.1% PBST was used to dehydrate the embryos stepwise in 10 min washes. The embryos were then rehydrated with decreasing methanol concentrations (75%, 50%, and 25%) and subjected to the RNAscope-based target retrieval by incubating the embryos in 1X target retrieval solution (Advanced Cell Diagnostics) at 100 °C for 15 min. Protease digestion of embryos using RNAscope Protease Plus (Advanced Cell Diagnostics) was performed for 15 min for 48 hpf embryos and 30 min for 96 hpf embryos at 40 °C by floating the tubes in a water bath. The probe hybridization and staining were performed using RNAscope Multiplex Fluoroscent v2 Assay kit (Advanced Cell Diagnostics), The target probe was hybridized at 40 °C for 2 h followed by washing in 1X wash buffer for 10 min. For RNA detection, the amplification steps were performed by incubation with the different amplifier solutions in a water bath at 40 °C followed by washing in 1X wash buffer for 10 min. The signal was then developed by incubating the embryos in HRP-C2 solution, followed by incubation with the fluorescent dye Opal 570, and then incubating in HRP blocker solution. All these steps were performed in 40 °C water bath for 15 min each, followed by washing with 1X wash buffer for 10 min twice. After the final wash, the embryos were immunostained by directly

proceeding to the blocking step in the whole-mount immunostaining protocol.

### Whole-mount immunostaining of zebrafish embryos

Embryos were treated with PTU at 24 hpf to avoid pigmentation. The embryos were anaesthetized with 0.4% (w/v) tricaine to stop the heartbeat, just before fixation, to prevent the heart from collapsing during fixation. 10 embryos were fixed overnight in 4% paraformaldehyde in 2 ml tubes. The fixative was washed three times using PBST. The embryos were permeabilized using proteinase-K (3 μg/ml) in PBST for 1 h. This treatment was followed by washes with PBDT buffer (PBS, 1% BSA, 1%DMSO, 0.5% Triton X-100), and blocking was performed for 1 h with 10% sheep serum in PBDT. Primary antibody (Mouse anti-FLAG M2, Sigma Aldrich, F3165,1:500) incubation was performed in a 4 °C nutator in PBDT buffer with 1% sheep serum (gentle shaking). After washing with PBDT, the embryos were incubated in secondary antibody (donkey anti-mouse conjugated to Alexa Fluor −488, −594, and −647, Life Technologies,1:500) in PBDT buffer with 1% sheep serum for 2 h at RT. This step was followed by 15 min incubation in DAPI (2 μg/ml) and final washes with PBST.

### Plasmid constructs

For overexpression of zebrafish *gpr126* and its mutant forms in mammalian cell lines, constructs were created using site-directed mutagenesis. HA-gpr126 fragment was amplified by PCR from pcDps-HA-gpr126-FLAG gifted by Ines Liebscher and cloned into the mammalian expression vector pAcGFP to create HA-*Gpr126*-GFP construct. Inserts with the *stl47* and *st49* mutations were generated through PCR by introducing complementary mutations in the primers, as listed in Table S2. The inserts with the mutations were assembled using the NEBuilder HiFi assembly master mix (New England Biolabs, Cat# E2621L) according to the manufacturer's instructions, and transformed into competent bacteria. To generate the *fli1a:gpr126-p2a-tdTomato* and *fli1a:gpr126CTF-p2a-tdTomato* plasmids, sequences encoding the Gpr126 and Gpr126-CTF were PCR amplified. The amplified coding sequences followed by *p2a-tdTomato* were cloned downstream of the endothelial-specific *fli1a* promotor into a Tol2-enabled vector using the Cold Fusion Cloning kit (System Biosciences, Cat# MC010B-1). The CTF construct was designed with the signal peptide followed by Gpr126-CTF tagged with HA at the N-terminus and FLAG at the C-terminus[9]. In addition, P2Y$_{12}$ N-terminus peptide was fused to the Gpr126 CTF to facilitate the plasma membrane localization of the receptor. The NTF construct used was NTF$^{\Delta GPS}$ as described before[22].

### ARPE19 cells and plasmid transfection

The ARPE-19 cell line (ATCC, CRL-2302) was cultured in Dulbecco's modified Eagle's medium/Nutrient mixture F-12 (DMEM/F-12) supplemented with 10% fetal bovine serum (FBS) and 100 U/ml penicillin, and 100 μg/ml streptomycin. Cells were maintained in a humidified atmosphere with 5% $CO_2$ at 37 °C. Plasmid transfection into ARPE-19 cells was carried out with 500 ng DNA per well of a 24-well plate using 1 μl Lipofectamine LTX (Thermo Fisher Scientific, Cat# 15338100)) according to the manufacturer's instructions. Transfection complexes were formed by incubating DNA with Lipofectamine LTX in Opti-MEM for 20 min at room temperature. For WB analysis, transfection was performed in 6-well plates, and the volumes were adjusted by using fourfold amounts of reagents.

### Production of lentiviral vectors

The lentiviral packaging plasmid psPAX2 and the VSV-G envelope plasmid pMD2.G were gifts from Didier Trono (psPAX2: Addgene plasmid #12260; http://n2t.net/addgene:12260; RRID: Addgene_12260; pMD2.G: Addgene plasmid #12259; http://n2t.net/addgene:12259; RRID: Addgene_12259). The transfer plasmid pLenti CMVtight Blast

DEST (w762-1) was a gift from Eric Campeau (w762-1: Addgene plasmid #26434; RRID: Addgene_26434). The coding sequences of HA-Gpr126-GFP wt and mutants were cloned into w762-1 by NEBuilder® HiFi DNA Assembly according to the manufacturer's instructions using the primers indicated in Table S2. To produce lentiviral vectors, psPAX2, pMD2.G, and the desired transfer plasmid were transfected in a 1:1:2 ratio into HEK293T (ATCC, CRL-3216) cells using PEI MAX (Polysciences, Cat# 24765-1). Supernatant containing lentiviral vectors was harvested 72 h after transfection, filtered through a 0.45 μm filter, and aliquots were snap frozen.

### Generation of stable ARPE-19 cell lines
Lentiviral vector aliquots were rapidly thawed at 37 °C and diluted in growth medium containing 10 μg/ml Polybrene (Sigma-Aldrich, Cat# 107689). Cells were transduced overnight in 6-well plates using 1 ml of diluted lentiviral vector. The following morning, medium was refreshed, and cells were selected with 10 μg/ml of blasticidin for transgene integration 72 h after transduction.

### Heart-specific RNA-sequencing
Larval hearts were collected from the gpr126 mutants and their WT cousins. 30 hearts from one clutch were obtained and pooled per biological replicate (4 biological replicates for each genotype). RNA was isolated using miRNeasy micro Kit (Qiagen) with on-column DNase digestion (RNase-Free DNase Set, Qiagen) to avoid contamination by genomic DNA. RNA and library preparation integrity were verified with LabChip Gx Touch (Perkin Elmer). RNA amounts were normalized, and 1 ng of total RNA was used as input for SMART-Seq® HT Kit following manufacture's protocol (Takara Bio). Sequencing was performed on the NextSeq 500 platform (Illumina) using a P3 flowcell with 75 bp single-end setup. The resulting raw reads were assessed for quality, adapter content, and duplication rates with FastQC (Andrews S. 2010, FastQC: a quality control tool for high throughput sequence data. Available online at: http://www.bioinformatics.babraham.ac.uk/projects/fastqc). Trimmomatic v0.39[78] was employed to trim reads after a quality drop below a mean of Q18 in a window of 5 nucleotides. Only reads above 30 nucleotides were employed for further analyses. Trimmed and filtered reads were aligned versus the zebrafish genome version danRer11 (ensembl release 99) using STAR v2.7.3a[79] with the parameter "--outFilterMismatchNoverLmax 0.1" to increase the maximum ratio of mismatches to mapped length to 10%.

### Full body RNA-sequencing
Full-length RNA was isolated from zebrafish larvae using the RNeasy mini kit (Qiagen) combined with on-column DNase digestion (RNase-Free DNase Set, Qiagen) to avoid contamination by genomic DNA. Library preparation was performed relying on the Illumina Stranded mRNA Prep kit (Illumina) using as input 200 ng of RNA for each sample. Sequencing was performed on a NovaSeq 6000 platform (Illumina) (paired-end; 2 × 159 bp). Conversion of reads to fastq format, and demultiplexing were performed with bcl2fastq v2.17. Raw reads were analyzed relying on the nf-core RNA-Seq pipeline v3.9[80] (https://doi.org/10.5281/zenodo.1400710) using default parameters. In short, reads were adapter- and quality-trimmed using Trim Galore v0.6.7 (Babraham Bioinformatics) and mapped to the Ensembl *danio rerio* genome assembly GRCz11 (release 109)[81] using STAR v2.7.10a[79] to generate BAM files.

### Differential exon usage analysis
Full body and heart-specific differential exon usage (DEU) analyses were performed in R v4.2.2 (https://www.R-project.org) relying on the DEXSeq package v1.44.0[82,83] following the official documentation (Appendix section 10.2 (https://bioconductor.riken.jp/packages/3.16/bioc/vignettes/DEXSeq/inst/doc/DEXSeq.html). First transcript annotations from the Ensembl *danio rerio* gene annotation file release 109,

with coordinates of exon boundaries, were extracted and stored in a TxDb object using the function makeTxDbFromGFF() from the GenomicFeatures R package v1.50.4[84]. The TxDb object was provided as input to the GenomicFeatures function exonicParts() to create non-overlapping exon counting bins. To this aim, the argument "linked.to.single.gene.only" was set to TRUE. Counting of read fragments overlapping with the exon counting bins was performed relying on summarizeOverlaps() from the R package GenomicAlignments v1.34.1[84] using the count method "Union". For heart-specific DEU analysis, the function was run with the arguments "nter.feature=FALSE", "singleEnd=TRUE", "ignore.strand=TRUE", and "fragments=FALSE". Instead, for full body DEU analysis, the following parameters were set: "singleEnd=FALSE", "inter.features=FALSE", and "fragments=TRUE". After read counting a DEXSeqDataSet object was built by calling DEXSeqDataSetFromSE() on the output of summarizeOveralps(). The estimateSizeFactors() and estimateDispersion() functions were used to estimate size factors and dispersions, respectively. Ultimately, DEU was performed through the DEXSeq function testForDEU() and differences in DEU were extracted using the function DEXSeqResults(). The plots of exon usage coefficient estimates were generated relying on the plotDEXSeq() function from DEXSeq, setting "expression=F" and "splicing=T". An exon counting bin was defined as differentially used if its adjusted p-value was lower than 0.05.

### RT-PCR for validating DEU analysis
PCR was carried out using 10 ng of cDNA with RedTaq DNA polymerase master mix from Genaxxon (Ulm, Germany). Primers used are provided in Table S2.

### Immunofluorescence
For detection of HA-tag via immunofluorescence staining, cells were rinsed once with D-PBS and were fixed with 4% formaldehyde/PBS for 10 min at room temperature. Formaldehyde-fixed cells were permeabilized for 10 min with 0.5% TritonX-100/PBS. Prior to antibody staining, samples were blocked for 20 min using 5% BSA in 0.2% Tween-20 in PBS. The primary antibody (Rabbit anti-HA, Abcam, ab9110) was diluted (1:500) in the blocking reagent and incubated with the sample overnight at 4 °C in a humidified chamber. After removal of primary antibody solution and three 5 min washes with 0.1% NP40/PBS, samples were incubated for 60 min with fluorophore-coupled secondary antibody (donkey anti-rabbit conjugated to Alexa Fluor −594, and −647, Life Technologies) diluted in blocking reagent (1:500). DNA was visualized with 0.5 μg/ml DAPI (4',6'-diamidino-2-phenylindole) in 0.1% NP40/PBS. After DAPI staining, cover slips were rinsed once with Millipore-filtered water and then mounted using Fluoromount-G mounting medium.

### Western blotting
Transfected ARPE19 cells were washed with ice-cold PBS and extracted in RIPA buffer supplemented with EDTA-free protease inhibitor cocktails (cOmplete, Roche # 11873580001). After 30 min incubation on ice, samples were sonicated, and lysates were cleared by centrifugation at 16,000 × g for 10 min at 4 °C. Lysates were analyzed by SDS-PAGE (4–12% NuPAGE Novex Bis-Tris gels) under reducing conditions and transferred to nitrocellulose membrane by wet transfer at 30 V and 300 mA for 1.5 h in 1x transfer buffer (25 mM Tris-HCl, pH 7.5, 192 mM glycine, 0.1% SDS, 10% methanol). The membrane was then blocked with 1x TBS, 0.05% Tween-20 and 10% non-fat milk and incubated with primary antibodies against HA and GFP (Rabbit anti-HA, Abcam, ab9110, 1:500; Mouse anti-GFP, Abcam, ab38689, 1:500). Enhanced chemiluminescence reagent (PerkinElmer, Waltham, MA) was used for protein detection. The cell supernatant was collected and centrifuged for 300 × g for 5 min at RT to remove any leftover cells and concentrated using a Vivaspin centrifugal concentrator (Sartorius).

### FACS sorting and quantitative PCR

Larval hearts from *Tg(myl7:GFP-CAAX)*; *Tg(kdrl:nls-mCherry))* at 48 and 96 hpf were manually dissected in DMEM + 10% FBS, centrifuged for 3 min at 5000 rpm, washed in 1 ml of Hanks' Balanced Salt Solution, and dissociated into single cells in 100 µl of Enzyme 1 and 5 µl of Enzyme 2 (Pierce Cardiomyocytes Dissociation Kit, ThermoFisher Scientific) for 30 min at 350 rpm shaking at 30 °C. The cells were centrifuged for 5 min at 5000 rpm, and the dissociation media was removed and replaced by DMEM + 10% FBS. The cells were then sorted using a BD FACSAria™ III—firstly by forward scatter amplitude and side scatter amplitude, and subsequently sorted by GFP and mCherry fluorescence into Trizol. More than 100 hearts were pooled together for each independent experiment. RNA extraction was then performed using Qiagen RNeasy Kit. At least 100 ng of total RNA was used for reverse transcription using Maxima First Strand cDNA synthesis kit (Thermo Fisher Scientific). DyNAmo ColorFlash SYBR Green qPCR Mix (Thermo Fisher Scientific) was used on a CFX connect Real-time System (Bio-Rad) with the following program: pre-amplification 95 °C for 7 min, amplification 95 °C for 10 s and 60 °C for 30 s (repeated for 39 cycles), melting curve 60-92 °C with increment of 1 °C every 5 s. Each point in the dot plots represents a biological replicate from three technical replicates. Gene expression values were normalized using the housekeeping gene *rpl13a* and fold changes were calculated using the $2^{-\Delta\Delta Ct}$ method. The qPCR primers for *gpr126* span Exon 3 to Exon 4.

### Zebrafish transgenesis

Tol2 transgenesis was used to generate the *Tg(myl7:gpr126NTF-p2a-tdTomato)*; *Tg(fli1:gpr126-p2a-tdTomato)* and *Tg(fli1:tdTomato-p2a-gpr126CTF)* lines. To generate stable lines, wild-type AB embryos (F0) were injected at the one-cell stage with an injection cocktail of transposase (tol2 mRNA, 12.5 pg per embryo) and plasmid DNA (25 pg per embryo). F0 embryos positive for tdTomato fluorescence were raised to adulthood and then screened for founder animals. The founders (three for each transgenic line) were further outcrossed to raise the F1 generation.

### Morpholino injection

In order to stop contractility and blood flow in the embryos, 0.25 ng of *tnnt2a* morpholino (5'-CATGTTTGCTCTGATCTGACACGCA-3') was injected into single-cell-stage embryos.

### Chemical treatments

ErbB2 signaling was inhibited by treating the embryos with 10 µM of ErbB2 inhibitor PD168393 (Calbiochem).10 µM working solution of PD168393 was prepared in 1x E3 media with 0.003% PTU. For the treatment, 10 embryos per well were placed in a 6-well plate containing the working solution of PD168393. As a control, 1% DMSO was used. The treatment was performed from 60 hpf to 120 hpf. At 120 hpf, the PD168393 solution was removed, and the embryos were washed three times with 1x E3 medium for 5 min and subsequently imaged.

WT and mutant embryos were treated with forskolin (2 × 6 h pulses, at 60-66 hpf and 84-90 hpf) at 25 µM or IBMX (3-isobutyl-1-methylxanthine) (continuous treatment, 60-96 hpf at 125 µM in 1x E3 medium with 0.003% PTU. Control embryos were treated with equivalent amounts of DMSO.

The ARPE-19 cells were treated with 3 µg/ml brefeldin A 24 h after transfection. Before the treatment, the cell medium was replaced with a serum-free medium, and the supernatant was collected 8 h after brefeldin A treatment.

### Imaging

Image acquisition for immunostained cells as well as zebrafish hearts was performed using Carl Zeiss AG LSM800 confocal laser scanning microscope equipped with the ZEISS blue software. For live imaging of stopped zebrafish hearts, embryos and larvae were mounted in 1% low-melting agarose on glass-bottom dishes (MatTek). The larvae were oriented through a micromanipulator under the Carl Zeiss AG Axiozoom V16 stereoscope to enhance optical access to the heart. In order to stop the heart beating before imaging, 0.2% (w/v) tricaine was added to the low-melting agarose. A 40x (1.1 numerical aperture [NA]) water-immersion objective was used to acquire the images. The optical sections were 1 µm thick.

### Image processing and Quantification

2D and 3D images were processed using ImageJ and the Fiji extension package. Figures were prepared with the help of Adobe Illustrator or Microsoft PowerPoint. The exon and protein structure in Figs. 1 and S7 were created by Illustrator for Biological Sequences 2.0[49]. The criteria we have chosen for normal trabeculation were 3 or more trabeculae with 3 or more cardiomyocytes for 96 hpf hearts and 5 or more trabeculae with 3 or more cardiomyocytes for 120 hpf hearts. The quantification of intensity profiles of HA- and GFP-tag in ARPE-19 cells was performed using the "Measure" tool. The GFP intensity for each condition was measured relative to the HA intensity. For the quantification of myocardial Notch activity, intensity profiling of the YFP signal was performed. The total YFP signal relative to the YFP signal in the valve endocardium was plotted as the intensity of myocardial Notch. Three-dimensional surface rendering was performed using the Surfaces function of Imaris x64 (Bitplane). Bright-field images obtained from the stereoscope were processed using Zen blue (ZEISS) software.

### Statistical analysis and reproducibility

Statistical analysis and *p*-value (p) calculation were performed using Graphpad Prism 8.3.0 and 10.2.0. Data are expressed as mean ± SEM or SD as indicated. Differences between the two groups were determined with a two-tailed Student's *t* distribution test. To compare multiple conditions, one-way ANOVA was performed, and when data were compared to control groups, correction for multiple comparisons was performed by using Dunnett's test. All tests were performed with a confidence level of 95%. All experiments were repeated at least three times unless otherwise specified.

### Reporting summary

Further information on research design is available in the Nature Portfolio Reporting Summary linked to this article.

## Data availability

The sequencing data generated in this study have been deposited in the European Nucleotide Archive (ENA) at EMBL-EBI under accession numbers PRJEB98567 and PRJEB98842. Source data are provided with this paper.

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

## Acknowledgements

We are grateful to Kelly Monk, Robert Becker, Salvador Cazorla Vazquez, and Gentian Musa for discussions and critical reading of the manuscript. We would like to thank Ines Liebscher for zebrafish Gpr126 constructs, Sebastian Zundler for providing IMARIS software, Arwin Groenewoud for Forskolin injections in the bloodstream, Darren Gilmour for providing *TgBAC(cdh2:cdh2-EGFP)*, as well as Jana Petzold, Christina Goula, and Jennifer Redlingshöfer for excellent care of the zebrafish facility. S.S. acknowledges the support from Filomena Ricciardi for sharing her expertise in trabeculation imaging. This work was supported by the Deutsche Forschungsgemeinschaft (DFG, INST 410/91-1 FUGG as well as FOR2149, Project 7, EN 453/10-1 and EN 453/10-2 to F.B.E., and project number 509149993, TRR 374 to F.F.), the Deutsche Herzstiftung e.V. to F.B.E., and Bavarian Equal Opportunities Sponsorship—Realisierung von Frauen in Forschung und Lehre (FFL)—Promotion Equal Opportunities for Women in Research and Teaching funding to S.S. The generation of *stl47* allele was supported by NIH NINDS F32 NS087786 funding to S.C.P. Work in the Stainier lab is supported in part by the Max Planck Society.

## Author contributions

S.S. designed, performed and analyzed most of the experiments. F.G. and A.G. performed the FACS sorting for endocardial and myocardial cells from zebrafish larval hearts and provided *gpr126* expression data in the sorted cells. S.V. generated the stable ARPE-19 cell lines and performed molecular validation of the bioinformatics data analyses; S.G. performed the heart-specific RNA-sequencing analysis. M.A. and F.F. performed the full-body RNA-sequencing analysis and the differential exon usage analysis. S.C.P. generated and provided the *stl47* allele. S.S. and F.B.E. designed all experiments, interpreted results, and wrote the manuscript. D.Y.R.S. provided constant guidance with the project as well as various transgenic lines and supervised F.G. and A.G. F.B.E. supervised the project. All the authors contributed to the discussion of data and refining of the manuscript.

## Funding

## Competing interests

The authors declare no competing interests.
