## [Peer Review file · Nature Communications]

Adgrg6/Gpr126 is required for compact wall integrity and establishing trabecular identity during cardiac trabeculation

Corresponding Author: Professor Felix Engel

Version 0:

Reviewer comments:

Reviewer #1

(Remarks to the Author)

The manuscript by Srivastava et al. explores the function of the Adhesion GPCR ADGRG6/Gpr126 in heart development. It follows up on previous studies of the Engel group (Patra et al. 2013) and shows that the receptor controls heart development and in particular, trabeculation in the zebrafish, potentially via two different mechanisms (adhesion and cell integrity) mediated separately by the receptor NTF and CTF. Finally, they propose an involvement of the receptor in the regulation of Notch activity.

The study by Srivastava et al. is very interesting and points towards exciting functions of the Adhesion GPCR ADGRG6/Gpr126, which also can give some indication on how Adhesion GPCRs in general function. However, many of the experiments are very preliminary and the interpretation sometimes is a little doubtful. I have several comments, especially regarding the use of the strains and constructs, which in my opinion need careful re-evaluation. In my opinion, the study in its current form is not complete and requires substantial more work to demonstrate many of the claims made, such as the CTF function and the regulation of Notch activity.

Major concerns:

1. st49 mutant: I appreciate that the mutant has been characterized previously. However, both, st49 and st47 disrupt essential domains of the receptor, the CUB and the GAIN, respectively, which carries the risk of (partially) misfolded proteins. This probably does not matter for st47 as there is not much protein left and I agree that it can be considered a complete loss of function. However, it does matter for st49. The authors try to address this issue by expressing the construct in cells. However, as Fig. 1B shows, the construct accumulates in the cytoplasm. The authors state that this is due to the missing 7TM. I would assume that a missing 7TM leads to a liberated protein, which gets secreted rather than retained, especially as the signal peptide is still present. This is at least what we see for other GPCRs. The authors also postulate that this occurs in the st49 strain (p. 15, l. 326). I am highly worried what really happens to the protein. It might lead to dominant-negative effects in the fish and thus, to phenotypes

I would like to see the experiments (or at least part of them) conducted in a fish strain carrying a "clean" version of the N terminus – liberated or even tethered to the membrane. I realize that this some effort, but it would generate much more solid data. With CRISPR-Cas this should be doable.

2. Fig. 1B: It looks like there is less membrane localization of Gpr126 st49, but increased cytoplasmic localization. The authors state that this is due to the missing 7TM. As mention in comment 1, I would assume that the missing 7TM leads to a liberated protein that gets secreted rather than retained as the signal peptide is still present. I would assume other reasons for the cytoplasmic expression are more likely such as a partially misfolded protein.

3. p. 9: Why does the st47 mutant not show a zygotic phenotype, but the st49 does? Maybe I am misunderstanding this, but if this is the case, it is a really interesting finding. It might either point towards a separate function of the NTF that is only visible when the CTF is absent, or it can be a dominant-negative effect of the st49 (see comment 1). The authors need to discriminate between the different options and explain this matter.

4. p. 10: How does the CTF in Tg(fli1a:gpr126 CTF-p2a-tdTomato) look like? Does it have a signal peptide? Also, from the description in the Methods section, it seems that it does not have a tag or anything on its N-terminal side. A pure CTF might be rather instable and not be expressed too well at the membrane. This has been previously described in the context of Gpr126 (Liebscher et al. 2015). The authors do show expression in Fig. S3B, however, demonstrating proper expression of

this construct at the cell membrane is absolutely necessary for the interpretation of the results. Thus, a tissue culture experiment would be good.

5. p 10: The authors claim a CTF and an NTF function of Gpr126 in heart development. I am a bit worried about the postulated CTF function in this context. I agree that the CTF is required, but I am not sure that it is a sole CTF function. It has been shown for quite some aGPCR that NTF and CTF can reconstitute to form a (function) receptor (e.g. Silva et al. 2009, Prömel et al. 2012). As in st49, the N terminus is still present, providing the CTF in this context could lead to such a reconstitution.

To verify the results and to prove the interpretation the authors provide, the following control experiments would be necessary:

- a) A rescue with a full-length Gpr126 construct specifically in this context.
- b) A rescue of the CTF in the st47 strain.

6. Independently of whether the postulated CTF function is based solely on the CTF or on a combination of NTF/CTF, it remains elusive of which nature this function is. Obviously, the activation of a G protein cascade is the most likely CTF function. However, the authors show no such thing. Gpr126 signaling has been characterized quite well in zebrafish, such as receptor activation by a tethered agonist-derived peptide (Liebscher et al. 2014). It would be interesting to see whether a specific activation of the aGPCR by such a peptide rescues the trabeculation phenotype described on p. 8/9. The results of such an experiment will prove the involvement of the receptor CTF and shed light on the mechanism behind it.

7. p. 13/14: How is the NTF in Tg(myI7:gpr126 NTF-p2a-tdTomato) designed? Is it the same as the construct in st49 or is it truly the NTF (stopping at the cleavage site within the GPS)?

8. p. 14/15: The modulation of Notch signaling by Gpr126 is interesting. However, this data is very preliminary and highly speculative. At this point, it is definitely not solid enough to claim that Gpr126 regulates Notch activity.

The authors find that abrogating almost the entire receptor does not have a function on Notch activity. St49 leads to enhanced Notch activity, but overexpression of the NTF does not. From this, the authors conclude that the CTF inhibits Notch activity. However, several other scenarios are equally possible: Firstly, it can mean that the st49 construct conveys a dominant-negative effect (here, it is essential to know whether Tg(myI7:gpr126 NTF-p2a-tdTomato) carries the same construct as st49 – see my comment 7 above). Secondly, a purely compensatory mechanism is conceivable. The authors need to put some more effort into obtaining more and solid data on whether and how Gpr126 has an effect on the Notch pathway. The following controls might help:

- a) A CTF expression in the st47 strain.
- b) An NTF expression in the st47 strain.
- c) A CTF expression in the st49 strain.

8. p. 14/15: A highly puzzling issue is that if the CTF inhibits Notch activity as the authors postulate, one would expect that st47, in which the CTF is also missing, likewise leads to an increased Notch activity. This is, however, not the case and needs further clarification. I am wondering what the authors' explanation for this. Also, additional essential questions need to be addressed such as: Does the expression of the NTF in the st49 strain alter the expression of the Notch receptor/ligand? Altered expression levels of Notch pathway components could help explain changed Notch activity.

10. p. 15: Again: How is the NTF in Tg(myI7:gpr126 NTF-p2a-tdTomato) designed? If it does not mimic st49, the data obtained with the construct cannot be compared to the ones on the st49 (Fig. 7A, B). If it does mimic st49, a potential dominant-negative effect of the construct cannot be excluded. Thus, in order to gain the most conclusive set of data, both constructs might need testing.

11. p. 14/15: Simply showing an overexpression of the NTF in a wild-type background does not address the question which parts of the aGPCR are required for a potential function in the control of Notch activity. Further, it might help to also do the reciprocal experiment and e.g. express the CTF to see if a more pronounced downregulation effect occurs.

Minor comments:

1. Fig 1C: Information on statistics is missing: How many samples?
2. There are some typos in the manuscript, e.g. p. 7 l. 147 "amino acid". Please check spelling and grammar carefully.

Reviewer #2

(Remarks to the Author)

In this manuscript, Srivastava et al. studied the functions of adhesion G protein-coupled receptor 126 (Gpr126) in zebrafish heart trabeculation by analyzing two truncation mutants stl47 and st49. They found that maternal zygotically stl47 mutant exhibited hypotrabeulation phenotype with randomly distributed N-cadherin at membranes of the compact muscle layer. The st49 mutant lacking the C-terminal GPS motif exhibited a multilayered ventricular wall with normal N-cadherin distribution and elevated Notch signaling. They concluded that the N-terminal fragment (NTF) of Gpr126 maintains cell-cell adhesion and compact wall integrity in the myocardium, while the C-terminal fragment (CTF) is essential to establish the trabecular identity of cardiomyocytes. The finding here would provide novel insights into heart development and clarify the function of Gpr126 in myocardium trabeculation. However, as noted below, the conclusions might be a bit premature and require further analysis or clarification. The manuscript is well written and easy to follow.

1. The authors used qPCR to show the relative expressions of *gpr126* in CMs and endocardium, which is less optimal. Demonstrating the protein localization, either by an antibody staining or by tagging the gene, would be useful for mechanistic interpretations (e.g., a strategy used in a co-author's previous paper: <https://doi.org/10.1242/dev.171207>). It is understandable that this request is likely beyond the scope of a revision. However, *in situ* hybridization of *gpr126* showing the expression locations during trabeculation would be helpful.
2. The authors only showed the myocardium in all figures. Since *gpr126* is considered an endocardial-enriched gene in the heart, it seems necessary to assess any endocardial defects and endocardial cell-cardiomyocyte interactions in the mutants. Otherwise, the mechanism described in the current version seems to lack a significant component.
3. Figure 1B. It is interesting that 36.67% of *stl47* MZ mutant larvae and 26.67% of *st49* MZ mutant larvae showed normal trabeculation. Could the authors speculate the reason? How do they define "normal trabeculation"?
4. The authors showed that mosaic myocardial-specific NTF overexpression induced local cardiomyocyte multilayering and concluded that NTF has a function in maintaining cell-cell adhesion in the compact layer cardiomyocytes. However, *gpr126* is considered an endocardial gene (Figure 3A). It is confusing how an endocardium secreted protein mediated cell adhesions within the myocardium. At least the outer layer of the multilayered myocardium seems to have no contact with the endocardium. Defining the expression domain of *gpr126* (as noted above) seems necessary to clarify this. Also, an endocardial-specific overexpression strategy should be considered.
5. Figure 7. The authors concluded that CTF signaling is required to inhibit myocardial Notch activity to allow cardiomyocytes to attain trabecular identity. However, the MZ *stl47* mutant also lacks the CTF but showed normal myocardial Notch activity. Please explain.
6. Figure S5. It is labeled as *fli1a:gpr126* NTF-p2a-tdTomato in the figure but *myl7:gpr126* NTF-p2a-tdTomato in the legend and main text. Please clarify. I assume it is *myl7*. The myocardium looks quite normal compared to the mosaic myocardial-specific NTF overexpression phenotype in Figure S4 (*myl7:gpr126* NTF-p2a-tdTomato). Please clarify what makes the difference.
7. Can the authors explain or speculate why zygotic *stl47* mutant but not *st49* mutant exhibited maternal effect?
8. Control group is missing in the experiment shown in Figure S5
9. Figure 4A and B lack quantification results.
10. Page 8, lines 151-153. "The analysis of zygotic *gpr126stl47/stl47* mutant hearts revealed no obvious gross morphological defects compared to WT siblings suggesting that *Gpr126* is not required for heart development in zebrafish (Figure S2)." This statement is not accurate. The result only suggests that a zygotic mutant did not show a heart development defect. One cannot conclude that *Gpr126* is not required for heart development.
11. *st49* and NTF Δ GPS are used to represent the same mutant. It would increase the readability by using one label across the manuscript. Similarly, both *gpr126stl47* and *gpr126stl47/stl47* were used to denote the homozygous mutant in the figures (the same for *st49*). Please be consistent.
12. Page 6, line 103. Please include citations for these 2 published mutant alleles.

Reviewer #3

(Remarks to the Author)

This is an interesting study that aims to clarify a long-standing controversy in the literature concerning the function of *Adgrg6/Gpr126* in the zebrafish heart. *Adgrg6/Gpr126* is a clinically important adhesion class GPCR (aGPCR) that is associated with a number of different disease states in humans, and so a better understanding of genetic variants in the zebrafish model is important and likely to be of general interest.

Several studies in mice have revealed a function for *Gpr126* in cardiac trabeculation. However, the murine cardiac defects did not appear to be recapitulated in the various zebrafish mutant alleles available, raising the possibility that a function for *Gpr126* in the heart is not conserved. Nevertheless, many of the available zebrafish alleles are hypomorphic and/or predict expression of truncated protein variants, which has been offered as an explanation for the lack of cardiac defects. Other alleles have not yet been examined for a cardiac phenotype. A morpholino study in zebrafish (Patra et al., ref. 13) supported a role for the N-terminal fragment (NTF) of *Gpr126* in cardiac trabeculation defects; as discussed in the current manuscript, genetic compensation could provide an explanation for the discrepancy between the morphant and mutant phenotypes. The Engel group have also already published a study linking *gpr126* regulation to cardiac trabeculation (Musa et al., 2019), which is not cited in the current manuscript. The role for *Gpr126* in the zebrafish heart therefore remains unclear.

The current manuscript re-visits this controversy through analysis of the zebrafish heart phenotype in two of the most severe truncating alleles available, using maternal-zygotic (MZ) as well as zygotic-only (Z-only) mutants to examine the phenotype. The authors clearly demonstrate that the MZ *stl47* and MZ and Z-only *st49* homozygous mutant zebrafish have defects in trabeculation of the endocardium, each showing variable penetrance and expressivity, and a range of different phenotypes (multilayering, hypotrabeulation or absence of trabeculation). Other data are promising but more preliminary, indicating

possible links to blood flow, *ErbB2* and Notch signalling.

The manuscript contradicts some previous findings (some of which are from the same group, or in papers with authors in common with the current manuscript). As the authors state on line 376, the work fails to corroborate their previous hypothesis that the NTFdelGPS is sufficient for cardiac trabeculation (line 376), instead now suggesting that the CTF plays a critical role. They also suggest that their data contradict the recent published interpretation that the heart phenotype in mouse results from a placental defect (Torregrosa-Carrion et al., 2021, ref. 15). These contradictions are not easy to follow in the manuscript. There are a number of concerns with presentation and interpretation of the data, and of citation and discussion of the previous literature, that need to be addressed.

Major comments

1. One clear but surprising new finding is that the st49 allele, which the authors demonstrate is likely to express the NTFdelGPS fragment, is the only *gpr126* allele described so far for which there is a cardiac trabeculation phenotype in zygotic (Z-only) mutants. This is contrary to the authors' previous published analysis of *gpr126* st49 mutants (Patra et al., 2013: '*gpr126*st49 mutants exhibited no pericardial edema or other cardiac abnormalities') and their demonstration that the NTFdelGPS alone was sufficient to rescue a trabeculation defect in *gpr126* morphants (Patra et al., 2013) (ref. 13). However, the interesting result of the Z-only phenotype in st49 is hidden in the manuscript—it is only presented as supplementary data (Fig. S2), and the title of the figure (Zygotic *gpr126*stl47 hearts do not exhibit trabeculation defects) completely overlooks the result. This figure should be moved to form a top row in the main Fig. 2, allowing a direct comparison of the Z-only and MZ phenotypes for each allele. The abstract should also mention this result.

The phenotypes shown are clearly incompletely penetrant and variably expressive for both the MZ and Z-only mutants for st49, and for MZ stl47 mutants, but this is not explicitly stated anywhere. Moreover, the distribution of cardiac phenotypes in the Z-only st49 mutants appears identical to that in MZ st49 mutants; given the low numbers of fish sampled (n=15 Z-only (Fig. S2) and n=30 MZ (Fig. 2)), there is no evidence for any significant difference in strength or nature of the phenotype between the Z-only and MZ fish. This finding should be presented more clearly, as suggested above. If the authors wish to document a difference in the distribution of cardiac phenotypes between Z-only st49 and MZ st49 fish, they will need to examine more animals and perform an appropriate statistical test on the data to confirm this.

2. One interpretation of the data is that the NTFdelGPS has a dominant effect to give rise to the unique cardiac phenotype (multilayering) seen in some embryos for the st49 allele. Do the heterozygotes for this allele show any cardiac phenotype?

3. The sections linking *Gpr126* to blood flow, and the *ErbB2* and Notch signalling pathways, are less well worked up and need additional experimental confirmation.

a. Figure 4. The description of the *ErbB2* inhibitor and blood flow experiments shown in this figure is confusing. The genetic backgrounds tested (over-expressed *gpr126*, or the MZ st49 mutant, Fig. 4) both already have blocked trabeculation, so how can the authors conclude that the *ErbB2* inhibitor blocks trabeculation in these backgrounds? Instead, both the inhibitor and lack of blood flow appear to rescue the multilayering phenotype in each background. This needs further discussion and explanation. Quantification of the phenotypes should also be provided in the figure legend.

b. The authors document an increase in Notch signalling in the MZ st49 mutants in Fig. 7, and show that the CTF can rescue the multilayering phenotype in this allele in Fig. 3. Importantly, can over-expression of CTF also restore normal levels of Notch signalling in MZ st49? This is an essential control for the interpretation of the mechanism here.

4. The authors also speculate on lines 421–422 that rescue by the CTF is likely to be through canonical G-protein-coupled signalling. This would be simple to test by raising cAMP with forskolin, a treatment that should rescue the cardiac phenotype in the st49 allele. This experiment is essential to support the argument.

5. Over-expression of the NTF in a wild-type background results in multilayering (Fig. S4). Does over-expression of the NTF in MZ stl47 mutants result in an st49-like multilayering phenotype? This would help to clarify the role of the NTF.

6. This also needs some more discussion in the manuscript: why does the maternal transcript fail to compensate for the loss of zygotic function for the st49 allele specifically? And why does the stl47 allele (which also lacks the CTF) not show any Z-only phenotype? Can the MZ stl47 phenotype also be rescued with forskolin?

7. The imaging in Figure 6 is not entirely clear. A counterstain for cell membranes is necessary to show the position of cells, especially as the epithelium is a curved surface. In addition, the chosen 'representative' example for MZ st49 does not have the multilayering phenotype. It is essential to show what happens to cadherin expression in those MZ st49 mutant hearts that have a multilayering phenotype.

Minor comments

8. The order of the alleles presented in Figure 7 (in both the image panels and the graph) should be swapped round to match that in the rest of the manuscript, i.e. stl47 first, and then st49.

9. The authors have undertaken a qPCR analysis of FACS-sorted cardiomyocytes (Fig. 3A) to confirm that *Gpr126* mRNA is enriched in the endocardium, which is helpful. However, the Introduction should state that the gene is known to be expressed in the zebrafish heart (since a lack of expression in this organ in the zebrafish could be a simple explanation for any species-specific differences in function), citing relevant studies. The authors could refer to Geng et al. (2013) for expression at 24–48 hpf, or their own study for expression at 48 hpf (Patra et al., 2013 – although confusingly, note that the overall expression pattern shown in this paper does not match that shown in Monk et al., 2009 or Geng et al., 2013).

10. In all relevant main and supplementary figures (including both images and diagrams), the authors should replace red with magenta, to make the images accessible to those with dichromatic colour vision. For the triple label in Figure 1, choose a suitable colour combination (e.g. magenta, green, white)

11. There are some omissions or inconsistencies in discussion of the previous literature, including papers authored by some of the authors of the current manuscript, which make it difficult to follow the arguments.

a. D'Amato et al. (2016) (ref. 46) is cited in the context of myocardial patterning, but the work in this paper to elucidate the relationship between Gpr126 and Notch signalling in the mouse and zebrafish is not mentioned or discussed anywhere. D'Amato et al. (2016) used several lines of evidence to demonstrate that cardiac expression of Gpr126 is a downstream target of Notch signalling, including documenting a loss of gpr126 expression in the hearts of zebrafish antimorphic *mib* mutants (loss-of-function for Notch signalling). How do these results fit with the model that the authors propose for the zebrafish, with the enhanced myocardial Notch signalling in MZ Gpr126-st49 mutants?

b. The authors do not cite or discuss previous work from their own group (Musa et al., 2019), in which they specifically linked miR-27a/b regulation of gpr126 expression to trabeculation in the zebrafish heart, and compared phenotypes to that of the gpr126 morphant. This paper appears to provide important background information that supports the current study. Why is this study not cited?

12. A schematic diagram summarising the authors' current model and thinking for how the NTF is working with the CTF, linking to Erbb2 and Notch signalling and cell polarisation, would be very helpful.

13. Introduction, Line 56 – do the authors mean to say 'is expressed in' rather than 'is essential for' here? It seems odd to say that the gene is essential for development of the heart but then to go on to say this is controversial and needs testing.

14. Much of the information presented in Supplementary Fig. S1 (panels A–C, i.e. both the sequence and the ear phenotype for both alleles) has been published by other groups elsewhere. It is good to see that the authors have been able to confirm these findings in their own hands, but it must be stated that they are confirming previous published findings, citing Petersen et al. (ref 65) for the analysis of *stl47*, and Monk et al. (ref 35) for *st49*, rather than presenting these as new data.

15. Please note that the amino acid at the position of the indel in the *stl47* allele is not a glutamine (Q), as previously published by Petersen et al. (2015) – it is an asparagine (N) (as the supplementary figure in the current manuscript clearly shows) (see Table 1 in Baxendale et al., *genesis*, 2021). This should be corrected in the text on line 104.

Version 1:

Reviewer comments:

Reviewer #1

(Remarks to the Author)

The revised version of the manuscript by Srivastava et al. has improved remarkably compared to the initial one. I am very content with the additional data and arguments the authors present. They are convincing and I believe that the manuscript is a thoroughly studied piece now that is interesting for a broad readership.

However, I have one minor thing I am still a bit concerned about. The authors show an interesting link between GPR126 and Notch activity, which is clearly one of the most interesting aspects of the manuscript. However, there are still many open questions regarding the interplay of the receptor and Notch. Thus, I suggest using more careful and less prominent phrasing e.g. in title and abstract and on the other side a few more details on the open questions and the impact in the discussion.

Reviewer #2

(Remarks to the Author)

The authors have added significant new data and addressed most of my concerns in the revised manuscript. However, a few concerns need further clarification or experimental validation. The authors aimed to clarify a controversy in the function of gpr126 in the zebrafish heart. It is crucial to provide high-quality, mechanistic evidence, notably when the authors claimed some conclusions that contradict previous findings, including their own.

1. I'm afraid I have to disagree with the authors' claim that the probability of successfully generating a knock-in is very low. Numerous zebrafish knock-in alleles, including precise tagging alleles, have been published (e.g., PMIDs: 34495314, 32412410, 33681181, 35378361, 36476416, 32064511, 36878640). It is doable to generate precise tagging of Gpr126. It would be the authors' call whether it is necessary for the current study.

2. The authors claimed that NTF is required for the junctional localization of N-cadherin in the working model. However, N-cadherin is still localized to junctions in the *stl47* mutant (Fig. 4, S9a). The provided evidence only supports a correlation, and the impaired junctional localization might result from other defects in the compact wall integrity. Notably, images in Figures 4 and S9a are snapshots of representative samples. The phenotype is not clear to me. The WT images in Fig. S9a look very similar to the *stl47* images in Fig. 4, depending on which confocal sections were chosen. It is not apparent how the schematic for the *stl47* group was generated, which does not recapitulate patterns in the image. The *stl47* heart looks

smaller than WT in the confocal section image but not in the maximum projection image. More sophisticated presentation and quantification of the z-stack images would be necessary to support the claims.

3. The authors claimed that CTF is required for N-cadherin relocalization. Again, the provided evidence only supports a correlation. There is no direct evidence supporting that CTF regulates N-cadherin localization. It is also unclear whether the impaired N-cad localization is an outcome or cause of the trabecular defect.

Reviewer #3

(Remarks to the Author)

The authors have made a number of revisions to the original manuscript, including extra experiments to address some of the reviewers' comments. Some of the new findings have confirmed previous findings, whereas others have changed their interpretations. In general, the data are important and will be of interest to the scientific community, especially those working on Gpr126 in different contexts and model systems. However, unanswered questions remain, and the text and some of the interpretations and conclusions are sometimes unclear. Citation of other work is sometimes erratic. There are a number of places where changes could be made to help the reader follow the paper more easily.

Analysis of the transcripts and proteins produced

Line 131 and Fig. S1D - It is surprising that neither mutant transcript is subject to nonsense-mediated mRNA decay, and that transcript levels appear normal. Have the authors sequenced variants here? Could there be alternative splicing (possibly heart-specific) or transcription start sites that could compensate for the predicted non-functional transcripts? This might explain the significant proportion of embryos in which trabeculation is normal in both Z and MZ mutants for both alleles. Even small amounts of alternatively spliced zygotic mRNA could result in functional protein.

Analysis of the tagged proteins in Fig. 1 shows that there is quantifiable GFP fluorescence in cells expressing both mutant versions, although the Western blot is clean. Is the GFP fluorescence above untransfected background levels? If so, are small amounts of protein produced from these transfected constructs?

Is maternal transcript detectable in WT and MZ stl47 and st49 embryos? As the authors are proposing a role for maternal transcript or protein, an in situ should be shown from early (pre-ZGA) embryos to show whether there are detectable levels of maternal mRNA.

Heart phenotypes in Z and MZ stl47 embryos

It needs to be pointed out more explicitly that 36.67% of MZ stl47 mutants have WT trabeculation, and over 60% have some degree of trabeculation. This indicates that under some circumstances (possibly for some genetic backgrounds), Gpr126 is not required for heart development. This result is shown in Fig. 1 but is not described explicitly in the text (lines 189–192).

Could the range of phenotypes seen in MZstl47 be explained by an overall general delay? What is the gross morphology of these embryos? – this has not been shown anywhere.

Secretion of the NTFdelGPS

The authors report that the NTFdelGPS is detected in the cytoplasm, supported by immunostaining and Western blots of cell lysates (Fig. 1D). They say cytoplasmic expression was detected 'in a minority of cells'. Does this mean only a minority of cells were transfected with the construct, or, of those transfected, only a minority show cytoplasmic expression? The authors now also report secretion of the NTFdelGPS. This is very important given its apparent non-autonomous function on the myocardium, and also because the anti-HA fluorescence and cell lysate blot shown in Fig. 1 imply that it is stuck in the cytoplasm. However, the main figure still shows the apparent cytoplasmic expression, while analysis of the supernatant to show potential protein secretion, which the argument hinges on, is hidden in the supplementary material (Fig. S2C). What is the relative amount of cytoplasmic vs secreted protein? – the Western blot should show the lysate and supernatant bands from the same samples side by side in the main figure.

Possible dominant effect of the NTF

We do not believe that the authors have sufficiently ruled out a possible dominant effect for the N terminal fragment produced by the st49 allele, a point that was queried by us and another reviewer for the original version.

The authors state on Line 221 – 'no heart phenotypes were observed in heterozygous gpr126st49 mutants.' It is essential to show the data and quantification of the phenotypes here, as the authors have done for WTs, Z homozygotes and MZ homozygotes. Given the incomplete penetrance of the phenotype in the homozygotes (26.6% of Z and MZ st49 mutants show normal trabeculation), this also needs to be quantified for the heterozygotes. This may require further work and genotyping, but is essential to rule out any dominant effect of the NTF.

Notably, the experiment presented in the previous manuscript (v1, Fig. S4), in which mosaic overexpression of the N terminus in the WT myocardium appeared to induce local multilayering—by definition a dominant effect—has now been removed from the current version of the manuscript.

Link to Notch signalling

20% of MZ stl47 mutants show a multilayering phenotype (Fig. 2D). Does this population also show an increase in Notch signalling?

Model - early and late requirements

The authors state in their rebuttal that there is an early requirement for the NTF in N-cad localisation (a prerequisite for trabeculation), and a late requirement for the CTF in trabeculation. They say this difference in timing explains the difference in phenotype between the two alleles. It is stated several times in the MS and in the rebuttal that the maternal protein is unlikely to perform the later function. However, in the Z stl47 mutants, both early and late roles would have to be performed by the maternal transcript or protein, resulting in normal trabeculation. Action of the maternal transcript for such a late function(96h+) is really puzzling. Thus, the proposed early/late functions in the model do not fit with the data shown.

If the maternal protein can perform both early and late functions (as demonstrated by the Z-only stl47 normal phenotype), then why is it unable to do this in the presence of zygotically-produced NTFdelGPS in the Z st49 mutants? This is still unexplained, unless the NTF blocks the effects of the maternal protein in a dominant manner, or there is some heart-specific compensation mechanism in the stl47 allele.

The authors do acknowledge that there are difficulties in their interpretation and that further work is needed at Line 413, but this should be stated up-front more explicitly, and the model should either be revised or presented with much more caution.

Pharmacological rescue experiments

Paragraph starting at Line 255. It is really important to cite Petersen et al. (2015) here for pharmacological rescue of the stl47 and st49 alleles. These authors only used forskolin, and importantly, showed that this could not rescue radial sorting or myelination in stl47 mutants.

For use of IBMX, Geng et al. (2013) is cited, which is appropriate. However, it should be clarified here that the published use of IBMX was with other alleles, not stl47 or st49. In addition, the concentration used in this paper was 100 μ M, not 125 μ M. Finally, IBMX is not an AC activator - it is a phosphodiesterase inhibitor, raising cAMP levels independently of AC activity and thus an important way to confirm that the CTF works through cAMP.

It is important to get these details correct, as they are required to interpret the allele-specific and domain-specific effects for these phenotypes, which are not straightforward.

Model – autonomy and non-autonomy

The authors have still not explained how expression and activity of the CTF in the endocardium can lead to effects on the myocardium, i.e. how the receptor acts in a non-autonomous manner. In other tissues (ear and Schwann cells), the phenotypes affect the cells in which the receptor is expressed. In the heart this appears to be different, but no suitable explanation is provided.

In this context, it is really important to point out the cell types in which the experiments were done. For example, the over-expression of the NTFdelGPS (lines 215–217) is done in the myocardium, whereas production of the NTF from the st49 allele is expected to be in the endocardium.

Other comments and essential corrections

The sentence at Lines 52–53 is an exact copy of a sentence in the Abstract from Zaidman et al., PNAS USA 2020, and should be reworded.

The citations of references 18–20 supporting the information on Lines 66–70 are three primary papers covering pulmonary function, COPD and periodontitis. These references do not cover the other diseases that the authors refer to (scoliosis, intellectual disability, cancers). The authors should either cite all relevant primary papers to support their statements, or they should cite a suitable review that covers all this information with comprehensive referencing.

Lines 88–89 ‘newly created zebrafish mutants’ (reference 17, Torregrosa-Carrion et al., 2021). It is essential to give full details of these alleles; one (bns342) is a promoter-less allele with undetectable transcript levels and normal cardiac trabeculation, and so it is important to point this out. In addition, this study did not rely on ‘gross morphological analyses’ - it presents confocal images (ref. 17, Fig. 3) at the same level of detail as the current manuscript. This information should be acknowledged, described and cited correctly.

Line 74 – depletion, not deletion

Lines 121–127 Introduction to the alleles used. The authors now cite the original papers at the start of this section, and have corrected the amino acid codon affected by the lesion, but still go on to present the data in the supplementary figures identifying the molecular lesions as their own. It would be best to repeat the citations throughout the paragraph to make it clear that this figure is confirming previously published data, or to explain up front that these alleles have previously been characterised and that the figure is confirming these published data.

Line 162 As the experiments here are done in cells, and not in embryos, the wording should be changed here to say ‘These data suggest that gpr126st49 mutants express and secrete the NTFdelGPS.’ (Not ‘indicate’.)

Line 170–183 – The title of this section is about stl47. However, the important result of the multilayering in st49 is also introduced here. It would help to have one section about stl47 and then another about st49.

Line 200 – it is stated that there are no obvious defects in the endocardium. However, it is not folded as normal, and this could be interpreted as a defect.

Line 209 – ‘experiments previously suggested that the NTFdelGPS is sufficient for trabeculation and heart development (ref. 21). It would help to say here something like ‘However, we are now revising that view’.

Line 212 – this should be Z, not MZ

Figure 4 - N-Cadherin data

These images are still difficult to interpret. It would be good to show individual channels in inverted greyscale to show as much detail as possible.

ErbB2 section (Lines 267–290)

All data for this section are presented in the supplementary material. This is still too preliminary and it is recommended that this section is removed from the manuscript.

Version 2:

Reviewer comments:

Reviewer #2

(Remarks to the Author)

The authors have addressed the concerns raised in the previous review. I appreciate that they have toned down the language in several conclusions, adopting a more cautious and balanced approach in interpreting the phenotypes and addressing the discrepancy with previous work. These revisions have improved the clarity and rigor of the manuscript.

Reviewer #3

(Remarks to the Author)

The authors have added several new supplementary figures to the MS, including new data, and have addressed many of the comments satisfactorily. These include some interesting new findings relating to the prevalence of the exon-6-skipped transcript isoform in the heart. As stated previously, the study is important and will be of value to the research community.

However, there are still a number of instances where the data and/or interpretations appear inconsistent or incomplete, and these may affect the way that the overall mechanistic model is presented. These issues are detailed below.

1. Other alleles

It is good that the authors have added details of other mutant alleles when discussing previous work. However, the Torregrosa-Carrión et al., 2021 study is still misquoted, as it was not 'based on gross morphological analyses' in the added text at lines 99–100 - this statement needs to be removed. The Torregrosa-Carrión paper presented detailed confocal imaging of trabeculae in transgenic lines at 96 hpf, together with quantitative analysis of transcript levels at 48 hpf (Torregrosa-Carrión et al., 2021, Fig. 3). The discrepancy with the current study can be explained possibly because this study only examined Z-only mutants, although even this is problematical, because the authors demonstrated a complete loss of transcript in zygotic bns342/bns342 mutants at 48 hpf (T-C et al., 2021, Fig. 3B), and concluded that gpr126 is not required for trabeculation in the zebrafish heart. (It will obviously be interesting to test whether there is a heart phenotype in MZ bns342/bns342 mutants.)

2. Figure S3b

As requested, the authors have added Fig. S3b to show the whole-embryo appearance for stl47 and st49 alleles. However, their description in the text and legend is at odds with the images shown. Gross abnormalities are evident for the stl47 allele in the figure: shorter body length, small head and eye, underdeveloped jaw, and pericardial oedema. If these phenotypes are representative, they need to be described. If not representative, another embryo should be shown. Note that incompletely penetrant pericardial oedema was previously reported for zygotic stl47/stl47 mutants (Fig. S1c, Petersen et al., 2015), matching the phenotype of the stl47 embryo in Fig. S3b. This published observation should be acknowledged and cited somewhere.

An additional issue with Fig. S3b is that the embryos look completely different to those shown in Figs. S1c and S10a, also labelled as 96 hpf. The puffy ears are not nearly as evident for either genotype, and thus it is not convincing that these fish are actually mutant for gpr126. We suggest that they are likely to be 72 hpf or younger – please check.

3. Figure S6

The authors state that they have previously shown that the NTF binds to the myocardium in mice as a justification for expressing it in the myocardium in zebrafish. However, they also confirm that Gpr126 is normally expressed in the endocardium and that the NTF is secreted. It would have been more relevant to express the NTF in endocardium, to mimic the situation in the mutants more closely. Expressing it in the myocardium adds an additional level of complexity to the interpretation. Do you see similar phenotypes with endocardial (krt4-driven) or endothelial (fli1-driven) expression of the NTF?

Lines 260–262 The wording here is difficult to follow. Suggest changing to:

"Therefore, we analyzed myocardial-specific (myl7-driven) mosaic expression of NTFΔGPS in WT larvae as well as stable expression in MZ *gpr126stl47* mutants,"

If the data from the stable line are available for MZ mutants, it is important to show the effects of expression of this line in sibling embryos (Z mutants and WTs), as these will be more relevant controls for this experiment than the mosaic overexpression experiment.

Fig. S6b. In Fig. 2d, it is stated that 20% (n=30) of MZstl47 hearts have a multilayered ventricular wall. In Fig. S6b, this proportion has now halved to 10% (n=8), and this value is used for comparison with the over-expression result (40%, n=10). However, is not clear how the percentage values for this graph in Fig. S6b were generated. Why are the data not presented like the stacked bars in Fig. 2? Given the low numbers for the Fig. S6b experiment (did the data pass a normality test?), and the Fig. 2d estimate of 20% (more accurate as based on higher numbers), obtaining 4/10 embryos with multilayering by chance is very possible. Thus, as they stand, the results shown in Fig. S6b are not convincing.

4. Figures 4, S11, S12

The description of the N-cadherin phenotype in MZstl47 mutants is now clearer, although it is still quite hard to see the differences between wild-type and mutant junctional arrangements in the images, and this was also pointed out by another reviewer. The additional images in Fig. S12 are helpful, although we still think that enlargements of selected regions of the MIPs would be useful. However, if the original images are retained, it would be helpful to include some labelling on the MIP images to highlight the main important features.

No additional images of MZst49 were provided, and it is stated that the MZst49 hearts have a wild-type arrangement of N-cadherin (line 377). To us, the MIP image of MZst49 (Fig. 4, lower right panel) has features that look more similar to the MZstl47 phenotype, especially towards the bottom and right hand side of the ventricle. If a phenotype is present, this would change the interpretation here (line 381), instead indicating that the NTF is not sufficient for the correct localisation of N-cadherin in the compact layer. Can the authors comment and add some additional images of the MZst49 hearts (it is stated that 12 samples were imaged) to Fig. S12, as for MZstl47?

5. Model (Figure 7)

The authors have revised their interpretation of the data, but the summary model in Fig. 7 remains unchanged, so it is hard to understand exactly what is new. It must be stated in the figure legend (as well as the text) that this is just a model for the role of *Gpr126*, as it incorporates a number of assumptions. Some basic details are unknown at the moment, such as any polarised location of the receptor on the endocardial cell plasma membrane. It seems unlikely that all NTF would be secreted and degraded, when the CTF persists – again, these details have not been tested.

What do the dotted black arrows represent? Is this a second unidentified signal? This relates to our earlier comment that the autonomy and/or non-autonomy of action should be clarified (terms that the authors have avoided, but seem to be relevant for the interpretation of the data). The authors propose a non-autonomous role for both the NTF and the CTF, expressed in endocardial cells, on the myocardium. For the CTF, such a non-autonomous role is unexpected and would be likely to need a second signal that can reach the myocardium. The authors do mention possible ways in which the endocardial cells could contact the myocardium across the cardiac jelly, but this is not depicted in the diagram.

The diagram also does not help to explain what the authors mean by a 'balance' between the NTF and CTF, which now seems to be a key feature of their model, but is not mentioned in the figure legend.

In addition, the Notch signalling in Fig. 7 is shown in the myocardium, whereas in the Introduction and Discussion, it is stated that endocardial Notch signalling is important for trabeculation (line 108, 558–9). The authors do cite work on Notch expression in cardiomyocytes at line 417, but this needs a more detailed description, including known timing of events. According to the imaging with Tg(TP1:VenusPest) in Fig. 6, levels of Notch signalling in the myocardium in WTs are normally low – the data here do not match the expression patterns sketched in Fig. 7. Some more explanation here, together with timing/embryo stages on the diagram, would be helpful.

6. Other minor (but important) comments/corrections:

- Line 70 'the membrane' should be 'the plasma membrane'

- Lines 103–119 – It would help to outline the normal timing of cardiac trabeculation events and Notch signalling in the different cell types in this paragraph, so that the reader can relate them to the stages used in the experimental work.

- Line 138 – predicting, not resulting

- Line 193 – it says 98 amino acids here, but 96 earlier in the MS and on the figure

- Lines 208–209 – a reference is needed to support this statement.

- Line 219 et seq. – it would help to state the corresponding n values here in the text with the % values, as well as in the figure legend.

- Line 231–232 ‘Gpr126 is required for cardiac chamber development in zebrafish’ - As 1/3 of MZ mutants of both genotypes have normal trabeculation, this statement should be qualified here to indicate that the phenotype shows incomplete penetrance and variable expressivity.
- Line 354-356: Suggest rewording here to highlight the rescue aspects of this experiment, as well as describing the data in the order that they appear in the figure, e.g. "Analysis of the heart revealed that both treatments could partially rescue the multilayering phenotype in MZ gpr126st49 mutants (Fig. S10b, upper panels). However, both treatments also blocked trabeculation, both in MZ gpr126st49 mutant hearts and in WT hearts (Fig. S10b, lower panels)."
- Line 368 – side, rather than site
- Line 378 – Fig. S11a, not Fig. 11a
- Line 553–4 – give authors and year for the bioRxiv paper, and list full details and link in the Bibliography, as for other references.
- Line 574 – births
- Fig. 1a, Fig. S7a The schematic diagram in Fig. 1a is not drawn to scale, and the term 'partial GAIN' is not explained anywhere. The similar diagram of the protein structure in Fig. S7a appears to be much more accurate in terms of scale - why not use this as a template to generate the diagram of the tagged protein in Fig. 1a? It would help the reader if both diagrams matched each other. Note, however, there are a couple of typos in Fig. S7a - the allele name is st49, not stl49, and only 6 TM domains are depicted. The st49 mutation is also shown at a different position relative to the GPS motif than that in Fig. 1a. This should be positioned as accurately as possible (and consistently) in both diagrams.
- Fig. 1b, e – Position of scale bars does not match between figure and legend.
- Expression of constructs in ARPE-19 cells and BFA treatment. Fig. 1f (stable expression) and Fig. S2c (transient transfection) – It is tricky to compare these figures as the lanes are presented in different orders. The position of bands relative to the size markers is inconsistent (WT bands are at or above the 115kDa marker in Fig. 1f, but well below it in Fig. S2c). Also, loading controls (e.g. of a constitutively secreted protein) are needed to confirm the reduction seen with BFA (for both figures).
- Fig. 3 – order of panels (a, d, b, c, e, f) is very difficult to follow.
- Fig. 3b and Line 326 - Is the P2Y12 N-terminus peptide the domain labelled P2A in the figure? There is no explanation of this in the figure legend.
- Fig. 4, 5a schematics – it would help to include the endocardial layer in these diagrams.
- Fig. 5A schematics, Fig. S6c – These figures will be difficult to interpret for anyone with a red-green colour vision deficiency.
- Fig. S6 – the data are not presented in the order they are described in the text. It would be easiest to follow if the mosaic WT data were presented first in the figure, following the order of the description in the text.
- Fig. S6b legend - should say ‘Quantification of a’ (not b).
- Fig. 6 – title and legend to figure: Notch should have an upper case N
- Fig. S10 – the title for this figure says that ‘Forskolin and IBMX treatment impairs trabeculation in ... MZ gpr126st49 hearts’. But trabeculation in these mutants is impaired anyway (Fig. 2a), so the title here is not an accurate reflection of the findings of the experiment. The text (line 355) instead says that the treatment rescues multilayering. Please clarify.
- Fig. S10a - there is some developmental delay in the treated embryos in this figure (head shape, jaw development and remaining yolk volume). This could account for some of the differences seen in the ears and hearts as well.
- Fig. S13a – as membranes are not imaged here, it is difficult to tell whether or not there is any multilayering phenotype in the WT hearts. As mosaic expression does result in multilayering (Fig. S6c), it is important to show whether there is any multilayering present, in addition to the lack of Notch activation.

Version 3:

Reviewer comments:

Reviewer #3

(Remarks to the Author)

The authors have addressed all the comments satisfactorily and have made a number of changes that have improved the clarity of the manuscript. We are now happy to recommend this interesting study for publication.

Point-by-point response to reviewer comments

We sincerely thank the reviewers for thoroughly examining our manuscript and providing helpful comments/suggestions to improve our manuscript. In the revised manuscript we have addressed all of the raised issues including adding a large amount of new experimental *in vitro* and *in vivo* data which further substantiate our conclusions and led to a significantly improved manuscript. Please find below a detailed response to the individual questions.

Reviewer #1:

The study by Srivastava et al. is very interesting and points towards exciting functions of the Adhesion GPCR ADGRG6/Gpr126, which also can give some indication on how Adhesion GPCRs in general function. However, many of the experiments are very preliminary and the interpretation sometimes is a little doubtful. I have several comments, especially regarding the use of the strains and constructs, which in my opinion need careful re-evaluation. In my opinion, the study in its current form is not complete and requires substantial more work to demonstrate many of the claims made, such as the CTF function and the regulation of Notch activity.

We thank the reviewer for the overall positive view on the importance of our study and the detailed comments and suggestions that allowed us to address the raised concerns resulting in a significantly improved manuscript.

Major concerns:

1. st49 mutant: I appreciate that the mutant has been characterized previously. However, both, st49 and st47 disrupt essential domains of the receptor, the CUB and the GAIN, respectively, which carries the risk of (partially) misfolded proteins. This probably does not matter for st47 as there is not much protein left and I agree that it can be considered a complete loss of function. However, it does matter for st49. The authors try to address this issue by expressing the construct in cells. However, as Fig. 1B shows, the construct accumulates in the cytoplasm. The authors state that this is due to the missing 7TM. I would assume that a missing 7TM leads to a liberated protein, which gets secreted rather than retained, especially as the signal peptide is still present. This is at least what we see for other GPCRs. The authors also postulate that this occurs in the st49 strain (p. 15, l. 326). I am highly worried what really happens to the protein. It might lead to dominant-negative effects in the fish and thus, to phenotypes. I would

like to see the experiments (or at least part of them) conducted in a fish strain carrying a “clean” version of the N terminus – liberated or even tethered to the membrane. I realize that this some effort, but it would generate much more solid data. With CRISPR-Cas this should be doable.

We agree with the reviewer that it is important to ensure that the *gpr126^{st49}* mutant expresses an NTF^{ΔGPS} that is not misfolded but is functional and secreted. We furthermore thank the reviewer for indicating that we have not properly presented our data resulting in the wrong assumption that the NTF^{ΔGPS} is accumulated in the cytoplasm. The corresponding image represents a minority of cells and was used to prove that the mutant forms are expressed. In our opinion the best proof that NTF^{ΔGPS} represents a functional NTF is to rescue *gpr126^{st49}* mutant phenotypes by expressing endocardial CTF. To address this issue, we have changed the text and added a large number of experiments:

1) We provide overview images from cell cultures post transfection to show that NTF^{ΔGPS} is not accumulated in the cytoplasm (Fig. S2a) and clarify on page 7, line 151: “In contrast, the mutant Gpr126 forms failed to localize to the membrane (Fig. 1b) and in a minority of cells cytoplasmic expression was detected (Fig. S2a, b). Notably, previously we have shown that ectopic expression of NTF^{ΔGPS} in Hela as well as H9c2 cells results in a stable, secreted protein.”

2) To validate that NTF^{ΔGPS} is secreted, we performed western blot analyses of cell culture medium in the absence and presence of brefeldin A (Fig. S2c). The results are stated on page 7, line 157: “To confirm that a stable, secreted NTF^{ΔGPS} is produced by *gpr126^{st49}* cDNA, the culture medium from ARPE-19 cells transfected with WT or *st49* constructs were analyzed by western blot (Fig. S2c). These data reveal that the culture medium contains NTF/NTF^{ΔGPS}. Moreover, treating the cells with brefeldin A, which blocks the trans-Golgi secretory pathway, significantly decreased the amount of NTF/NTF^{ΔGPS} in the medium. These data indicate that *gpr126^{st49}* zebrafish mutants express and secrete NTF^{ΔGPS}, which previously has been reported to be sufficient for heart development.”

3) We state on page 7, line 155: “Thus, the failure to localize to the membrane and the reduced number of HA-positive cells could be explained with the loss of the 7TM domain in both truncated forms.”

4) Importantly, if the *gpr126^{st49}* mutation would result in the expression of a “(partially) misfolded protein” possibly exhibiting dominant effects, it should not be possible to rescue this mutant phenotype by overexpression of the CTF. Therefore, we had provided in the original version data regarding these rescue experiments. To further strengthen these data, we have now added additional evidence for *gpr126* being expressed in the endocardium in zebrafish heart (RNAscope, Fig. S5, page 10, line 229) and that overexpressed CTF localizes in the transgenic animals in endothelial and endocardial cells at the membrane (Fig. 3c, end of page 10). Furthermore, we provide new data showing that overexpression of CTF in the MZ *gpr126^{st49}* mutants, which restores normal trabeculation, also restores myocardial Notch activity (Fig. 6e, f, page 16, line 362).

5) Furthermore, if there were dominant negative effects of the *st49*-encoded protein, we would see them in the heterozygotes. We did not observe any multilayering phenotypes in the heterozygous *gpr126^{st49}* larvae consistent to the absence of myelin phenotypes in heterozygous *gpr126^{st49}* in Petersen et al. 2015 (PMID: 25695270) (page 10, line 219).

Collectively, these data show that NTF^{ΔGPS} represents a functional NTF.

We would also like to clarify that the fact that *gpr126^{st49}* mutants, lacking CTF, show enhanced Notch activity but MZ *gpr126^{st47}* mutants, lacking NTF and CTF, does not, is not due to a

dominant effect of NTF^{ΔGPS} but due to the fact that NTF is required for compact wall integrity before trabeculation is initiated. To clarify this point we added on page 15, line 345:” It has been demonstrated that Notch signaling is activated during trabeculation upon cardiomyocyte delamination in a subset of compact wall cardiomyocytes to inhibit their delamination³⁸⁻⁴⁰. As MZ *gpr126*^{stl47} mutants are characterized by hypotrabeulation, no change in myocardial Notch activity is expected.”

Comment on “With CRISPR-Cas this should be doable”: We would like to note that the suggested experiments would require precise gene editing which requires that cells utilize homologous DNA repair (HDR). While HDR efficiently works in several model organisms including mouse, non-homologous end joining (NHEJ) is the preferred mechanism of repair in zebrafish. This is the reason why it is extremely difficult to generate zebrafish with precise mutations (PMID: 27795562).

2. Fig. 1B: It looks like there is less membrane localization of Gpr126 st49, but increased cytoplasmic localization. The authors state that this is due to the missing 7TM. As mention in comment 1, I would assume that the missing 7TM leads to a liberated protein that gets secreted rather than retained as the signal peptide is still present. I would assume other reasons for the cytoplasmic expression are more likely such as a partially misfolded protein.

This issue has been addressed in detail above (see response to Point 1).

3. p. 9: Why does the st47 mutant not show a zygotic phenotype, but the st49 does? Maybe I am misunderstanding this, but if this is the case, it is a really interesting finding. It might either point towards a separate function of the NTF that is only visible when the CTF is absent, or it can be a dominant-negative effect of the st49 (see comment 1). The authors need to discriminate between the different options and explain this matter.

We agree with the reviewer that this is an interesting finding.

Option 1: “dominant-negative effect”

This has been excluded as explained under Point 1.

Option 2: “separate function of the NTF”

Our data show that the NTF is required earlier in development than CTF. NTF is required early in development for compact wall formation by regulating N-cadherin localization, a prerequisite for proper trabeculation. In contrast, the CTF is required later for the subsequent control of Notch activity and attainment of trabecular identity. As maternal mRNA is only transiently present until early embryonic stages, these findings provide a likely explanation for why the Z *gpr126*^{stl47} mutant shows no phenotype, but the Z *gpr126*^{stl49} does.

Notably, our data are in agreement with a previous report in the peripheral nervous system, where it has been shown that MZ *gpr126*^{stl47} mutants exhibited a more severe phenotype than Z *gpr126*^{stl47} mutants. In contrast, the severity of Z and MZ *gpr126*^{stl49} mutant phenotype were similar (PMID: 25695270).

To address this issue, we have modified the Discussion section accordingly (page 18, line 404).

4. p. 10: How does the CTF in Tg(fli1a:gpr126 CTF-p2a-tdTomato) look like? Does it have a signal peptide? Also, from the description in the Methods section, it seems that it does not have a tag or anything on its N-terminal side. A pure CTF might be rather instable and not be

expressed too well at the membrane. This has been previously described in the context of Gpr126 (Liebscher et al. 2015). The authors do show expression in Fig. S3B, however, demonstrating proper expression of this construct at the cell membrane is absolutely necessary for the interpretation of the results. Thus, a tissue culture experiment would be good.

We thank the reviewer for pointing out that we did not describe the construct in enough detail. Please note, that the cytoplasmic red staining in Fig. S3d represents tdTomato which is separately expressed from CTF and thus cannot localize to the membrane. The utilized construct to generate the *Tg(fli1a:gpr126CTF-p2a-tdTomato)* line is similar to the construct used by Liebscher et al. 2015 and encodes the CTF with a signal peptide as well as an HA-tag at the N-terminus and FLAG-tag at the C terminus. This CTF construct was shown to be functional and displayed increased basal activities in cAMP assays (PMID: 30849857). In addition, P2Y₁₂ N-terminus peptide was fused to the Gpr126 CTF to facilitate the plasma membrane localization of the receptor.

To address this issue, we have modified the Methods section accordingly (page 25, line 587) and added a scheme of the construct to Fig. 3. In addition, we provide anti-FLAG staining of 48 hpf *Tg(fli1a:gpr126CTF-p2a-tdTomato)* embryos that show membrane localization of the CTF in endothelial and endocardial cells *in vivo* (Fig. 3c). Notably, this construct was sufficient to rescue trabeculation- (Fig. 3d,e) and Notch-related (Fig. 6e,f) phenotypes in *gpr126^{stl49}* mutants.

5. p 10: The authors claim a CTF and an NTF function of Gpr126 in heart development. I am a bit worried about the postulated CTF function in this context. I agree that the CTF is required, but I am not sure that it is a sole CTF function. It has been shown for quite some aGPCR that NTF and CTF can reconstitute to form a (function) receptor (e.g. Silva et al. 2009, Prömel et al. 2012). As in *st49*, the N terminus is still present, providing the CTF in this context could lead to such a reconstitution.

To verify the results and to prove the interpretation the authors provide, the following control experiments would be necessary:

a) A rescue with a full-length Gpr126 construct specifically in this context.

b) A rescue of the CTF in the *st47* strain.

We thank the reviewer for pointing out the issue and agree that our data do not prove that CTF has an NTF-independent function but that it is required for heart development. Accordingly, we have removed all statements that indicate that CTF has an NTF-independent function.

In addition, we tested whether the CTF can rescue phenotypes in MZ *gpr126^{stl47}* mutants. We overexpressed CTF in the endothelium of MZ *gpr126^{stl47}* embryos and observed that CTF overexpression failed to rescue the adherens junction defects in MZ *gpr126^{stl47}* mutant (Fig. S9b). Moreover, overexpression of CTF in MZ *gpr126^{stl47}* mutants also failed to alter Notch activity (Fig. S10c, d). These data substantiate the conclusion that CTF does not have an NTF-independent effect.

6. Independently of whether the postulated CTF function is based solely on the CTF or on a combination of NTF/CTF, it remains elusive of which nature this function is. Obviously, the activation of a G protein cascade is the most likely CTF function. However, the authors show no such thing. Gpr126 signaling has been characterized quite well in zebrafish, such as receptor activation by a tethered agonist-derived peptide (Liebscher et al. 2014). It would be interesting to see whether a specific activation of the aGPCR by such a peptide rescues the trabeculation phenotype described on p. 8/9. The results of such an experiment will prove the involvement of the receptor CTF and shed light on the mechanism behind it.

We agree with the reviewer that it is likely that CTF signals via a “activation of a G protein cascade”. Previously, it has been shown that ear and PNS phenotypes can be rescued in the absence of the CTF by forskolin or IBMX treatment, which are both ubiquitously inducing cAMP signaling. As stated on page 11, line 258, “Therefore, we tested whether forskolin/IBMX treatment can also rescue the heart phenotype in *gpr126^{st49}* mutants. We confirmed that these treatments rescued the puffy ear phenotype (Fig. S7a). Both treatments blocked trabeculation in WT hearts as well as multilayering in *gpr126^{st49}* hearts (Fig. S7b). As forskolin and IBMX inhibit also in WT hearts trabeculation, it remains unclear whether Gpr126 acts through cAMP activation to promote trabeculation.”

It has to be noted that trabeculation is a dynamic process and general elevation of cAMP signaling have a deleterious effect on trabeculation. We have also added new text in the discussion on page 19 line 443; “It is likely that CTF signals in the endocardium through canonical G protein-coupled signaling, as observed in myelination during peripheral nervous system development. However, as the treatment with cAMP elevating drugs affected also trabeculation in WT hearts, it could not be determined whether CTF acts through canonical G-protein signaling. The effect on WT hearts is in agreement with recent reports that adrenergic signaling strengthens cardiomyocyte cohesion, altered cAMP levels are associated with impaired trabeculation, and that compartmentalized cAMP signaling is essential for proper heart development.”

Regarding the use of a tethered agonist-derived peptide: This approach cannot be executed as such peptides only function in the presence of CTF, yet, neither *gpr126^{st47}* or *gpr126^{st49}* mutants express the CTF.

7. p. 13/14: How is the NTF in *Tg(myI7:gpr126 NTF-p2a-tdTomato)* designed? Is it the same as the construct in *st49* or is it truly the NTF (stopping at the cleavage site within the GPS)?

The NTF encoded in *Tg(myI7:gpr126 NTF-p2a-tdTomato)* is NTF^{ΔGPS}; the same as in the *gpr126^{st49}* mutants. To address this issue, we have modified the Methods section providing this information (page 25, line 587) and ensured that we defined NTF^{ΔGPS} in the beginning of the manuscript (Abstract as well as Introduction) and consistently used NTF^{ΔGPS}.

8. p. 14/15: The modulation of Notch signaling by Gpr126 is interesting. However, this data is very preliminary and highly speculative. At this point, it is definitely not solid enough to claim that Gpr126 regulates Notch activity.

The authors find that abrogating almost the entire receptor does not have a function on Notch activity. *St49* leads to enhanced Notch activity, but overexpression of the NTF does not. From this, the authors conclude that the CTF inhibits Notch activity. However, several other scenarios are equally possible: Firstly, it can mean that the *st49* construct conveys a dominant-negative effect (here, it is essential to know whether *Tg(myI7:gpr126 NTF-p2a-tdTomato)* carries the same construct as *st49* – see my comment 7 above). Secondly, a purely compensatory mechanism is conceivable. The authors need to put some more effort into obtaining more and solid data on whether and how Gpr126 has an effect on the Notch pathway. The following controls might help:

- a) A CTF expression in the *st47* strain.
- b) An NTF expression in the *st47* strain.
- c) A CTF expression in the *st49* strain.

We agree with the reviewer that we do not show that Gpr126 or CTF directly regulate myocardial Notch activity but that Gpr126 is required.

To further substantiate a link between Gpr126 and myocardial Notch, we have performed all the suggested experiments which are described on page 16, line 360: “Notably, NTF^{ΔGPS} overexpression in MZ *gpr126^{st47}* mutants [reviewer suggestion b], which leads to myocardial multilayering (Fig. 4b,c), also results in a substantial increase of Notch activity at 120 hpf (Fig. 6c, d). Overexpression of CTF in the MZ *gpr126^{st49}* [reviewer suggestion c] mutants, which restores normal trabeculation, also restores the myocardial Notch activity (Fig. 6e, f).”

Furthermore, we tested overexpression of NTF and CTF in the WT background (page 16, line 364): “Overexpression of NTF^{ΔGPS} and CTF in the WT background did not lead to any substantial changes in the myocardial Notch activity (Fig. S10a, b). Additionally, overexpression of CTF in MZ *gpr126^{st47}* mutants also failed to alter Notch activity (Fig. S10c, d).” [reviewer suggestion a].

We conclude: “This suggests that CTF signaling is required to inhibit myocardial Notch activity to allow cardiomyocytes to attain trabecular identity.” (page 16, line 366).

In addition, we provide an explanation why no alteration of Notch signaling was observed in the MZ *gpr126^{st47}* mutants (page 20, line 460): “The MZ *gpr126^{st47}* mutants however, do not form a proper compact wall due to the absence of NTF and hence do not reach the developmental stage where myocardial Notch activity could be enhanced.”

Finally, we removed in the text all notions that Gpr126 or CTF regulate myocardial Notch but that the Gpr126/CTF is required for modulating myocardial Notch activity or that impaired signaling of CTF causes alterations in myocardial Notch activity.

Regarding the possible interpretations:

- a) The possibility that NTF^{ΔGPS} conveys a dominant effect has been excluded (see response to Point 1).
- b) Compensatory mechanism: It has been shown that Notch signaling in cardiomyocytes has an inhibitory effect on trabeculae formation. Thus, a compensatory mechanism due to lack of trabeculation would not be the activation of Notch, which we observed here, but its inhibition.

9. p. 14/15: A highly puzzling issue is that if the CTF inhibits Notch activity as the authors postulate, one would expect that *st47*, in which the CTF is also missing, likewise leads to an increased Notch activity. This is, however, not the case and needs further clarification. I am wondering what the authors' explanation for this. Also, additional essential questions need to be addressed such as: Does the expression of the NTF in the *st49* strain alter the expression of the Notch receptor/ligand? Altered expression levels of Notch pathway components could help explain changed Notch activity.

Please see our response to Point 1 and 8. In addition, please note that our data indicating that NTF and CTF are required at different timepoints during development is in agreement with observations during myelination where NTF is required for radial sorting which is a prerequisite for axon wrapping and myelin sheath formation which depends on CTF signaling (PMID: 25695270) (page 18, line 413).

We also reasoned that due to the various methods of Notch signaling regulation, we examined the most time-sensitive reagent available in zebrafish (*Tg(TP1:Venus-PEST)*). We would also like to note that Notch signaling occurs not only in cardiomyocytes but also in endocardial cells explained now on page 20, line 451. Moreover, Notch signaling is very strong and constantly active in AVC and OFT (as shown in Fig. 6), which makes it very difficult to do

in-situ hybridization for Notch pathway components. This is the reason that in previous publications with myocardial Notch, the only readout has been the Venus-PEST (PMID: 27926871, 33208950, 34363825). Additionally, myocardial Notch signaling is poorly understood and the identification of compact wall cardiomyocyte-specific Notch targets is beyond the scope of this manuscript.

10. p. 15: Again: How is the NTF in *Tg(myI7:gpr126 NTF-p2a-tdTomato)* designed? If it does not mimic *st49*, the data obtained with the construct cannot be compared to the ones on the *st49* (Fig. 7A, B). If it does mimic *st49*, a potential dominant-negative effect of the construct cannot be excluded. Thus, in order to gain the most conclusive set of data, both constructs might need testing.

The NTF encoded in *Tg(myI7:gpr126 NTF-p2a-tdTomato)* is NTF^{AGPS}, same as the protein encoded in *gpr126^{st49}* mutants and based on our rescue experiments, we conclude that it does not have a dominant negative effect (see response to Point 1 and Point 7).

11. p. 14/15: Simply showing an overexpression of the NTF in a wild-type background does not address the question which parts of the aGPCR are required for a potential function in the control of Notch activity. Further, it might help to also do the reciprocal experiment and e.g. express the CTF to see if a more pronounced downregulation effect occurs.

Please see our response to Point 8.

Minor comments:

1. Fig 1C: Information on statistics is missing: How many samples?

We have added the sample number in the figure legends.

2. There are some typos in the manuscript, e.g. p. 71. 147 “amino acid”. Please check spelling and grammar carefully.

We thank the reviewer for pointing out the typos and errors in the manuscript. We have carefully reviewed the manuscript and corrected spelling and grammar.

Reviewer #2 (Remarks to the Author):

In this manuscript, Srivastava et al. studied the functions of adhesion G protein-coupled receptor 126 (*Gpr126*) in zebrafish heart trabeculation by analyzing two truncation mutants *stl47* and *st49*. They found that maternal zygotic *stl47* mutant exhibited hypotrabeculation phenotype with randomly distributed N-cadherin at membranes of the compact muscle layer. The *st49* mutant lacking the C-terminal GPS motif exhibited a multilayered ventricular wall with normal N-cadherin distribution and elevated Notch signaling. They concluded that the N-terminal fragment (NTF) of *Gpr126* maintains cell-cell adhesion and compact wall integrity in the myocardium, while the C-terminal fragment (CTF) is essential to establish the trabecular identity of cardiomyocytes. The finding here would provide novel insights into heart development and clarify the function of *Gpr26* in myocardium trabeculation. However, as noted below, the conclusions might be a bit premature and require further analysis or clarification. The manuscript is well written and easy to follow.

We thank the reviewer for his/her constructive comments about our work and appreciate the suggestions.

1. The authors used qPCR to show the relative expressions of *gpr126* in CMs and endocardium, which is less optimal. Demonstrating the protein localization, either by an antibody staining or by tagging the gene, would be useful for mechanistic interpretations (e.g., a strategy used in a co-author's previous paper: <https://doi.org/10.1242/dev.171207>). It is understandable that this request is likely beyond the scope of a revision. However, *in situ* hybridization of *gpr126* showing the expression locations during trabeculation would be helpful.

After testing all commercially available anti-Gpr126 antibodies, we could find no reliable working antibody in mice, rat, or zebrafish. This is a common problem which is being faced by all labs working on Gpr126. Although precise tagging of Gpr126 is an ideal method, due to the preference of zebrafish in using non-homologous end joining instead of homologous DNA repair, we believe that the probability of successfully generating a knock in is very low and beyond the scope of this manuscript.

To address this issue, we decided for direct visualization of *gpr126* transcripts by combining RNAscope in whole mount zebrafish embryos with phalloidin immunostaining at 48 hpf (before the onset of trabeculation) and 96 hpf (after the onset of trabeculation). The obtained data shown in Fig. S5 confirm our qPCR data that *gpr126* is expressed in the endocardium in zebrafish (see page 10, line 229).

2. The authors only showed the myocardium in all figures. Since *gpr126* is considered an endocardial-enriched gene in the heart, it seems necessary to assess any endocardial defects and endocardial cell-cardiomyocyte interactions in the mutants. Otherwise, the mechanism described in the current version seems to lack a significant component.

Previously, we have shown that the NTF binds to the myocardium (PMID: 24082093). Consequently, there is a myocardial and an endocardial component.

To address this issue, we have provided data showing that there are no obvious defects in the endocardium or the cardiac jelly between the endocardium and the myocardium at 72 and 96 hpf. We have added this information to the revised manuscript and state on page 9, line 197: "Notably, MZ *gpr126*^{stl47} and MZ *gpr126*^{st49} mutants in the background of *Tg(myf7:mKATE-CAAX); Tg(krt4:GFP)*^{sqet33} which label the myocardium and the endocardium, respectively, did not exhibit obvious defects in the endocardium or the cardiac jelly between the endocardium and the myocardium at 72 and 96 hpf (Fig. S3)."

Please note, that the interaction between endocardium and myocardium is still poorly understood (see page 20, line 466) and while a deeper analysis of the endocardium is of great interest to us, it is beyond the scope of this manuscript.

3. Figure 1B. It is interesting that 36.67% of *stl47* MZ mutant larvae and 26.67% of *st49* MZ mutant larvae showed normal trabeculation. Could the authors speculate the reason? How do they define "normal trabeculation"?

It is true that the mutants exhibit variable penetrance and expressivity. It is fairly common in zebrafish mutants that even progenies of the same mutant fish exhibit different penetrance (PMID: 36533556).

The criteria we have chosen for normal trabeculation was 3 or more trabeculae with 3 or more cardiomyocytes for 96 hpf hearts and 5 or more trabeculae with 3 or more cardiomyocytes for 120 hpf hearts. This information has been added to the Methods section (see page 30, line 694).

4. The authors showed that mosaic myocardial-specific NTF overexpression induced local cardiomyocyte multilayering and concluded that NTF has a function in maintaining cell-cell

adhesion in the compact layer cardiomyocytes. However, *gpr126* is considered an endocardial gene (Figure 3A). It is confusing how an endocardium secreted protein mediated cell adhesions within the myocardium. At least the outer layer of the multilayered myocardium seems to have no contact with the endocardium. Defining the expression domain of *gpr126* (as noted above) seems necessary to clarify this. Also, an endocardial-specific overexpression strategy should be considered.

As requested, we have defined the expression domain of *gpr126* (see response to Point 1), which is exclusively expressed in the endocardium in zebrafish larvae.

Several secreted factors from endocardial cells during cardiac development simultaneously modulate cardiomyocyte proliferation and maturation, which are key cellular processes critical for ventricular compaction. How these endocardial factors affect the myocardium is poorly understood (see page 20, line 470). Please note, that it has also recently been reported in zebrafish that the endocardium has direct contact with the myocardium (PMID: 35225788). In addition, we have previously shown (PMID: 24082093) that the NTF^{ΔGPS} is secreted from the endocardium, binds to the myocardium, and exerts its function. Furthermore, we would like to point out that the compact layer in the zebrafish heart ventricle is single layered, at least during the early stages of development. The NTF is acting at a stage when there is a single layered myocardial wall with no trabeculae formed yet. Regarding the multilayering phenotype in *gpr126*^{st49} mutants, the NTF acts at the single myocardial layer stage to maintain the cell-cell adhesion between the cardiomyocytes. Yet, at the onset of trabeculation, lack of CTF results in increased Notch activity resulting in decreased delamination and multilayering.

5. Figure 7. The authors concluded that CTF signaling is required to inhibit myocardial Notch activity to allow cardiomyocytes to attain trabecular identity. However, the MZ *stl47* mutant also lacks the CTF but showed normal myocardial Notch activity. Please explain.

We would like to clarify that the fact that *gpr126*^{st49} mutants, lacking CTF, show enhanced Notch activity but MZ *gpr126*^{stl47} mutants, lacking NTF and CTF, does not, is not due to a dominant effect of NTF^{ΔGPS} but due to the fact that NTF is required for compact wall integrity before trabeculation is initiated. To clarify this point we added on page 15, line 345: "It has been demonstrated that Notch signaling is activated during trabeculation upon cardiomyocyte delamination in a subset of compact wall cardiomyocytes to inhibit their delamination. As MZ *gpr126*^{stl47} mutants are characterized by hypotrabeulation, no change in myocardial Notch activity is expected." We have also added an explanation in the Discussion (page 20, line 460): "The MZ *gpr126*^{stl47} mutants, however, do not form a proper compact wall due to the absence of NTF and hence do not reach the developmental stage where myocardial Notch activity could be enhanced."

Please note that our data indicating that NTF and CTF are required at different stages during development are in agreement with observations during myelination where NTF is required for radial sorting which is a prerequisite for axon wrapping and myelin sheath formation which depends on CTF signaling, and we have also added this in the Discussion. (page 18, line 413)

6. Figure S5. It is labeled as *fli1a:gpr126* NTF-p2a-tdTomato in the figure but *myl7:gpr126* NTF-p2a-tdTomato in the legend and main text. Please clarify. I assume it is *myl7*. The myocardium looks quite normal compared to the mosaic myocardial-specific NTF overexpression phenotype in Figure S4 (*myl7:gpr126* NTF-p2a-tdTomato). Please clarify what makes the difference.

We thank the reviewer for pointing out this labeling error. It is *myl7* and we have changed the figure accordingly (Fig S10a).

The observed differences are most likely due to the differences in expression: a) different copy number. b) uniform (transgenic) vs. mosaic (plasmid injection) NTF overexpression (results in adhesion heterogeneity between cardiomyocytes).

As the mosaic expression was performed in WT, these data are difficult to interpret. In addition, plasmid injections in single cell-stage embryos are difficult to control. Therefore, we have decided to remove these data and instead added data obtained by stable overexpression of NTF^{ΔGPS} in the myocardium of MZ *gpr126^{stl47}* mutants. This resulted in multilayering (Fig. S4a,b)(page 10, line 215) and enhanced myocardial Notch activity (Fig S6c,d)(page 16, line 360) as observed in *gpr126^{stl49}* mutants. This validates the role of NTF^{ΔGPS} in maintaining compact wall architecture.

7. Can the authors explain or speculate why zygotic *stl47* mutant but not *st49* mutant exhibited maternal effect?

Our data show that the NTF is required earlier in development than CTF. NTF is required early in development for compact wall formation by regulating N-cadherin localization, a prerequisite for proper trabeculation. In contrast, the CTF is required later for the subsequent control of Notch activity and attainment of trabecular identity. As maternal mRNA is only transiently present until early embryonic stages, these findings provide a likely explanation for why the Z *gpr126^{stl47}* mutant exhibits a maternal effect, but the Z *gpr126^{st49}* does not.

Notably, our data are in agreement with a previous report in the peripheral nervous system, where it has been shown that MZ *gpr126^{stl47}* mutants exhibited a more severe phenotype than Z *gpr126^{stl47}* mutants. In contrast, the severity of Z and MZ *gpr126^{st49}* mutant phenotype were similar (PMID: 25695270).

To address this issue, we have modified the Discussion section accordingly (page 18, line 404).

8. Control group is missing in the experiment shown in Figure S5

It is unclear to the authors what the reviewer means. In this figure (now Fig. S10a), we wanted to determine whether Notch activity is affected in NTF^{ΔGPS} overexpressing hearts (*Tg(myl7:gpr126NTF-p2a-tdTomato)*) compared to control WT hearts (*Tg(TP1:Venus-Pest)*).

Please note that we have added now additional experiments to further support that Gpr126 is required for myocardial Notch activity (page 16, line 360): “Notably, NTF^{ΔGPS} overexpression in MZ *gpr126^{stl47}* mutants, which leads to myocardial multilayering (Fig. 4b,c), also results in a substantial increase of Notch activity at 120 hpf (Fig. 6c, d). Overexpression of CTF in the MZ *gpr126^{st49}* mutants, which restores normal trabeculation, also restores the myocardial Notch activity (Fig. 6e, f).” Furthermore, we tested overexpression of NTF and CTF in the WT background (page 16, line 364): “Overexpression of NTF^{ΔGPS} and CTF in the WT background did not lead to any substantial changes in the myocardial Notch activity (Fig. S10a, b). Additionally, overexpression of CTF in MZ *gpr126^{stl47}* mutants also failed to alter Notch activity (Fig. S10c, d).” We conclude: “This suggests that CTF signaling is required to inhibit myocardial Notch activity to allow cardiomyocytes to attain trabecular identity.” (page 16, line 366).

9. Figure 4A and B lack quantification results.

As requested, we have added quantifications (now Fig. S8b,c).

10. Page 8, lines 151-153. “The analysis of zygotic *gpr126^{stl47/stl47}* mutant hearts revealed no obvious gross morphological defects compared to WT siblings suggesting that *Gpr126* is not required for heart development in zebrafish (Figure S2).” This statement is not accurate. The result only suggests that a zygotic mutant did not show a heart development defect. One cannot conclude that *Gpr126* is not required for heart development.

To address this issue, we changed the text on page 8, line 176 to “These data suggested initially that *Gpr126* is not required for heart development in zebrafish and that $\text{NTF}^{\Delta\text{GPS}}$ exhibits a dominant effect. However, the lack of a phenotype in Z *gpr126^{stl47}* mutant hearts could also be explained by compensation through maternal deposition of WT mRNA or protein into the egg.”

11a. *st49* and *NTFdeltaGPS* are used to represent the same mutant. It would increase the readability by using one label across the manuscript.

Please note that “*st49*” represents the mutated allele, while *St49* represents the expression construct. Both of these encode for the protein $\text{NTF}^{\Delta\text{GPS}}$. We have changed the manuscript to clarify this point. In addition, please note that the *Tg(myf7:gpr126NTF-p2a-tdTomato)* encodes also $\text{NTF}^{\Delta\text{GPS}}$ but is not the same as the “*st49*” allele (endogenous *gpr126* coding sequence containing the respective mutation).

11b. Similarly, both *gpr126^{stl47}* and *gpr126^{stl47/stl47}* were used to denote the homozygous mutant in the figures (the same for *st49*). Please be consistent.

Thank you for indicating this issue. The manuscript has been changed accordingly.

12. Page 6, line 103. Please include citations for these 2 published mutant alleles.

We have included the citations (page 6, line 121).

Reviewer #3 (Remarks to the Author):

This is an interesting study that aims to clarify a long-standing controversy in the literature concerning the function of *Adgrg6/Gpr126* in the zebrafish heart. *Adgrg6/Gpr126* is a clinically important adhesion class GPCR (aGPCR) that is associated with a number of different disease states in humans, and so a better understanding of genetic variants in the zebrafish model is important and likely to be of general interest.

Several studies in mice have revealed a function for *Gpr126* in cardiac trabeculation. However, the murine cardiac defects did not appear to be recapitulated in the various zebrafish mutant alleles available, raising the possibility that a function for *Gpr126* in the heart is not conserved. Nevertheless, many of the available zebrafish alleles are hypomorphic and/or predict expression of truncated protein variants, which has been offered as an explanation for the lack of cardiac defects. Other alleles have not yet been examined for a cardiac phenotype. A morpholino study in zebrafish (Patra et al., ref. 13) supported a role for the N-terminal fragment (NTF) of *Gpr126* in cardiac trabeculation defects; as discussed in the current manuscript, genetic compensation could provide an explanation for the discrepancy between the morphant and mutant phenotypes. The Engel group have also already published a study linking *gpr126* regulation to cardiac trabeculation (Musa et al., 2019), which is not cited in the current manuscript. The role for *Gpr126* in the zebrafish heart therefore remains unclear.

The current manuscript re-visits this controversy through analysis of the zebrafish heart phenotype in two of the most severe truncating alleles available, using maternal-zygotic (MZ) as well as zygotic-only (Z-only) mutants to examine the phenotype. The authors clearly

demonstrate that the MZ *stl47* and MZ and Z-only *st49* homozygous mutant zebrafish have defects in trabeculation of the endocardium, each showing variable penetrance and expressivity, and a range of different phenotypes (multilayering, hypotrabeculation or absence of trabeculation). Other data are promising but more preliminary, indicating possible links to blood flow, *ErbB2* and Notch signalling.

The manuscript contradicts some previous findings (some of which are from the same group, or in papers with authors in common with the current manuscript). As the authors state on line 376, the work fails to corroborate their previous hypothesis that the NTFdelGPS is sufficient for cardiac trabeculation (line 376), instead now suggesting that the CTF plays a critical role. They also suggest that their data contradict the recent published interpretation that the heart phenotype in mouse results from a placental defect (Torregrosa-Carrion et al., 2021, ref. 15). These contradictions are not easy to follow in the manuscript. There are a number of concerns with presentation and interpretation of the data, and of citation and discussion of the previous literature, that need to be addressed.

We thank the reviewer for carefully reading the manuscript and the constructive comments, which we have all addressed by modifying the text as well as adding a significant amount of new experimental data.

Major comments

1. One clear but surprising new finding is that the *st49* allele, which the authors demonstrate is likely to express the NTFdelGPS fragment, is the only *gpr126* allele described so far for which there is a cardiac trabeculation phenotype in zygotic (Z-only) mutants. This is contrary to the authors' previous published analysis of *gpr126 st49* mutants (Patra et al., 2013: '*gpr126st49* mutants exhibited no pericardial edema or other cardiac abnormalities') and their demonstration that the NTFdelGPS alone was sufficient to rescue a trabeculation defect in *gpr126* morphants (Patra et al., 2013) (ref. 13). However, the interesting result of the Z-only phenotype in *st49* is hidden in the manuscript—it is only presented as supplementary data (Fig. S2), and the title of the figure (Zygotic *gpr126stl47* hearts do not exhibit trabeculation defects) completely overlooks the result. This figure should be moved to form a top row in the main Fig. 2, allowing a direct comparison of the Z-only and MZ phenotypes for each allele. The abstract should also mention this result.

As requested, we have added the data for the zygotic mutants for both alleles in Fig. 2. In addition, we have added this information in the abstract (page 2, line 35) and discussed why only Z *gpr126^{st49}*, but not Z *gpr126^{stl47}*, mutants exhibit a phenotype (page 18, line 404).

As discussed already in the previous manuscript (now page 18, line 417), in Patra et al. (PMID: 24082093) the mutants and morphants were only observed until 80 hpf, as morpholinos lose their effect at later stages. Here, we observed multilayering in the *gpr126^{st49}* mutants at 96 hpf.

The phenotypes shown are clearly incompletely penetrant and variably expressive for both the MZ and Z-only mutants for *st49*, and for MZ *stl47* mutants, but this is not explicitly stated anywhere. Moreover, the distribution of cardiac phenotypes in the Z-only *st49* mutants appears identical to that in MZ *st49* mutants; given the low numbers of fish sampled ($n=15$ Z-only (Fig. S2) and $n=30$ MZ (Fig. 2)), there is no evidence for any significant difference in strength or nature of the phenotype between the Z-only and MZ fish. This finding should be presented more clearly, as suggested above. If the authors wish to document a difference in the distribution of cardiac phenotypes between Z-only *st49* and MZ *st49* fish, they will need to examine more animals and perform an appropriate statistical test on the data to confirm this.

We have altered the text to point out that both mutants exhibit incomplete penetrance and variable expressivity (page 9, line 210).

In regards to differences in strength or nature of the phenotype between Z and MZ *gpr126*^{st49}, we do not expect any difference considering the late requirement of CTF (see Discussion regarding maternal effect on page 18, line 404. Our current data is in agreement with this conclusion (Fig. 2). This data is also in concordance with the study from Petersen *et al* (PMID: 25695270), where both Z and MZ *gpr126*^{st49} mutants exhibit similar phenotypic severity.

2. One interpretation of the data is that the NTFdelGPS has a dominant effect to give rise to the unique cardiac phenotype (multilayering) seen in some embryos for the st49 allele. Do the heterozygotes for this allele show any cardiac phenotype?

We did not observe any multilayering phenotypes in the heterozygous *gpr126*^{st49} larvae consistent to the absence of myelin phenotypes in heterozygous *gpr126*^{st49} in Petersen *et al.* 2015 (PMID: 25695270) (page 10, line 219). This indicates that the observed phenotypes are not due to the dominant effect of NTF^{ΔGPS}.

To further exclude the possibility that the *gpr126*^{st49} mutant phenotypes are due to a dominant effect of NTF^{ΔGPS}, we added a large number of additional experiments:

1) We provide evidence that NTF^{ΔGPS} encoded by the *st49* allele is a stable, secreted protein (Fig. 1, Fig. S2). Notably, previously we have already shown that ectopic expression of NTF^{ΔGPS} in Hela as well as H9c2 cells results in a stable, secreted protein.”

2) Importantly, if the *gpr126*^{st49} mutation would result in the expression of a misfolded protein exhibiting dominant effects, it should not be possible to rescue this mutant phenotype by overexpression of the CTF. Therefore, we had provided in the original version data regarding these rescue experiments. To further strengthen these data, we have now added additional evidence for *gpr126* being expressed in the endocardium in zebrafish heart (RNAscope, Fig. S5, page 10, line 229) and that overexpressed CTF localizes in the transgenic animals in endothelial and endocardial cells at the membrane (Fig. 3c, end of page 10). Furthermore, we provide new data showing that overexpression of CTF in the MZ *gpr126*^{st49} mutants, which restores normal trabeculation, also restores myocardial Notch activity (Fig. 6e, f, page 16, line 362).

Collectively, these data show that NTF^{ΔGPS} represents a functional NTF.

3. The sections linking Gpr126 to blood flow, and the Erbb2 and Notch signalling pathways, are less well worked up and need additional experimental confirmation.

a. Figure 4. The description of the Erbb2 inhibitor and blood flow experiments shown in this figure is confusing. The genetic backgrounds tested (over-expressed *gpr126*, or the MZ *st49* mutant, Fig. 4) both already have blocked trabeculation, so how can the authors conclude that the Erbb2 inhibitor blocks trabeculation in these backgrounds? Instead, both the inhibitor and lack of blood flow appear to rescue the multilayering phenotype in each background. This needs further discussion and explanation. Quantification of the phenotypes should also be provided in the figure legend.

We thank the reviewer for pointing out that this point needs more clarification.

ErbB2 signaling and blood flow are crucial regulators of trabeculation. In the absence of either of these factors in a WT heart, there is complete absence of trabeculation and the heart wall remains monolayered. The multilayering phenotype of *gpr126*^{st49} mutants and aneurysm-like phenotype of the *gpr126* overexpressing line raised the question whether Gpr126 has a trabeculation promoting effect that might act in parallel to ErbB2 signaling overriding the

requirement of ErbB2 signaling. Therefore, we tested whether inhibition of ErbB2 signaling has an effect on the phenotypes observed in *gpr126^{st49}* mutants and the *gpr126* overexpressing line.

To address this issue, we have included this argument on page 12, line 277. In addition, we quantified the data as requested (Fig. S8b,c).

b. The authors document an increase in Notch signalling in the MZ st49 mutants in Fig. 7, and show that the CTF can rescue the multilayering phenotype in this allele in Fig. 3. Importantly, can over-expression of CTF also restore normal levels of Notch signalling in MZ st49? This is an essential control for the interpretation of the mechanism here.

We thank the reviewer for suggesting this important control experiment. We performed the suggested experiments and observed that the overexpression of CTF restores normal levels of Notch signaling in MZ *gpr126^{st49}* mutants, and have included it in the manuscript. (page 16, line 362) (Fig. 6, f).

4. The authors also speculate on lines 421–422 that rescue by the CTF is likely to be through canonical G-protein-coupled signalling. This would be simple to test by raising cAMP with forskolin, a treatment that should rescue the cardiac phenotype in the st49 allele. This experiment is essential to support the argument.

We agree with the reviewer that it is likely that CTF signals via activation of a G protein cascade. Previously, it has been shown that ear and PNS phenotypes can be rescued in the absence of the CTF by forskolin or IBMX treatment, which are both ubiquitously inducing cAMP signaling. As stated on page 11, line 258, “Therefore, we tested whether forskolin/IBMX treatment can also rescue the heart phenotype in *gpr126^{st49}* mutants. We confirmed that these treatments rescued the puffy ear phenotype (Fig. S7a). Both treatments blocked trabeculation in WT hearts as well as multilayering in *gpr126^{st49}* hearts (Fig. S7b). As forskolin and IBMX inhibit also in WT hearts trabeculation, it remains unclear whether Gpr126 acts through cAMP activation to promote trabeculation.”

It has to be noted that trabeculation is a dynamic process and general elevation of cAMP signaling have a deleterious effect on trabeculation. We have also added new text in the discussion on page 19 line 443; “It is likely that CTF signals in the endocardium through canonical G protein-coupled signaling, as observed in myelination during peripheral nervous system development. However, as the treatment with cAMP elevating drugs affected also trabeculation in WT hearts, it could not be determined whether CTF acts through canonical G-protein signaling. The effect on WT hearts is in agreement with recent reports that adrenergic signaling strengthens cardiomyocyte cohesion, altered cAMP levels are associated with impaired trabeculation, and that compartmentalized cAMP signaling is essential for proper heart development.”

5. Over-expression of the NTF in a wild-type background results in multilayering (Fig. S4). Does over-expression of the NTF in MZ stl47 mutants result in an st49-like multilayering phenotype? This would help to clarify the role of the NTF.

We thank the reviewer for this suggestion. We performed the suggested experiment and observed that a stable overexpression of NTF in the myocardium of MZ *gpr126^{stl47}* mutants leads to multilayering (Fig. S4a,b)(page 10, line 215) and enhanced myocardial Notch activity (Fig S6c,d)(page 16, line 360) as observed in *gpr126^{st49}* mutants. This validates the role of NTF^{ΔGPS} in maintaining compact wall architecture.

As the mosaic expression was performed in WT, these data are difficult to interpret. In addition, plasmid injections in single cell-stage embryos are difficult to control. Therefore, we have decided to remove these data and instead added data obtained by stable overexpression of NTF^{ΔGPS} in the myocardium of MZ *gpr126^{stl47}* mutants.

6. This also needs some more discussion in the manuscript: why does the maternal transcript fail to compensate for the loss of zygotic function for the *st49* allele specifically? And why does the *stl47* allele (which also lacks the CTF) not show any Z-only phenotype? Can the MZ *stl47* phenotype also be rescued with forskolin?

Our data show that the NTF is required earlier in development than CTF. NTF is required early in development for compact wall formation by regulating N-cadherin localization, a prerequisite for proper trabeculation. In contrast, the CTF is required later for the subsequent control of Notch activity and attainment of trabecular identity. As maternal mRNA is only transiently present until early embryonic stages, these findings provide a likely explanation for why the Z *gpr126^{stl47}* mutant exhibits a maternal effect, but the Z *gpr126^{st49}* does not.

Notably, our data are in agreement with a previous report in the peripheral nervous system, where it has been shown that MZ *gpr126^{stl47}* mutants exhibited a more severe phenotype than Z *gpr126^{stl47}* mutants. In contrast, the severity of Z and MZ *gpr126^{st49}* mutant phenotype were similar (PMID: 25695270).

To address this issue, we have modified the Discussion section accordingly (page 18, line 404).

7. The imaging in Figure 6 is not entirely clear. A counterstain for cell membranes is necessary to show the position of cells, especially as the epithelium is a curved surface. In addition, the chosen 'representative' example for MZ *st49* does not have the multilayering phenotype. It is essential to show what happens to cadherin expression in those MZ *st49* mutant hearts that have a multilayering phenotype.

As requested, we have added the membrane counterstaining using *Tg(myl7:mKATE-CAAX)* which further substantiates our conclusion (Fig. S9a, c).

Regarding the multilayering in *gpr126^{st49}* mutants: Please note that the multilayering phenotype cannot be observed at 72 hpf but only from 96 hpf on. Therefore, in Fig. 4 WT and mutant hearts have a single layered compact myocardial wall, where lateral localization of N-cadherin is only affected in MZ *gpr126^{stl47}* mutants, suggesting defects in compact wall formation. In Fig. 5, we show the N-cadherin phenotype for *gpr126^{st49}* mutants exhibiting multilayering where the N-cadherin remodeling is affected.

Minor comments

8. The order of the alleles presented in Figure 7 (in both the image panels and the graph) should be swapped round to match that in the rest of the manuscript, i.e. *stl47* first, and then *st49*.

We have swapped the order as suggested in both, the image panel as well as the graph.

9. The authors have undertaken a qPCR analysis of FACS-sorted cardiomyocytes (Fig. 3A) to confirm that *Gpr126* mRNA is enriched in the endocardium, which is helpful. However, the Introduction should state that the gene is known to be expressed in the zebrafish heart (since a lack of expression in this organ in the zebrafish could be a simple explanation for any species-specific differences in function), citing relevant studies. The authors could refer to Genq et al. (2013) for expression at 24–48 hpf, or their own study for expression at 48 hpf (Patra et al.,

2013 – although confusingly, note that the overall expression pattern shown in this paper does not match that shown in Monk et al., 2009 or Geng et al., 2013).

We would like to thank the reviewer for pointing this out. We have edited the text accordingly (page 4, line 72). In addition, we decided for direct visualization of *gpr126* transcripts by combining RNAscope in whole mount zebrafish embryos with phalloidin immunostaining at 48 hpf (before the onset of trabeculation) and 96 hpf (after the onset of trabeculation). The obtained data shown in Fig. S5 confirm our qPCR data that *gpr126* is expressed in the endocardium in zebrafish (see page 10, line 229).

10. In all relevant main and supplementary figures (including both images and diagrams), the authors should replace red with magenta, to make the images accessible to those with dichromatic colour vision. For the triple label in Figure 1, choose a suitable colour combination (e.g. magenta, green, white)

We have changed all figures to make them accessible to those with dichromatic color vision.

11. There are some omissions or inconsistencies in discussion of the previous literature, including papers authored by some of the authors of the current manuscript, which make it difficult to follow the arguments.

a. D'Amato et al. (2016) (ref. 46) is cited in the context of myocardial patterning, but the work in this paper to elucidate the relationship between Gpr126 and Notch signalling in the mouse and zebrafish is not mentioned or discussed anywhere. D'Amato et al. (2016) used several lines of evidence to demonstrate that cardiac expression of Gpr126 is a downstream target of Notch signalling, including documenting a loss of *gpr126* expression in the hearts of zebrafish antimorphic *mib* mutants (loss-of-function for Notch signalling). How do these results fit with the model that the authors propose for the zebrafish, with the enhanced myocardial Notch signalling in MZ *Gpr126-st49* mutants?

It is known that endocardial Notch lies upstream of most of the genetic pathways required for trabeculation like EphrinB2, Bmp, as well as Nrg-ErbB2 signaling. In the manuscript of D'Amato et al. (2016), it has been emphasized through various lines of evidence that endocardial deletion of Notch signaling abrogates the cardiac expression of *Gpr126* in zebrafish and mice, which suggested that *Gpr126* is an important Notch effector throughout ventricular chamber development in the endocardium. The relationship of myocardial Notch activity and *Gpr126*, however, was not studied. The role of myocardial Notch activity during trabeculation has just recently gained attention, especially in zebrafish studies. In Priya et al. 2020 (PMID: 33208950) and Han et al. 2016 (PMID: 27357797), it was shown that there is a myocardial Notch-mediated fate specification at play during trabeculation where the compact layer cardiomyocytes activate a Notch reporter, while the trabecular layer cardiomyocytes do not.

Our study indicates that *Gpr126* has a role to play in this fate determination of compact vs. trabecular layer cardiomyocytes. In the *gpr126^{st49}* mutant, the compact wall is properly formed due to the presence of NTF^{ΔGPS} and the initial delamination events occur triggering the myocardial Notch activity in the compact wall cardiomyocytes. The absence of CTF however leads to increased myocardial Notch activity.

To address this issue, we have clarified on page 20, line 451 that D'Amato et al. (2016) studied the relation of *Gpr126* and Notch in endocardial cells, while we discovered a new relationship between *Gpr126* and myocardial Notch.

b. The authors do not cite or discuss previous work from their own group (Musa et al., 2019), in which they specifically linked miR-27a/b regulation of *gpr126* expression to trabeculation in

the zebrafish heart, and compared phenotypes to that of the *gpr126* morphant. This paper appears to provide important background information that supports the current study. Why is this study not cited?

We thank the reviewer for pointing this out. We have cited the study and added the text accordingly on page 4, line 84.

12. A schematic diagram summarising the authors' current model and thinking for how the NTF is working with the CTF, linking to *ErbB2* and Notch signalling and cell polarisation, would be very helpful.

We have added a schematic diagram of our current model (Fig. 7).

13. Introduction, Line 56 – do the authors mean to say 'is expressed in' rather than 'is essential for' here? It seems odd to say that the gene is essential for development of the heart but then to go on to say this is controversial and needs testing.

We have changed the text to “*Adgrg6/Gpr126* has been reported to be required for the proper development of several tissues/organs including the peripheral nervous system, intervertebral disc, ear, and placenta”(page 3, line 64) and added text about the cardiac expression of *gpr126* (page 4, line 72).

14. Much of the information presented in Supplementary Fig. S1 (panels A–C, i.e. both the sequence and the ear phenotype for both alleles) has been published by other groups elsewhere. It is good to see that the authors have been able to confirm these findings in their own hands, but it must be stated that they are confirming previous published findings, citing Petersen et al. (ref 65) for the analysis of *stl47*, and Monk et al. (ref 35) for *st49*, rather than presenting these as new data.

To address this issue, we have cited the papers which first described the alleles and the phenotypes.

15. Please note that the amino acid at the position of the indel in the *stl47* allele is not a glutamine (Q), as previously published by Petersen et al. (2015) – it is an asparagine (N) (as the supplementary figure in the current manuscript clearly shows) (see Table 1 in Baxendale et al., *genesis*, 2021). This should be corrected in the text on line 104.

We would like to thank the reviewer for pointing this out. We have corrected the text accordingly.

Point-by-point response

We thank the reviewers for their in-depth review providing additional constructive comments to further improve our manuscript. In the revised manuscript we have addressed all the raised points by changing the text and/or providing a significant amount of new data.

Reviewer #1:

The revised version of the manuscript by Srivastava et al. has improved remarkably compared to the initial one. I am very content with the additional data and arguments the authors present. They are convincing and I believe that the manuscript is a thoroughly studied piece now that is interesting for a broad readership. However, I have one minor thing I am still a bit concerned about. The authors show an interesting link between GPR126 and Notch activity, which is clearly one of the most interesting aspects of the manuscript. However, there are still many open questions regarding the interplay of the receptor and Notch. Thus, I suggest using more careful and less prominent phrasing e.g. in title and abstract and on the other side a few more details on the open questions and the impact in the discussion.

We thank the reviewer for the overall positive comment. We have changed the title to “Adgrg6/Gpr126 is required for compact wall integrity and establishing trabecular identity during cardiac trabeculation” and used less prominent phrasing in the abstract. In addition, we have modified the Discussion on page 24 (line 550 to 555) clarifying that there are open questions regarding the interplay of Gpr126 receptor and Notch.

Reviewer #2:

The authors have added significant new data and addressed most of my concerns in the revised manuscript. However, a few concerns need further clarification or experimental validation. The authors aimed to clarify a controversy in the function of gpr126 in the zebrafish heart. It is crucial to provide high-quality, mechanistic evidence, notably when the authors claimed some conclusions that contradict previous findings, including their own.

We thank the reviewer for acknowledging the importance of our new data. We agree that high-quality, mechanistic evidence is needed to “contradict previous findings”. Please note that neither the involvement of Notch nor N-cadherin was the matter of controversy (both have previously not been analyzed in any Gpr126-modified animal model) but the overall involvement of Gpr126 in heart development. In this regard, we provide high-quality data (going in much more detail than the previous study, providing transgenic rescue experiments) and on top provide evidence for different roles of NTF and CTF and a link between GPR126 and Notch activity.

Contradictory findings, our own work: We have previously concluded that the NTF of Gpr126 is sufficient for proper heart development. This conclusion was drawn on morphological features, meaning we simply looked at zebrafish embryos in which cardiomyocytes are labeled with GFP. A major limitation or difference to our current study was that morpholino-based studies did not allow to characterize morphants beyond 80 hpf. In addition, similar to the study by the de la Pompa group (Torregrosa-Carrion *et al.*, Sci Adv 2021), we did not look at any other markers. Our detailed study (initiated and requested by our co-author Didier Stainier, also one of the authors in the

Torregrosa-Carrion *et al.* study) resulted in the conclusion that the NTF is not sufficient for proper heart development and that the lack of the CTF causes multilayering. Therefore, we revisited our previous data. Notably, the published data (Fig. 4B, Patra *et al.*, Proc Natl Acad Sci U S A. 2013) clearly show a multilayered heart. We had stated “MO2 morphants were characterized by pericardial edema, lack of trabeculation, and reduced expression of mbp and krox20/egr2 at the PLLn. Injection of 100 pg of capped drgpr126-NTF^{ΔGPS} mRNA encoding amino acids 1–783 (variant 1) rescued the pericardial edema and ventricular trabeculation phenotype in a subset of the MO2 morphants (Fig. 4 A–C).” Apparently, we focused only on whether there is no trabeculation (single layer of cardiomyocytes) or trabeculae (wall with luminal projections) but not multilayering. This underlines the importance of the analysis of Notch and N-cadherin in our current study.

To clarify this point, we have changed the text on page 11, line 255 to 259: “As our new data appeared to be in contrast to our previous findings, we revisited our previously published data regarding overexpressing NTF^{ΔGPS} in gpr126 morphants²². While previously the focus was only on the “re-appearance of trabeculae”, a closer look revealed that the data could also be interpreted as hypertrabeculation or multilayering (note, the analysis was limited to 80 hpf due to the morpholino-based approach).”

Contradictory findings, work by de la Pompa’s group: We have provided a detailed explanation on page 19, line 459, to page 20, line 473, why the group of de la Pompa might not have seen the phenotype. Importantly, we have expanded the analysis beyond the characterization of hearts in which cardiomyocytes are labeled, identifying defects in N-cadherin localization and Notch activation (unique phenotype). We determined that maternal RNA can compensate the depletion of gpr126. In addition, our data suggest that the CTF is required to balance NTF-mediated signaling preventing overactivation of myocardial Notch signaling. This conclusion is based on a number of assays including *in vivo* rescue experiments.

To clarify the point of the need of balanced signaling, we have added a new paragraph to the Discussion on page 20, line 473 to 485, which reads: “Another possible explanation is that a balance of NTF- and CTF-mediated signaling is required for proper trabeculation. This is supported by the observation of a substantial increase of Notch activity in MZ gpr126^{st49} mutants which could be normalized by the overexpression of CTF. Overall, all lines with an increased expression of NTF compared to CTF exhibit the multilayering phenotype with the exception of the NTF^{ΔGPS} overexpression line (Fig. S13a). Notably, in this line cells express zygotic and maternal CTF and NTF and thus the ectopic expression of NTF might not be high enough to significantly impair the balance between NTF and CTF or enough CTF is present to prevent increased Notch activity. In contrast, mosaic expression, usually mediating a markedly higher expression was sufficient to induce local multilayering. In addition, the mosaic expression might result in adhesion heterogeneity between cardiomyocytes. Overall, our data suggest that the CTF is required to balance NTF-mediated signaling preventing overactivation of myocardial Notch signaling.”

1. I'm afraid I have to disagree with the authors' claim that the probability of successfully generating a knock-in is very low. Numerous zebrafish knock-in alleles, including precise tagging alleles, have been published (e.g., PMIDs: 34495314, 32412410, 33681181, 35378361, 36476416, 32064511, 36878640). It is doable to generate precise tagging of Gpr126. It would be the authors' call whether it is necessary for the current study.

We agree that various studies have managed to successfully generate a knock-in in zebrafish and it is doable to generate precise tagging of Gpr126. However, the efficiency of a knock-in in zebrafish is very variable depending upon the gRNA used and the genomic location of the target gene. So far, our approaches haven't been very efficient and also other Gpr126 researchers have not been successful with this approach. In this study, we performed RNAScope analysis, which was the most time-convenient way to identify the localization of Gpr126 transcript in the endocardium. In our opinion the available data are clearly showing the expression of Gpr126 in the endocardium. Therefore, we have decided not to generate a zebrafish knock-in.

2. The authors claimed that NTF is required for the junctional localization of N-cadherin in the working model. However, N-cadherin is still localized to junctions in the stl47 mutant (Fig. 4, S9a). The provided evidence only supports a correlation, and the impaired junctional localization might result from other defects in the compact wall integrity. Notably, images in Figures 4 and S9a are snapshots of representative samples. The phenotype is not clear to me. The WT images in Fig. S9a look very similar to the stl47 images in Fig. 4, depending on which confocal sections were chosen. It is not apparent how the schematic for the stl47 group was generated, which does not recapitulate patterns in the image. The stl47 heart looks smaller than WT in the confocal section image but not in the maximum projection image. More sophisticated presentation and quantification of the z-stack images would be necessary to support the claims.

We thank the reviewer for pointing out that we did not explain well enough the N-cadherin phenotype and that our claim "NTF is required for the junctional localization of N-cadherin" is misleading. In the Introduction, we provide a detailed description on N-cadherin localization during heart development and cite the relevant literature (page 5, line 110 to 114): "Second, cardiomyocyte crowding triggers their delamination during which adherens junctions between cardiomyocytes are remodeled³⁹. N-cadherin, the major component of cardiac adherens junctions, re-localizes from the lateral to the basal domain in compact wall cardiomyocytes. In contrast, trabecular cardiomyocytes exhibit a punctate N-cadherin distribution on their entire surface³⁹"

In addition, we have now better specified the phenotype. Regarding the N-cadherin localization in Fig. 4 and Fig. S9a (now S11a), we clarify that in the WTs, the N-cadherin localization is limited to the junctions (typical short straight lines between the neighboring cells or Y-shaped patterns; see also original N-cadherin publication by Didier Stainier's group). However, in the stl47 mutants, the N-cadherin localization is random, present still at the junctions but also at the apical and/or basal membrane (therefore, we have added green lines in the schematic at the basal, apical, and junctional sites). To clarify this, we wrote on page 15, line 366, to page 16, line 382: "Single-plane confocal images of WT hearts showed that N-cadherin specifically localizes laterally in the membrane of compact layer cardiomyocytes with some expansion to the basal site (Y-shape pattern) (Fig. 4, Fig. S11a) as previously published³⁹. Maximum intensity projection images confirmed the lateral localization of N-cadherin in WT hearts (Fig. 4, Fig. S11a).

In contrast, in MZ *gpr126^{stl47}* mutant hearts, N-cadherin was distributed randomly, still laterally at the junctions but also at the apical and/or basal membrane of the compact layer cardiomyocytes, which was also obvious in maximum intensity projections (Fig. 4). The maximum intensity projection images exhibit an organized distribution of N-cadherin in WT hearts, however in the MZ *gpr126^{stl47}* mutant hearts, the N-cadherin distribution is haphazard and mis-organized (Fig. S12). Notably, in the MZ *gpr126^{st49}* mutants, which express NTF^{ΔGPS}, N-cadherin was localized in a WT-like pattern at the junctions (laterally) of compact layer cardiomyocytes (Fig.4, Fig. 11a). Overexpression of CTF alone in MZ *gpr126^{stl47}* mutants did not rescue the compact wall integrity defects (Fig. S11b), further indicating the requirements of NTF to mediate cardiomyocyte adhesion. These data indicate that the NTF^{ΔGPS} is sufficient for the proper localization of adherens junctions in cardiomyocytes of the compact wall prior to trabeculation.”

The N-cadherin phenotype is clear and cannot be due to “where confocal sections were chosen” (see figures 4, S11, S12 and the original N-cadherin publication by Didier Stainier’s group). It is an on/off effect and thus is not quantified. To further clarify this point, we have provided Z-projections of 7 or 8 different hearts from WT as well as *stl47* larvae, respectively. These Z-stack images show again clear differences between the WT and *Stl47* hearts (Figure S12). The WT hearts exhibit N-cadherin localization at the junctions suggesting organized adherens junctions, while in the *stl47* hearts, the haphazard/random distribution suggests the disorganization of adherens junctions.

Finally, we agree with the reviewer that the N-cadherin localization defects could be secondary to other compact wall integrity defects. Yet, our data clearly show that *Gpr126* is required for the proper localization of N-cadherin. Consequently, we have changed the title of the manuscript to “*Adgrg6/Gpr126* is required for compact wall integrity and establishing trabecular identity during cardiac trabeculation” and toned down our conclusions in the manuscript and model and stated that NTF is required for compact wall integrity. Here, we would like to emphasize that the aim of our manuscript is to point out that NTF and CTF have different functions during trabeculation, where NTF has a role in the correct formation of the compact wall which is essential for the CTF to play its role in providing trabecular identity to cardiomyocytes. Currently, the most feasible way to look for compact integrity defects is N-cadherin mislocalization.

3. The authors claimed that CTF is required for N-cadherin relocalization. Again, the provided evidence only supports a correlation. There is no direct evidence supporting that CTF regulates N-cadherin localization. It is also unclear whether the impaired N-cad localization is an outcome or cause of the trabecular defect.

We agree that the impaired N-cadherin localization as well as the apico-basal polarity defects could be an outcome or the cause of trabecular defects. However, please note that they are the only cellular mechanisms known so far in zebrafish to identify trabecular cardiomyocytes from non-trabecular cardiomyocytes, and we investigated them to emphasize that the trabecular identity is compromised in *st49* mutants.

To address this point, we have changed the title of the manuscript to “*Adgrg6/Gpr126* is required for compact wall integrity and establishing trabecular identity during cardiac trabeculation” and modified the model and now state that the CTF is required for trabecular

identity. Note, the overall aim of our manuscript is to point out that NTF and CTF have different functions during trabeculation, where NTF has a role in correct formation of the compact wall which is essential for the CTF to play its role in providing trabecular identity to cardiomyocytes.

Reviewer #3:

The authors have made a number of revisions to the original manuscript, including extra experiments to address some of the reviewers' comments. Some of the new findings have confirmed previous findings, whereas others have changed their interpretations. In general, the data are important and will be of interest to the scientific community, especially those working on Gpr126 in different contexts and model systems. However, unanswered questions remain, and the text and some of the interpretations and conclusions are sometimes unclear. Citation of other work is sometimes erratic. There are a number of places where changes could be made to help the reader follow the paper more easily.

We thank the reviewer for pointing out the importance of our work and the additional suggestions how to further improve our manuscript.

Analysis of the transcripts and proteins produced Line 131 and Fig. S1D - It is surprising that neither mutant transcript is subject to nonsense-mediated mRNA decay, and that transcript levels appear normal. Have the authors sequenced variants here? Could there be alternative splicing (possibly heart-specific) or transcription start sites that could compensate for the predicted non-functional transcripts? This might explain the significant proportion of embryos in which trabeculation is normal in both Z and MZ mutants for both alleles. Even small amounts of alternatively spliced zygotic mRNA could result in functional protein.

We thank the reviewer for this very insightful comment.

1) Note, that it has previously shown that *gpr126* mRNA was detected at the same level in WT and *gpr126^{st49}* mutants by *in situ* hybridization utilizing riboprobes spanning exons 12 to 18 (PMID: 19745155, Figure S5), confirming that at least in *gpr126^{st49}* mutants no nonsense-mediated mRNA decay occurs.

To clarify this point, we have added the following text to page 6, line 146: "These data suggest that the mutant *gpr126* mRNAs are not subjected to nonsense-mediated RNA decay, confirming previous data for *gpr126^{st49}* 15."

2) Alternative splicing is a common feature of adhesion GPCR's (PMID: 31363148) and compensation by a heart-specific splice isoform could be a feasible explanation for the variable penetrance of the trabeculation phenotypes in *stl47* mutants as well as the reason why *stl47* zygotic mutants do not show a similar phenotype as *st49* mutants. Therefore, as suggested by the reviewer, we performed RNA sequencing of 3 dpf whole larvae as well as 4 dpf larval hearts for WT, *stl47*, and *st49* to assess the presence of splice isoforms. We looked at the differential exon usage (new Figure S7). Our data suggest that alternative splicing in the heart is not compensating for the predicted non-functional transcripts. In addition, our data show that the predominant *gpr126* isoform in the heart lacks exon counting bin (B) 008, corresponding to exon 6, of the canonical transcript isoform. Notably, skipping of exon 6 results in deletion of 23 aa in

zebrafish affecting gpr126 protein conformation and signaling. Consistently, the construct used in this study for rescuing NTF function lacks exon 6. For details see pages 12 and 13.

Analysis of the tagged proteins in Fig. 1 shows that there is quantifiable GFP fluorescence in cells expressing both mutant versions, although the Western blot is clean. Is the GFP fluorescence above untransfected background levels? If so, are small amounts of protein produced from these transfected constructs?

The GFP fluorescence in the transfected cells were at the same level as the untransfected cells. The quantification never showed the fluorescence value as null, even though there was no detectable fluorescence. The western blot analysis with anti-GFP antibody confirmed that there were no detectable levels of GFP being produced from the two mutant constructs.

Is maternal transcript detectable in WT and MZ stl47 and st49 embryos? As the authors are proposing a role for maternal transcript or protein, an *in situ* should be shown from early (pre-ZGA) embryos to show whether there are detectable levels of maternal mRNA.

It has been described before that 96% of aGPCR have at least some level of maternal expression (PMID: 25715737), including detectable gpr126 expression based on qPCR at 3 hpf, suggesting the presence of maternal mRNA. In addition, we have performed *in situ* hybridization analyses for pre ZGA embryos to visualize the maternal transcript and observed that WT as well as the mutants exhibit gpr126 expression at 2 hpf as shown in Fig. S3b. The data are described on page 9, line 209 to 215: "It has been described that 96% of aGPCR have at least some levels of maternal expression⁴⁶. The qPCR data for Gpr126 show detectable expression at 3 hpf, suggesting the presence of maternal mRNA⁴⁶. To confirm the maternal deposition of gpr126 RNA, we performed *in situ* hybridization in embryos prior to zygotic genome activation to visualize the maternal transcript and observed that WT as well as the MZ gpr126^{stl47} and MZ gpr126^{st49} mutants have gpr126 expression at 2 hpf (Fig. S3a)."

To further clarify the issue of maternal compensation and that this has been described before, we have added the following text to the Discussion (page 19, line 459): "The analysis of the gpr126 mutant zebrafish lines revealed that zygotic stl47 mutants exhibit no trabeculation phenotype. This suggested that maternal gpr126 mRNA can compensate the loss of zygotic gpr126. Consequently, MZ gpr126^{stl47} mutants exhibit defective trabeculation. However, this raised the question how a maternal mRNA can compensate for such an extended time period the lack of zygotic mRNA. Notably, it had already previously been shown that MZ stl47 mutants exhibited a more prominent phenotype regarding Schwann cell development. Importantly, at 96 hpf 42% of the Z stl47 mutants showed normal myelin basic protein expression while none of MZ stl47 mutants expressed myelin basic protein⁴³. Collectively, our data indicate that the maternal gpr126 mRNA and/or the encoded protein stability is high enough to allow normal trabeculation."

Heart phenotypes in Z and MZ *stl47* embryos. It needs to be pointed out more explicitly that 36.67% of MZ *stl47* mutants have WT trabeculation, and over 60% have some degree of trabeculation. This indicates that under some circumstances (possibly for some genetic backgrounds), Gpr126 is not required for heart development. This result is shown in Fig. 1 but is not described explicitly in the text (lines 189–192).

To address this point, we have modified the text and described explicitly the variable penetrance of the phenotypes observed (page 9, line 222): “Thus, in total 63.4% of the MZ *gpr126^{stl47}* mutants exhibited trabeculation defects, whereas 36.6% exhibited normal trabeculation. We did not observe any other gross morphological defects in the MZ *gpr126^{stl47}* mutant larvae or developmental delays compared with WT larvae (Fig. S3b).”

We agree that different genetic backgrounds are responsible for variable phenotype, as known for many genes also in humans. Yet, trabeculation is a dynamic process, and previous analyses of other mutant lines have revealed that it is common to observe different levels of severity regarding trabeculation phenotypes. We suspect that compact wall integrity defects lead to reduced trabeculation in general, and in severe cases leads to a complete absence of trabeculation in MZ *stl47* mutants.

Could the range of phenotypes seen in MZ*stl47* be explained by an overall general delay? What is the gross morphology of these embryos? – this has not been shown anywhere.

We did not observe any overall general delay and provide now in Fig. S3b more data. The text has been changed as follows (page 9, line 223): “We did not observe any other gross morphological defects in the MZ *gpr126^{stl47}* mutant larvae or developmental delays compared with WT larvae (Fig. S3b).”

Secretion of the NTFdelGPS. The authors report that the NTFdelGPS is detected in the cytoplasm, supported by immunostaining and Western blots of cell lysates (Fig. 1D). They say cytoplasmic expression was detected ‘in a minority of cells’. Does this mean only a minority of cells were transfected with the construct, or, of those transfected, only a minority show cytoplasmic expression? The authors now also report secretion of the NTFdelGPS. This is very important given its apparent non-autonomous function on the myocardium, and also because the anti-HA fluorescence and cell lysate blot shown in Fig. 1 imply that it is stuck in the cytoplasm. However, the main figure still shows the apparent cytoplasmic expression, while analysis of the supernatant to show potential protein secretion, which the argument hinges on, is hidden in the supplementary material (Fig. S2C). What is the relative amount of cytoplasmic vs secreted protein? – the Western blot should show the lysate and supernatant bands from the same samples side by side in the main figure.

We thank the reviewer for this important suggestion. We meant “of those transfected, only a minority show cytoplasmic expression”. To further clarify this point, we generated stable ARPE-19 lines overexpressing the WT and the mutant forms and observed that the cytoplasmic localization was significantly reduced in the stable lines (Fig. 1e,f), suggesting that the truncated proteins are not getting stuck in the cytoplasm. The western blot analysis of the lysates and supernatants of WT and *St49* overexpressing cells showed that the NTF^{ΔGPS} is secreted and treatment with brefeldin A markedly blocked the secretion. We have also provided the lysate and supernatant bands for the transient overexpression in Figure S2c.

To clarify this point we have changed the text as follows:

Page 7, line 168: “In contrast, the mutant Gpr126 forms failed to localize to the membrane (Fig. 1b) and in a minority of transfected cells cytoplasmic expression was detected (Fig. S2a, b).”

Page 8, line 179: “Given that transient overexpression of proteins can lead to protein artifacts, we generated stable ARPE-19 lines expressing WT as well as St49 and St47 mutants of Gpr126 for a more reliable analysis (Fig. 1e). We observed membrane localization of WT Gpr126, while the mutant forms failed to localize to the membrane. In addition, there was no detectable cytoplasmic accumulation of the mutant forms, indicating that the truncated forms are not retained in the cytoplasm. WB analyses of the lysates and culture medium from the stable lines, with and without treatment with brefeldin A, confirmed that NTF^{ΔGPS} is secreted in the St49 overexpressing cells (Fig. 1f).”.

Possible dominant effect of the NTF. We do not believe that the authors have sufficiently ruled out a possible dominant effect for the N terminal fragment produced by the st49 allele, a point that was queried by us and another reviewer for the original version. The authors state on Line 221 – ‘no heart phenotypes were observed in heterozygous gpr126st49 mutants.’ It is essential to show the data and quantification of the phenotypes here, as the authors have done for WTs, Z homozygotes and MZ homozygotes. Given the incomplete penetrance of the phenotype in the homozygotes (26.6% of Z and MZ st49 mutants show normal trabeculation), this also needs to be quantified for the heterozygotes. This may require further work and genotyping, but is essential to rule out any dominant effect of the NTF.

We thank the reviewer for indicating that it is not well described in the manuscript why a dominant effect of NTF^{ΔGPS} can be excluded. In the revised manuscript, we have now clearly described why we conclude that the NTF^{ΔGPS} has no dominant effect:

- 1) Z and MZ heterozygous animals of both mutants, stl47 and st49 mutants, exhibit no phenotypes (see Fig. S5 and page 11, line 250)
- 2) CTF overexpression can rescue the gpr126^{st49} mutant phenotype (see Fig. 3 and page 14, line 311).
- 3) Overexpression of NTF^{ΔGPS} in WT animals does not cause a heart phenotype or Notch activation (Figure S13a).

Notably, the experiment presented in the previous manuscript (v1, Fig. S4), in which mosaic overexpression of the N terminus in the WT myocardium appeared to induce local multilayering—by definition a dominant effect—has now been removed from the current version of the manuscript.

We agree that the first conclusion after observing a phenotype upon gene overexpression might be that there is a dominant effect. Yet, there are also other explanations. We thank the reviewer for forcing us to think more about the data to come up with an explanation, as based on our data a dominant effect can be excluded (see above). If one considers the results of all mutant and rescue analyses, the fact that st49 mutants lacking the CTF exhibit enhanced Notch activity but NTF^{ΔGPS} overexpression in WT does not, as well as the role of the maternal RNA, one possible explanation is that the CTF is required to balance NTF-mediated signaling preventing overactivation of myocardial Notch signaling.

To clarify this point, we have added the following text to the Discussion on page 20, line 467, which reads: “Collectively, our data indicate that the maternal *gpr126* mRNA and/or the encoded protein stability is high enough to allow normal trabeculation. However, this raises the question, why maternal *gpr126* mRNA/protein is not sufficient for normal trabeculation in *gpr126^{st49}* mutants. The only difference is the presence of additional NTF^{ΔGPS}. Thus, one possible explanation is that NTF^{ΔGPS} acts in a dominant manner. However, our data argue against a dominant effect as for example heterozygous *St49* mutants show no phenotypes. Another possible explanation is that a balance of NTF- and CTF-mediated signaling is required for proper trabeculation. This is supported by the observation of a substantial increase of Notch activity in MZ *gpr126^{st49}* mutants which could be normalized by the overexpression of CTF. Overall, all lines with an increased expression of NTF compared to CTF exhibit the multilayering phenotype with the exception of the NTF^{ΔGPS} overexpression line (Fig. S13a). Notably, in this line cells express zygotic and maternal CTF and NTF and thus the ectopic expression of NTF might not be high enough to significantly impair the balance between NTF and CTF or enough CTF is present to prevent increased Notch activity. In contrast, mosaic expression, usually mediating a markedly higher expression was sufficient to induce local multilayering. In addition, the mosaic expression might result in adhesion heterogeneity between cardiomyocytes. Overall, our data suggest that the CTF is required to balance NTF-mediated signaling preventing overactivation of myocardial Notch signaling.”

Link to Notch signaling. 20% of MZ *stl47* mutants show a multilayering phenotype (Fig. 2D). Does this population also show an increase in Notch signalling?

We did not observe any *stl47* mutants with enhanced Notch signaling. We visualized active Notch signaling in *stl47* mutants with a membrane reporter line and did not observe increased myocardial Notch signaling in the *stl47* mutants exhibiting multilayering, as shown in the figure below.

Model - early and late requirements. The authors state in their rebuttal that there is an early requirement for the NTF in N-cad localisation (a prerequisite for trabeculation), and a late requirement for the CTF in trabeculation. They say this difference in timing explains the difference in phenotype between the two alleles. It is stated several times in the MS and in the rebuttal that the maternal protein is unlikely to perform the later function. However, in the Z stl47 mutants, both early and late roles would have to be performed by the maternal transcript or protein, resulting in normal trabeculation. Action of the maternal transcript for such a late function(96h+) is really puzzling. Thus, the proposed early/late functions in the model do not fit with the data shown. If the maternal protein can perform both early and late functions (as demonstrated by the Z-only stl47 normal phenotype), then why is it unable to do this in the presence of zygotically-produced NTFdelGPS in the Z st49 mutants? This is still unexplained, unless the NTF blocks the effects of the maternal protein in a dominant manner, or there is some heart-specific compensation mechanism in the stl47 allele. The authors do acknowledge that there are difficulties in their interpretation and that further work is needed at Line 413, but this should be stated up-front more explicitly, and the model should either be revised or presented with much more caution.

We thank the reviewer for stressing this point which made us re-evaluating all data and re-thinking our conclusions. As detailed in our response to the points above we conclude that our data suggest that a balance of NTF and CTF is required (see above and Discussion on page 20, line 467).

Pharmacological rescue experiments: Paragraph starting at Line 255. It is really important to cite Petersen et al. (2015) here for pharmacological rescue of the stl47 and st49 alleles. These authors only used forskolin, and importantly, showed that this could not rescue radial sorting or myelination in stl47 mutants. For use of IBMX, Geng et al. (2013) is cited, which is appropriate. However, it should be clarified here that the published use of IBMX was with other alleles, not stl47 or st49. In addition, the concentration used in this paper was 100 μ M, not 125 μ M. Finally, IBMX is not an AC activator - it is a phosphodiesterase inhibitor, raising cAMP levels independently of AC activity and thus an important way to confirm that the CTF works through cAMP. It is important to get these details correct, as they are required to interpret the allele-specific and domain-specific effects for these phenotypes, which are not straightforward.

Thank you for spotting this. We have changed the text accordingly and added the required details regarding IBMX and forskolin (see page 15, line 346).

Model – autonomy and non-autonomy: The authors have still not explained how expression and activity of the CTF in the endocardium can lead to effects on the myocardium, i.e. how the receptor acts in a non-autonomous manner. In other tissues (ear and Schwann cells), the phenotypes affect the cells in which the receptor is expressed. In the heart this appears to be different, but no suitable explanation is provided. In this context, it is really important to point out the cell types in which the experiments were done. For example, the over-expression of the NTFdelGPS (lines 215–217) is done in the myocardium, whereas production of the NTF from the st49 allele is expected to be in the endocardium.

Note, we have previously shown that the NTF binds to the myocardium in mouse hearts and thus, we have utilized a cardiac promoter to express NTF. In the revised manuscript, we have provided this information and made sure that it is clearly stated in which cell type which experiments were

performed (page 11, line 260): “Therefore, we analyzed mosaic cardiac expression of NTF^{ΔGPS} in WT larvae as well as stable overexpression of NTF^{ΔGPS} in cardiomyocytes (Tg(myI7:gpr126NTF-p2a-tdTomato) in MZ gpr126^{stt47} mutants, as our previous data indicate that NTF acts as a secreted molecule on cardiomyocytes²².

Regarding the mechanism how the endocardial CTF affects the fate of cardiomyocytes, we have provided possible explanations in the Discussion (see page 23, line 556). Importantly, “how endocardial factors affect cardiomyocyte behavior remains largely unknown and is currently under investigation. Recently it has been suggested that endocardial protrusions regulate communication between endocardium and myocardium by modulating Nrg/ErbB2 signaling^{58,59}”

Other comments and essential corrections

The sentence at Lines 52–53 is an exact copy of a sentence in the Abstract from Zaidman et al., PNAS USA 2020, and should be reworded.

We have reworded the sentence.

The citations of references 18–20 supporting the information on Lines 66–70 are three primary papers covering pulmonary function, COPD and periodontitis. These references do not cover the other diseases that the authors refer to (scoliosis, intellectual disability, cancers). The authors should either cite all relevant primary papers to support their statements, or they should cite a suitable review that covers all this information with comprehensive referencing.

We have cited the review from Baxendale et al (PMID: 33735533) which cover all the diseases which were referred.

Lines 88–89 ‘newly created zebrafish mutants’ (reference 17, Torregrosa-Carrion et al., 2021). It is essential to give full details of these alleles; one (bns342) is a promoter-less allele with undetectable transcript levels and normal cardiac trabeculation, and so it is important to point this out. In addition, this study did not rely on ‘gross morphological analyses’ - it presents confocal images (ref. 17, Fig. 3) at the same level of detail as the current manuscript. This information should be acknowledged, described and cited correctly.

We have changed the text to add more details about the alleles (see page 4, line 101).

Line 74 – depletion, not deletion

We have changed the text, as suggested.

Lines 121–127 Introduction to the alleles used. The authors now cite the original papers at the start of this section, and have corrected the amino acid codon affected by the lesion, but still go on to present the data in the supplementary figures identifying the molecular lesions as their own. It would be best to repeat the citations throughout the paragraph to make it clear that this figure is confirming previously published data, or to explain up front that these alleles have previously been characterized and that the figure is confirming these published data.

We have repeated the citations and added text to state that we are confirming previously published data (see page 6, line 135).

Line 162 As the experiments here are done in cells, and not in embryos, the wording should be changed here to say 'These data suggest that *gpr126^{st49}* mutants express and secrete the NTFdelGPS.' (Not 'indicate'.)

We have changed the text, as suggested (see page 8, line 187).

Line 170–183 – The title of this section is about *stl47*. However, the important result of the multilayering in *st49* is also introduced here. It would help to have one section about *stl47* and then another about *st49*.

As recommended, we have introduced in the revised manuscript two sub-sections to describe the phenotypes of the two mutant lines separately.

Line 200 – it is stated that there are no obvious defects in the endocardium. However, it is not folded as normal, and this could be interpreted as a defect.

We agree with the reviewer that there is a difference in the folding. However, it should be noted that the endocardium and myocardium are in very close contact to each other during the analyzed stages and thus the missing folding appears to be due to the lack of trabeculation.

To address this point, we have added the following text to page 10, line 243: "In contrast, no obvious defect in the endocardium or the cardiac jelly between the endocardium and the myocardium was detected at 72 and 96 hpf, when analyzing MZ *gpr126^{stl47}* and *gpr126^{st49}* mutants in the background of Tg(*myl7:mKATE-CAAX*); Tg(*krt4:GFP*)^{sqet33 46}, which label the myocardium and the endocardium, respectively (Fig. S4). The only apparent difference is that the endocardium is not folded as normal, which appears to be due to the lack or a reduced number of trabeculae."

Line 209 – 'experiments previously suggested that the NTFdelGPS is sufficient for trabeculation and heart development (ref. 21). It would help to say here something like 'However, we are now revising that view'.

We thank the reviewer for spotting that we did not clarify well that we revised our previous conclusion that the NTF is sufficient for proper heart development. In the revised manuscript we have clarified/discussed this point on page 11 (line 255), page 11 (line 272), and page 21 (line 508).

Line 212 – this should be Z, not MZ

We have corrected the text accordingly.

Figure 4 - N-Cadherin data: These images are still difficult to interpret. It would be good to show individual channels in inverted greyscale to show as much detail as possible.

In our opinion, the main problem is that we have not properly described the N-cadherin phenotype and do not think that inverted greyscale images will help.

To address this point, we provide in the Introduction a detailed description on N-cadherin localization during heart development and cite the relevant literature (page 5, line 110): "Second, cardiomyocyte crowding triggers their delamination during which adherens junctions between cardiomyocytes are remodeled³⁹. N-cadherin, the major component of cardiac adherens junctions, re-localizes from the lateral to the basal domain in compact wall cardiomyocytes. In

contrast, trabecular cardiomyocytes exhibit a punctate N-cadherin distribution on their entire surface³⁹.”

In addition, we have now better specified the phenotype. Regarding the N-cadherin localization in Fig. 4 and Fig. S9a (now S11a), we clarify that in the WT, the N-cadherin localization is limited to the junctions (typical short straight lines between the neighboring cells or Y-shaped patterns; see also original N-cadherin publication by Didier Stainier’s group). However, in the *stl47* mutants, the N-cadherin localization is random, present still at the junctions but also at the apical and/or basal membrane (therefore, we have added green lines in the schematic at the basal, apical, and junctional sites). To clarify this, we wrote on page 15, line 366, to page 16, line 382: “Single-plane confocal images of WT hearts showed that N-cadherin specifically localizes laterally in the membrane of compact layer cardiomyocytes with some expansion to the basal site (Y-shape pattern) (Fig. 4, Fig. S11a) as previously published³⁹. Maximum intensity projection images confirmed the lateral localization of N-cadherin in WT hearts (Fig. 4, Fig. S11a). In contrast, in MZ *gpr126^{stl47}* mutant hearts, N-cadherin was distributed randomly, still laterally at the junctions but also at the apical and/or basal membrane of the compact layer cardiomyocytes, which was also obvious in maximum intensity projections (Fig. 4). The maximum intensity projection images exhibit an organized distribution of N-cadherin in WT hearts, however in the MZ *gpr126^{stl47}* mutant hearts, the N-cadherin distribution is haphazard and mis-organized (Fig. S12). Notably, in the MZ *gpr126^{stl49}* mutants, which express NTF^{AGPS}, N-cadherin was localized in a WT-like pattern at the junctions (laterally) of compact layer cardiomyocytes (Fig.4, Fig. 11a). Overexpression of CTF alone in MZ *gpr126^{stl47}* mutants did not rescue the compact wall integrity defects (Fig. S11b), further indicating the requirements of NTF to mediate cardiomyocyte adhesion. These data indicate that the NTF^{AGPS} is sufficient for the proper localization of adherens junctions in cardiomyocytes of the compact wall prior to trabeculation.”

To further clarify this point, we have provided Z-projections of 7 or 8 different hearts from WT as well as *stl47*, respectively. These Z-stack images show again clear differences between the WT and *Stl47* hearts (Figure S12). The WT hearts exhibit N-cadherin localization at the junctions suggesting organized adherens junctions, while in the *stl47* hearts, the haphazard/random distribution suggests the disorganization of adherens junctions.

ErbB2 section (Lines 267–290): All data for this section are presented in the supplementary material. This is still too preliminary and it is recommended that this section is removed from the manuscript.

We have removed this data from the manuscript.

Point-by-point response

Reviewer #2 (Remarks to the Author):

The authors have addressed the concerns raised in the previous review. I appreciate that they have toned down the language in several conclusions, adopting a more cautious and balanced approach in interpreting the phenotypes and addressing the discrepancy with previous work. These revisions have improved the clarity and rigor of the manuscript.

We thank the reviewer for the many constructive comments and suggestions that greatly helped to improve the manuscript.

Reviewer #3 (Remarks to the Author):

The authors have added several new supplementary figures to the MS, including new data, and have addressed many of the comments satisfactorily. These include some interesting new findings relating to the prevalence of the exon-6-skipped transcript isoform in the heart. As stated previously, the study is important and will be of value to the research community.

Thank you very much. We would like to thank the reviewer for acknowledging the importance of our work and for the constructive suggestions to improve and refine our manuscript.

However, there are still a number of instances where the data and/or interpretations appear inconsistent or incomplete, and these may affect the way that the overall mechanistic model is presented. These issues are detailed below.

Thank you for spotting the inconsistencies of our manuscript. We have addressed all the remaining points as detailed below.

1. Other alleles

a) It is good that the authors have added details of other mutant alleles when discussing previous work. However, the Torregrosa-Carrión et al., 2021 study is still misquoted, as it was not 'based on gross morphological analyses' in the added text at lines 99–100 - this statement needs to be removed. The Torregrosa-Carrión paper presented detailed confocal imaging of trabeculae in transgenic lines at 96 hpf, together with quantitative analysis of transcript levels at 48 hpf (Torregrosa-Carrión et al., 2021, Fig. 3).

We agree that “gross morphological analyses” is misleading. What we meant was that *Torregrosa-Carrión et al.* analyzed the mutants only based on the labeling of endocardial and myocardial membranes and, in contrast to our study, did not utilize additional markers such as N-cadherin or markers to visualize depolarization.

To address this point, we have removed the statement “based on gross morphological analysis” (page 4, line 100).

*b) The discrepancy with the current study can be explained possibly because this study only examined Z-only mutants, although even this is problematical, because the authors demonstrated a complete loss of transcript in zygotic *bns342/bns342* mutants at 48 hpf (T-C et al., 2021, Fig. 3B), and concluded that *gpr126* is not required for trabeculation in the zebrafish heart. (It will obviously be interesting to test whether there is a heart phenotype in MZ *bns342/bns342* mutants.)*

We agree with the reviewer that the discrepancy in both studies might be explained by the fact that *Torregrosa-Carrión et al.* studied only zygotic mutants. That the authors did not observe a phenotype can have several reasons (e.g., also compensation). Notably, *gpr126*^{*bns342/342*} embryos have an inflated swim bladder in their Fig 3E, in contrast to the *gpr126*^{*bns342/342*} embryo shown in their Fig. S10, whose appearance resembles our mutants (Fig. S1c). Clarifying why *Torregrosa-*

Carrión et al. did not observe a heart phenotype in their *gpr126* mutants is beyond the scope of this manuscript.

To address this point, we have added the following sentence to the Discussion (page 22, line 514): “It will be interesting to test whether there is a heart phenotype in MZ *gpr126*^{*bns341*} and MZ *gpr126*^{*bns342*} mutants.”

2. Figure S3b

*As requested, the authors have added Fig. S3b to show the whole-embryo appearance for *stl47* and *st49* alleles. However, their description in the text and legend is at odds with the images shown. Gross abnormalities are evident for the *stl47* allele in the figure: shorter body length, small head and eye, underdeveloped jaw, and pericardial oedema. If these phenotypes are representative, they need to be described. If not representative, another embryo should be shown. Note that incompletely penetrant pericardial oedema was previously reported for zygotic *stl47/stl47* mutants (Fig. S1c, Petersen et al., 2015), matching the phenotype of the *stl47* embryo in Fig. S3b. This published observation should be acknowledged and cited somewhere. An additional issue with Fig. S3b is that the embryos look completely different to those shown in Figs. S1c and S10a, also labelled as 96 hpf. The puffy ears are not nearly as evident for either genotype, and thus it is not convincing that these fish are actually mutant for *gpr126*. We suggest that they are likely to be 72 hpf or younger – please check.*

a) We sincerely thank the reviewer for spotting these points. The embryos shown in former Fig. S3 were actually 72 hpf embryos.

To address this point, we have now added images of 96 hpf embryos to Fig. S3. The majority of mutant embryos showed neither an obvious shorter body length, smaller head and/or eye, or underdeveloped jaw.

b) It is correct that we observed sporadically cardiac oedema in the MZ *stl47* mutants (not appearing in all clutches), as described in *Petersen et al.* (2015, Fig. S1c). Embryos with strong cardiac oedema were excluded from this study.

To address this point, we have added the following two sentences to the Results (page 10, line 229): “Note, a pericardial oedema was sporadically observed in MZ *gpr126*^{*stl47*} mutants, as previously reported⁴³. Embryos with strong pericardial oedema were excluded from this study.”

3. Figure S6

*The authors state that they have previously shown that the NTF binds to the myocardium in mice as a justification for expressing it in the myocardium in zebrafish. However, they also confirm that *Gpr126* is normally expressed in the endocardium and that the NTF is secreted. It would have been more relevant to express the NTF in endocardium, to mimic the situation in the mutants more closely. Expressing it in the myocardium adds an additional level of complexity to the interpretation. Do you see similar phenotypes with endocardial (*krt4*-driven) or endothelial (*fli1*-driven) expression of the NTF?*

The aim of our experimental approach was actually to reduce the complexity. As we currently do not know how NTF is secreted from the endocardium and transported to the myocardium and whether the CTF is involved or required in these processes, we decided to express NTF^{ΔGPS} in cardiomyocytes, as we had previously identified these cells as a target of NTF. In addition, we argued that if we see an effect, this will further support our hypothesis that the NTF acts on cardiomyocytes.

While the endocardial expression of the NTF could further verify the secretory role of NTF on the myocardium, we believe that these experiments will not add substantial information to the manuscript and thus are beyond the scope of this manuscript.

Lines 260–262 The wording here is difficult to follow. Suggest changing to:

*"Therefore, we analyzed myocardial-specific (*myl7*-driven) mosaic expression of NTF Δ GPS in WT larvae as well as stable expression in MZ *gpr126stl47* mutants,"*

As requested, we have changed the text, however, we changed the order as we first show data for stable expression and then for mosaic expression (page 11, line 269).

If the data from the stable line are available for MZ mutants, it is important to show the effects of expression of this line in sibling embryos (Z mutants and WTs), as these will be more relevant controls for this experiment than the mosaic overexpression experiment.

To address this request, we have added a representative image of WT embryos overexpressing NTF Δ GPS (*Tg(myl7:gpr126NTF-p2a-tdTomato)*) at 96 hpf in Fig. S6a. Stable NTF overexpression in WT embryos did not lead to any multilayering (Fig. S6b). The text has been changed accordingly (page 12, line 273).

Fig. S6b. In Fig. 2d, it is stated that 20% (n=30) of MZstl47 hearts have a multilayered ventricular wall. In Fig. S6b, this proportion has now halved to 10% (n=8), and this value is used for comparison with the over-expression result (40%, n=10). However, is not clear how the percentage values for this graph in Fig. S6b were generated. Why are the data not presented like the stacked bars in Fig. 2? Given the low numbers for the Fig. S6b experiment (did the data pass a normality test?), and the Fig. 2d estimate of 20% (more accurate as based on higher numbers), obtaining 4/10 embryos with multilayering by chance is very possible. Thus, as they stand, the results shown in Fig. S6b are not convincing.

We thank the reviewer for pointing out this inconsistency. We have revisited the images, quantified them according to Fig. 2d, added one more batch of stl47 MZ embryos, and presented the data as stacked bars in Fig. S6b, similar to Fig. 2d. These data demonstrate that NTF overexpression in stl47 MZ embryos enhances multilayering.

4. Figures 4, S11, S12

The description of the N-cadherin phenotype in MZstl47 mutants is now clearer, although it is still quite hard to see the differences between wild-type and mutant junctional arrangements in the images, and this was also pointed out by another reviewer. The additional images in Fig. S12 are helpful, although we still think that enlargements of selected regions of the MIPs would be useful. However, if the original images are retained, it would be helpful to include some labelling on the MIP images to highlight the main important features.

To address this point, we have added to Fig. 4 zoom-ins of the maximum projections. In addition, we labeled the outline of the cardiomyocytes and utilized arrows and arrowheads to clarify the phenotype. One can clearly see that WT and st49 hearts exhibit clear boundaries of the N-cadherin localization at the lateral junctions, while in stl47 mutants, the boundaries are blurred and they appear more haphazard.

No additional images of MZst49 were provided, and it is stated that the MZst49 hearts have a wild-type arrangement of N-cadherin (line 377). To us, the MIP image of MZst49 (Fig. 4, lower right panel) has features that look more similar to the MZstl47 phenotype, especially towards the bottom and right hand side of the ventricle. If a phenotype is present, this would change the interpretation here (line 381), instead indicating that the NTF is not sufficient for the correct localisation of N-cadherin in the compact layer. Can the authors comment and add some additional images of the MZst49 hearts (it is stated that 12 samples were imaged) to Fig. S12, as for MZstl47?

We hope that the zoom-ins in Fig. 4 help the reviewer to understand the phenotype. In addition, we have added 7 additional examples of MZ st49 hearts in Fig. S12.

We would like to point out that the maximum projections are an additional way to visualize the N-cadherin mis-localization phenotype, which might be not as obvious as the phenotype shown in mid-sagittal sections. Both representations of the phenotype together clearly show that the N-cadherin is mis-localized in the *stl47* hearts.

Note, there is some variability in the maximum projection data as trabeculation is a dynamic process which starts at around 60 hpf. Thus, there might be already some sparse trabecular cardiomyocytes at 72 hpf which affect the appearance of clear boundaries. Furthermore, multilayering defects in *st49* mutants might also affect the maximum projections explaining the slight differences in N-cadherin localization at the lateral junctions between WT and *st49* hearts. Notably, mid-sagittal sections in WT and *st49* embryos clearly show that their hearts do not exhibit any mis-localized N-cadherin in the compact wall cardiomyocytes.

The mis-localization of N-cadherin in the compact wall cardiomyocytes in *stl47* mutants show that NTF is required for the correct localization of N-cadherin in the compact wall, whereas in *st49* mutants, the N-cadherin is correctly localized at the lateral side in compact wall cardiomyocytes, but fail to relocalize to the basal membrane in the inner layer cardiomyocytes, as observed in trabecular cardiomyocytes, suggesting the CTF is required for trabecular identity.

5. Model (Figure 7)

The authors have revised their interpretation of the data, but the summary model in Fig. 7 remains unchanged, so it is hard to understand exactly what is new. It must be stated in the figure legend (as well as the text) that this is just a model for the role of Gpr126, as it incorporates a number of assumptions. Some basic details are unknown at the moment, such as any polarised location of the receptor on the endocardial cell plasma membrane. It seems unlikely that all NTF would be secreted and degraded, when the CTF persists – again, these details have not been tested.

As requested, we have changed the Figure legend (including the title) of Fig. 7 as well as the Discussion (page 25, line 591) to clarify that our conclusion represents a model that has the indicated limitations.

What do the dotted black arrows represent? Is this a second unidentified signal? This relates to our earlier comment that the autonomy and/or non-autonomy of action should be clarified (terms that the authors have avoided, but seem to be relevant for the interpretation of the data). The authors propose a non-autonomous role for both the NTF and the CTF, expressed in endocardial cells, on the myocardium. For the CTF, such a non-autonomous role is unexpected and would be likely to need a second signal that can reach the myocardium. The authors do mention possible ways in which the endocardial cells could contact the myocardium across the cardiac jelly, but this is not depicted in the diagram.

The dotted black arrows were indicating that the CTF is required to determine the fate of trabecular vs non-trabecular cardiomyocytes but the signaling to the myocardium is unknown.

To address this point, we have removed the arrows and added to Fig. 7 the information that “it has been suggested that endocardial protrusions regulate communication between endocardium and myocardium by modulating Nrg/ErbB2 signaling”, which is also stated in the Introduction (page 24, line 576). In addition, we added to the figure legend of Fig. 7: “The endocardial cells form protrusions to interact directly with the cardiomyocytes and facilitate CTF signaling to act on the myocardium.”

The diagram also does not help to explain what the authors mean by a ‘balance’ between the NTF and CTF, which now seems to be a key feature of their model, but is not mentioned in the figure legend.

This is described in detail on page 20, line 485. To address this point, we have added the following text to the figure legend of Figure 7: “Our data suggest that the CTF is required to balance NTF-mediated signaling preventing overactivation of myocardial Notch signaling.” and to the

Discussion (page 25, line 591) “The NTF/CTF balance is essential for the Notch-mediated lateral inhibition, which is integral for the myocardial pattern formation (Fig. 7).”.

In addition, the Notch signalling in Fig. 7 is shown in the myocardium, whereas in the Introduction and Discussion, it is stated that endocardial Notch signalling is important for trabeculation (line 108, 558–9). The authors do cite work on Notch expression in cardiomyocytes at line 417, but this needs a more detailed description, including known timing of events.

To address this point, we have added more information to the Introduction and the Discussion:

- Page 5, line 107: “. First, endocardial Notch signaling, a critical early regulator of trabeculation²⁹⁻³¹, which in zebrafish is activated in the endocardium at 24 hpf and becomes restricted to atrioventricular valves and the outflow tract by 72 hpf³²,...”
- Page 5, line 120: “The first appearance of Notch activity in ventricular cardiomyocytes overlaps with the onset of trabeculation (60 hpf).”
- page 24, line 578: “and heterogeneous expression of *erbb2* in the single layer myocardium leads to tension heterogeneity initiating trabeculation, which triggers Notch activity in adjacent cardiomyocytes, suppressing *erbb2* preventing their delamination⁶⁴.”

In brief, endocardial Notch signaling is an activator of trabeculation, whereas myocardial Notch activity inhibits the delamination of cardiomyocytes.

*According to the imaging with *Tg(TP1:VenusPest)* in Fig. 6, levels of Notch signalling in the myocardium in WTs are normally low – the data here do not match the expression patterns sketched in Fig. 7. Some more explanation here, together with timing/embryo stages on the diagram, would be helpful.*

Note, the myocardial Notch activity is only activated in compact wall cardiomyocytes. In addition, the here utilized standard Notch reporter line *Tg(TP1:VenusPest)* drives the expression of destabilized VenusPEST protein, which has a very short half-life. Thus, only cells with active Notch signaling are labelled. This is the reason, why the levels of Notch signaling in the myocardium are low.

In our model (Fig. 7), we show Notch activity in the two cardiomyocytes next to a delaminating cardiomyocyte. To clarify the timing, we have now added the time periods of compact wall formation (48-72 hpf) and trabeculation (72-120 hpf).

6. Other minor (but important) comments/corrections:

- *Line 70 ‘the membrane’ should be ‘the plasma membrane’*

We have changed the text accordingly (page 3, line 71).

- *Lines 103–119 – It would help to outline the normal timing of cardiac trabeculation events and Notch signalling in the different cell types in this paragraph, so that the reader can relate them to the stages used in the experimental work.*

As requested, we have now expanded the paragraph about cardiac trabeculation in the Introduction providing information on the timing of cardiac trabeculation as well as myocardial and endocardial Notch signaling (page 4 line 103).

- *Line 138 – predicting, not resulting*

We have changed the text accordingly (page 6, line 142).

- *Line 193 – it says 98 amino acids here, but 96 earlier in the MS and on the figure*

Thank you for spotting this. We have corrected the text to 96 (page 8, line 197).

• *Lines 208–209 – a reference is needed to support this statement.*

We have added the reference (page 9, line 214).

• *Line 219 et seq. – it would help to state the corresponding n values here in the text with the % values, as well as in the figure legend.*

We have added the information to the main text (page 9 line 224). The n numbers were already provided in the figure legend. In addition, we have provided all n values for all experiments in the Source Data.

• *Line 231–232 ‘Gpr126 is required for cardiac chamber development in zebrafish’ - As 1/3 of MZ mutants of both genotypes have normal trabeculation, this statement should be qualified here to indicate that the phenotype shows incomplete penetrance and variable expressivity.*

To address this point, we have changed the text to: “These data indicate that maternal contribution of Gpr126 rescues the zygotic loss in around two third of *gpr126*^{st47} mutants, indicating incomplete penetrance, and that Gpr126 is required for cardiac chamber development in zebrafish.” (page 10, line 238).

• *Line 354-356: Suggest rewording here to highlight the rescue aspects of this experiment, as well as describing the data in the order that they appear in the figure, e.g. "Analysis of the heart revealed that both treatments could partially rescue the multilayering phenotype in MZ *gpr126st49* mutants (Fig. S10b, upper panels). However, both treatments also blocked trabeculation, both in MZ *gpr126st49* mutant hearts and in WT hearts (Fig. S10b, lower panels)."*

We thank the reviewer for the suggestion and have replaced the text accordingly (page 15, line 364).

• *Line 368 – side, rather than site*

We have changed the text accordingly (page 16, line 380).

• *Line 378 – Fig. S11a, not Fig. 11a*

We have changed the text accordingly (page 16, line 390).

• *Line 553–4 – give authors and year for the bioRxiv paper, and list full details and link in the Bibliography, as for other references.*

This paper is now published in *Developmental Cell* and we have cited it accordingly (page 24, line 568).

• *Line 574 – births*

We have changed the text accordingly (page 25, line 600).

• *Fig. 1a, Fig. S7a The schematic diagram in Fig. 1a is not drawn to scale, and the term 'partial GAIN' is not explained anywhere. The similar diagram of the protein structure in Fig. S7a appears to be much more accurate in terms of scale - why not use this as a template to generate the diagram of the tagged protein in Fig. 1a? It would help the reader if both diagrams matched each other. Note, however, there are a couple of typos in Fig. S7a - the allele name is st49, not stl49, and only 6 TM domains are depicted. The st49 mutation is also shown at a different position relative to the GPS motif than that in Fig. 1a. This should be positioned as accurately as possible (and consistently) in both diagrams.*

We thank the reviewer for their suggestion. We have now modified the schematic diagram in Fig. 1a using the Fig. S7a *gpr126* scheme as a template and have corrected the typos in Fig. S7a.

• *Fig. 1b, e – Position of scale bars does not match between figure and legend.*

We have placed the scale bars in both Fig1b and 1e and updated the figure legends.

• *Expression of constructs in ARPE-19 cells and BFA treatment. Fig. 1f (stable expression) and Fig. S2c (transient transfection) – It is tricky to compare these figures as the lanes are presented in different orders. The position of bands relative to the size markers is inconsistent (WT bands are at or above the 115kDa marker in Fig. 1f, but well below it in Fig. S2c). Also, loading controls (e.g. of a constitutively secreted protein) are needed to confirm the reduction seen with BFA (for both figures).*

Thank you for spotting this. We have now presented the lanes in both figures in the same order. In addition, we have corrected the position of the size markers.

• *Fig. 3 – order of panels (a, d, b, c, e, f) is very difficult to follow.*

We have put the panels in order.

• *Fig. 3b and Line 326 - Is the P2Y12 N-terminus peptide the domain labelled P2A in the figure? There is no explanation of this in the figure legend.*

Thank you for spotting this lack of information. No, the P2Y12 N-terminus peptide is not the domain labelled P2A in Fig. 3b. The P2A peptide is the self-cleaving peptide present between the Gpr126 CTF and the tdTomato coding sequences which allows for cleavage right after protein translation to separate the Gpr126 CTF from the tdTomato fluorescent protein. The P2Y12 N-terminus peptide is attached to the HA tag to enhance the localization of the CTF to the membrane.

To address this point, we have changed the figure legend of Fig. 3b to: “Schematic of the construct used in *Tg(fli1a:gpr126CTF-p2a-tdTomato)*. The p2a peptide allows for cleavage right after protein translation separating Gpr126 CTF and tdTomato fluorescent protein. The P2Y12 N-terminus peptide (blue line between HA tag and GPS) is fused to the HA tag to facilitate membrane localization of Gpr126 CTF.”

• *Fig. 4, 5a schematics – it would help to include the endocardial layer in these diagrams.*

As requested, we have added the endocardial layer in the schematics in Fig. 4 and 5a.

• *Fig. 5A schematics, Fig. S6c – These figures will be difficult to interpret for anyone with a red-green colour vision deficiency.*

We have changed the colors to be easily interpreted by people with color vision deficiencies.

• *Fig. S6 – the data are not presented in the order they are described in the text. It would be easiest to follow if the mosaic WT data were presented first in the figure, following the order of the description in the text.*

Actually, the data are presented in the figure as they are described in the text. We first describe the stable overexpression of NTF^{ΔGPS} in WT and *gpr126^{st147}* larvae (Fig S6a, b) and then describe the mosaic WT data (Figure S6c).

• *Fig. S6b legend - should say ‘Quantification of a’ (not b).*

We have changed the text according to the changes made to the figure (new quantification).

• *Fig. 6 – title and legend to figure: Notch should have an upper case N*

Thank you. We have changed the text accordingly.

• *Fig. S10 – the title for this figure says that ‘Forskolin and IBMX treatment impairs trabeculation in ... MZ *gpr126st49* hearts’. But trabeculation in these mutants is impaired anyway (Fig. 2a), so*

the title here is not an accurate reflection of the findings of the experiment. The text (line 355) instead says that the treatment rescues multilayering. Please clarify.

Thank you for spotting this. We have changed the title to “Forskolin and IBMX treatment rescues multilayering in MZ *gpr126*^{st49} hearts but impairs trabeculation in WT hearts.”

• Fig. S10a - there is some developmental delay in the treated embryos in this figure (head shape, jaw development and remaining yolk volume). This could account for some of the differences seen in the ears and hearts as well.

Please note, WT and mutant embryos were treated with forskolin (2x 6 h pulses, at 60-66 hpf and 84-90 hpf) at 25 μ M or IBMX (3-isobutyl-1-methylxanthine) (continuous treatment, 60-96 hpf at 125 μ M in 1x E3 medium with 0.003% PTU. Thus, an effect on development should be limited.

• Fig. S13a – as membranes are not imaged here, it is difficult to tell whether or not there is any multilayering phenotype in the WT hearts. As mosaic expression does result in multilayering (Fig. S6c), it is important to show whether there is any multilayering present, in addition to the lack of Notch activation.

This point is addressed by the changes made to Fig. S6. We have added a representative confocal image of WT hearts with stable overexpression of NTF Δ GPS in the myocardium ((*Tg(myI7:gpr126NTF-p2a-tdTomato)*) crossed with the cardiomyocyte membrane reporter line *Tg(myI7:EGFP-hsa.HRAS)* and quantified trabeculation phenotypes. These data show that myocardial NTF Δ GPS overexpression of in WT hearts does not cause multilayering.